# Deep Clustering in Subglacial Radar Reflectance Reveals Subglacial Lakes

Sheng Dong[2,3,5], Lei Fu[1,*], Xueyuan Tang[2,4,*], Zefeng Li[3], and Xiaofei Chen[5]

[1]Hubei Subsurface Multiscale Imaging Key Laboratory, School of Geophysics and Geomatics, China University of Geosciences, Wuhan, China
[2]Key Laboratory of Polar Science, MNR, Polar Research Institute of China, Shanghai, China
[3]School of Earth and Space Sciences, University of Science and Technology of China, Hefei, China
[4]School of Oceanography, Shanghai Jiao Tong University, Shanghai, China
[5]Department of Earth and Space Sciences, Southern University of Science and Technology, Shenzhen, China

**Correspondence:** Lei Fu (fulei@cug.edu.cn) and Xueyuan Tang (tangxueyuan@pric.org.cn)

**Abstract.** Ice-penetrating radar (IPR) imaging is a valuable tool for observing the internal structure and bottom of ice sheets. Subglacial water bodies, also known as subglacial lakes, generally appear as distinct, bright, flat, and continuous reflections in IPR images. In this study, we use available IPR images in the Gamburtsev Subglacial Mountains to extract one-dimensional reflector waveform features of the ice-bedrock interface. We apply a deep learning method to reduce the dimension of the reflector features. An unsupervised clustering method is then used to separate different types of reflector features, including a reflector type corresponding to subglacial lakes. The derived clustering labels are used to detect features of subglacial lakes in IPR images. Using this method, we compare the new detections with the known lakes inventory. The results indicate that this new method identified additional subglacial lakes that were not previously detected, and some previously known lakes are found to correspond to other reflector clusters. This method can offer automatic detections of subglacial lakes and provide new insight for subglacial studies.

## 1 Introduction

Subglacial water, i.e., water between bedrock and ice sheet, is formed through a complex interplay of factors such as subglacial pressure, friction heat, geothermal flux, and surface water injection (Robin, 1955; Siegert, 2000; Cuffey and Paterson, 2010; Pattyn, 2010). Subglacial lakes play an important role in subglacial water networks, which can also impact ice flow and dynamics (Kamb, 1987; Stearns et al., 2008; Siegfried et al., 2016; Kazmierczak et al., 2022). Investigation of water storage in subglacial lakes can provide insights into estimating the contribution of ice sheet meltwater to sea level rise (King et al., 2020; Fettweis et al., 2013) and the history of former climate change and ice sheet evolution (Dowdeswell and Siegert, 1999). In addition, subglacial lake sediments may also contain information that records the historical evolution of ice sheets (Smith et al., 2018). The extreme conditions of low temperature and absent sunlight create unique subglacial lacustrine ecosystems (Christner et al., 2014; Mikucki et al., 2016).

Ice-Penetrating Radio detection and ranging (IPR) can be used to detect the subsurface features of ice sheets (Bailey, 1964; Robin et al., 1969, 1970; Carter et al., 2007; Paden et al., 2010; Arnold et al., 2020). The thickness of the subglacial water layer and sediment characteristics at the bottom of lakes are also investigated with active seismic surveys (Paden et al., 2010; Arnold et al., 2020) and gravimetry and electromagnetic methods (Studinger et al., 2004; Key and Siegfried, 2017). These observations have been used to construct the first Global Subglacial Lake Inventory (Livingstone et al., 2022).

Subglacial lakes can be identified in radar images due to their distinct, bright, flat, and specular reflection characteristics (Oswald and Robin, 1973). Because of the specific reflection characteristics of subglacial lakes in IPR images (Schroeder et al., 2013), the manual extraction of the visual features was initially applied (Siegert and Ridley, 1998; Gades et al., 2000; Dowdeswell and Evans, 2004). With the increase in IPR data, semi-automatic methods based on ice-bottom roughness features and reflected signal power have been developed to search for lake candidates (Carter et al., 2007; Bowling et al., 2019). Automatic methods based on experts' experience and physical modeling (Lang et al., 2022; Hao et al., 2023), as well as machine learning methods (Gifford and Agah, 2012; Ilisei et al., 2018) have also been proposed in subglacial lake detection. These methods have shown that improved selection rules and thresholds can enhance detection accuracy and efficiency. However, these methods were based on assumptions of physical modeling or learning from previous detection experience, which may lead to potential inaccurate detections. In the past decades, IPR surveys have collected large amounts of radar images, which enable the analysis of basal radar reflectance features even if the interpretation of basal radar reflectance features is absent.

In recent years, deep learning has been applied as a powerful tool to detect different features in IPR images, including bedrock interfaces (Xu et al., 2017; Rahnemoonfar et al., 2017; Dong et al., 2021; Liu-Schiaffini et al., 2022), internal ice layers (Yari et al., 2020; Varshney et al., 2020; Dong et al., 2021), snow accumulation layers (Varshney et al., 2021) and subglacial waters(Gifford and Agah, 2012; Ilisei and Bruzzone, 2015; Ilisei et al., 2018). Moreover, deep learning applied to IPR has also contributed to estimates of ice thickness (e.g., Tang et al., 2022; Wang et al., 2023).

The Center for Remote Sensing and Integrated Systems (CReSIS, https://data.cresis.ku.edu/#ACRDU) released an extensive collection of historical radar images recorded in the Antarctic and Greenland ice sheet (Arnold et al., 2020). These datasets have driven various investigation of recent subglacial studies (e.g., Varshney et al., 2021; Zeising et al., 2022). By utilizing the precise label in the CReSIS dataset for reflectors from ice bottom, it is now feasible to construct a comprehensive catalog of basal reflector characteristics, facilitating further analysis of reflector features.

In this study, We follow the known lakes inventories (Wolovick et al., 2013; Livingstone et al., 2022) to investigate subglacial lakes in the Gamburtsev Subglacial Mountains region. We select IPR images in the region of the Gamburtsev Subglacial Mountains from CReSIS database. We crop these images around the ice bottom, to obtain a set of one-dimensional waveforms that capture the ice bottom reflectance characteristics. Using this data, we train a Variational Auto-Encoder (VAE, Kingma and Welling, 2013) to reconstruct the one-dimensional waveform features of basal reflectors. We then apply K-means clustering methods (MacQueen, 1967) in the VAE's latent space to analyze similar reflection features and separate them into different clusters. We identify a cluster of reconstructed reflectors with sharp, steep, and symmetric waveform characteristics corresponding to the subglacial lakes observed in field radar images. Furthermore, we apply a conventional method based on the linear relationship between depth and peak reflected power to filter the candidate subglacial lakes from latent space clustering.

By using this workflow, we can obtain an automatic approach in subglacial lakes detection. To validate the results, we compare the distributions of subglacial lakes by this method with the existing inventories. This automated method can improve the efficiency of the detection of subglacial lakes. By collecting and verifying the waveform characteristics of subglacial reflectors, the accuracy of subglacial lakes can also be improved. Additionally, this approach can be extended to detect and label other

clusters of subglacial features, providing valuable reference data for further studies of subglacial environments.

## 2   Data and Methods

In this section, we will introduce the workflow of the ice bottom reflection feature clustering method, as shown in Figure 1, which includes the extraction and sampling of ice bottom reflector features (Figure 1a), the feature reconstruction and latent vector encoding by the variational auto-encoder (Figure 1b), the unsupervised clustering of ice bottom reflector features (Figure

1c), and the implementation of subglacial lake detection (Figure 1d).

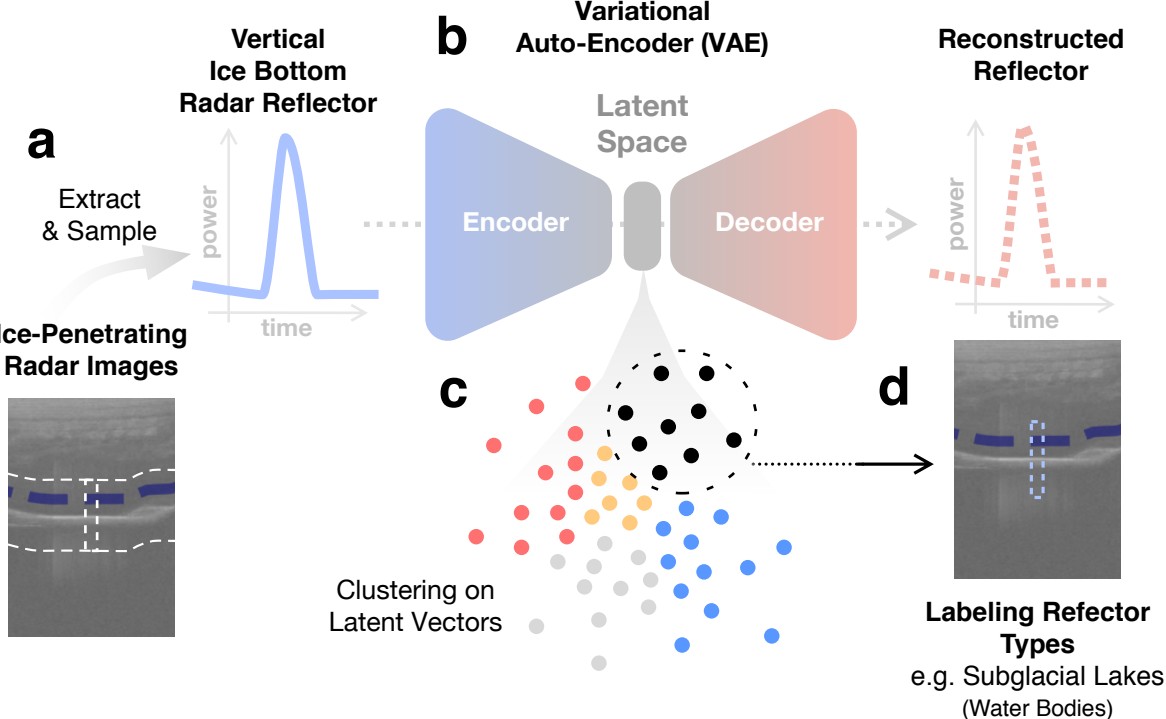

**Figure 1.** Workflow for subglacial lakes detection: (a) Extract and sample the ice bottom reflector trace by trace in IPR images. (b) The VAE encodes and reconstructs the sampled ice bottom reflector. (c) Unsupervised clustering on the encoded latent vectors. (d) Trace the ice bottom reflector corresponding to the subglacial lakes cluster.

## 2.1 Ice bottom reflectors

The utilized airborne radar images were collected during the December 2008-January 2009 Antarctic's Gamburtsev Province Project (AGAP) (https://data.bas.ac.uk/full-record.php?id=GB/NERC/BAS/PDC/01544) from the CReSIS database. According to the lakes inventory (Wolovick et al., 2013; Livingstone et al., 2022), multiple known subglacial lakes are located in this study area. We focus on the dataset from the southern camp of Dome A (AGAP-S), which comprises 2,715 IPR images with a central frequency of 150 Hz, a bandwidth of 10 MHz, and a transmitting power of 800 W (Wolovick et al., 2013). We use the L1B data product (CSAPR_standard), which employs focused synthetic aperture radar processing on each channel and motion compensation during data pre-processing (Arnold et al., 2020). The radar images have an average spatial along-track trace spacing of 14 m and a time sample step of $10^{-7}$ s, equivalent to a sample range of 8.4 m in ice. The radar images also contain the positions of ice bottom reflectors, which were extracted by hybrid manual-automatic method (CReSIS, 2016).

In this study, we perform a series of data processing steps to extract the ice bottom reflector signals from the radar images. First, we transfer the echo power to decibel scale for each radar image by $[X]_{db} = 10 * \log_{10}(X)$, where $X$ is the pixel value from the images. Second, we use the bed picks in the dataset to truncate the 1-D data within $\pm 200$ sampling points near the bed reflector position for every single vertical trace. Third, we align the 1-D trace data by centering the traces according to their maximum value (peak echo) to correct minor position misfits of semi-automatic reflector picking. This step ensures that the maximum value of bottom reflector signal features always resided at the center of the 1-D trace data. To reduce the interference from other englacial radar features such as the internal ice layers and the potential multiple diffractions from bedrock, we apply constant time windows for each trace near the peak signal values. The width of the time window should contain the main part of the signal waveform. We truncate $-64$ to $+63$ data sampling points around the peak signal centers, which maintain a fixed length of 128 for each ice-bottom reflection signal. As the sampling rates of the radar images in this region are identical, the sample ranges of the ice bottom reflector features are also consistent.

To enhance the reflector features and minimize the impact of sampling noise in radar images, a constant Gaussian filter with a kernel sigma value of 4 is applied to all the extracted trace reflector data. Last, all dynamic ranges of reflector features from different radar images are normalized into 0-1 to reduce the influence of background echo power in the radar image and accentuate the reflector features. The normalization in each reflector waveform reduces the complexity of data samples, which also accelerates the following training process and enables the waveform downsampling to a small 2*1 vector. By following the steps above, we generated and collected an ice bottom reflector waveform feature dataset with 1488600 1-D Z-axis radar echo traces.

## 2.2 Variational Auto-Encoder

Variational Auto-Encoder (VAE) was first proposed by Kingma and Welling (2013) and designed for image and signal processing. As an auto-encoder, VAE consists of an encoder and a decoder: the encoder reduces the data sizes and downsamples the input data to vectors in latent space; the following decoder reconstructs the latent vectors to approach and match the raw input data. After training, the encoded latent vectors can be considered dimension-reduced representations of the input data.

VAE now has various applications in Earth science studies, such as geophysical inversion (Cheng and Jiang, 2020; Liu et al., 2022; Lopez-Alvis et al., 2021), shale petroleum prediction (Li and Misra, 2017), engineering seismic analysis(Esfahani et al., 2021) and seismic mechanism analysis (Li, 2022; Ma et al., 2022).

In this study, we employ VAE to reduce the dimension of the reflector waveform features from the ice bottom. The VAE architecture is shown in Figure 2a, which consists of fully connected layers, including an input layer of 128 neurons, an encoder and a decoder consisting of two hidden layers with 128 neurons, and an output layer with 128 neurons. To perform a two-dimensional clustering in the latent vectors, we design the bottleneck (latent space) between the encoder and decoder to a small size of 2 * 1 following Li (2022). The 2-D latent space is also easier presenting the spatial distributions of the latent vectors. The loss function used in VAE training follows Kingma and Welling (2013) and Li (2022):

$$\text{Loss } = \text{MSE} + \text{KL} \tag{1}$$

where the MSE represents mean squared error, which measures the average difference between the predicted and actual values, while the KL represents Kullback–Leibler divergence, which measures the dissimilarity between the probability distribution of the Latent Space and a Gaussian distribution:

$$\text{MSE} = ||X' - X|| \tag{2}$$

$$\text{KL} = -0.5 \cdot \sum_{i=1}^{n} (1 + \log(\sigma_i^2) - \mu_i^2 - \sigma_i^2) \tag{3}$$

where $X$ and $X'$ are raw input reflectors and VAE reconstructed reflectors respectively. The MSE in loss function is applied to calculate reconstruction misfit and $KL$ divergence for estimating and reducing the difference between the distribution and the normal distribution in latent space. $n$ represents the dimension of the latent space $Z$, which was preset to 2 in this study, and $\sigma$ and $\mu$ are the variance and mean of the latent space respectively. The Adam Optimizer (Kingma and Ba, 2014) is employed to accelerate the training process.

We use the randomly shuffled reflector datasets to train and validate VAE. $90\%$ of the data are used for training VAE, while the remaining $10\%$ served as a validation set. The VAE is updated by a full training dataset during different epochs in training. Due to the similar reflector features after single-trace normalizations and large data amount applied in training, the training loss rapidly descends and no longer changes after the first epoch, thus we stopped the training at epoch 4 (Figure S1).

To illustrate the VAE's reconstruction performance, we randomly select different reflectors from the validation set to demonstrate the reconstruction of ice bottom reflector features (Figure 2b, c). Subfigures in Figure 2b show a group of symmetrical reflectors and their corresponding reconstruction. The reconstructed reflector features (orange waveforms) remain the width and trend of raw input reflector features (blue waveforms). Due to the low-dimension bottleneck with 2*1 size applied in the latent space, the high-frequency detailed features in reflectors feature are unattainable and thus discarded by VAE. Figure 2c demonstrated a group of asymmetric reflector features and the corresponding VAE reconstruction. The comparison between inputs and reconstructions suggests that the asymmetric trends of the reflector feature are also successfully reconstructed, as well as the width waveform feature. In general, VAE can reconstruct the features of both symmetric and asymmetric ice bottom

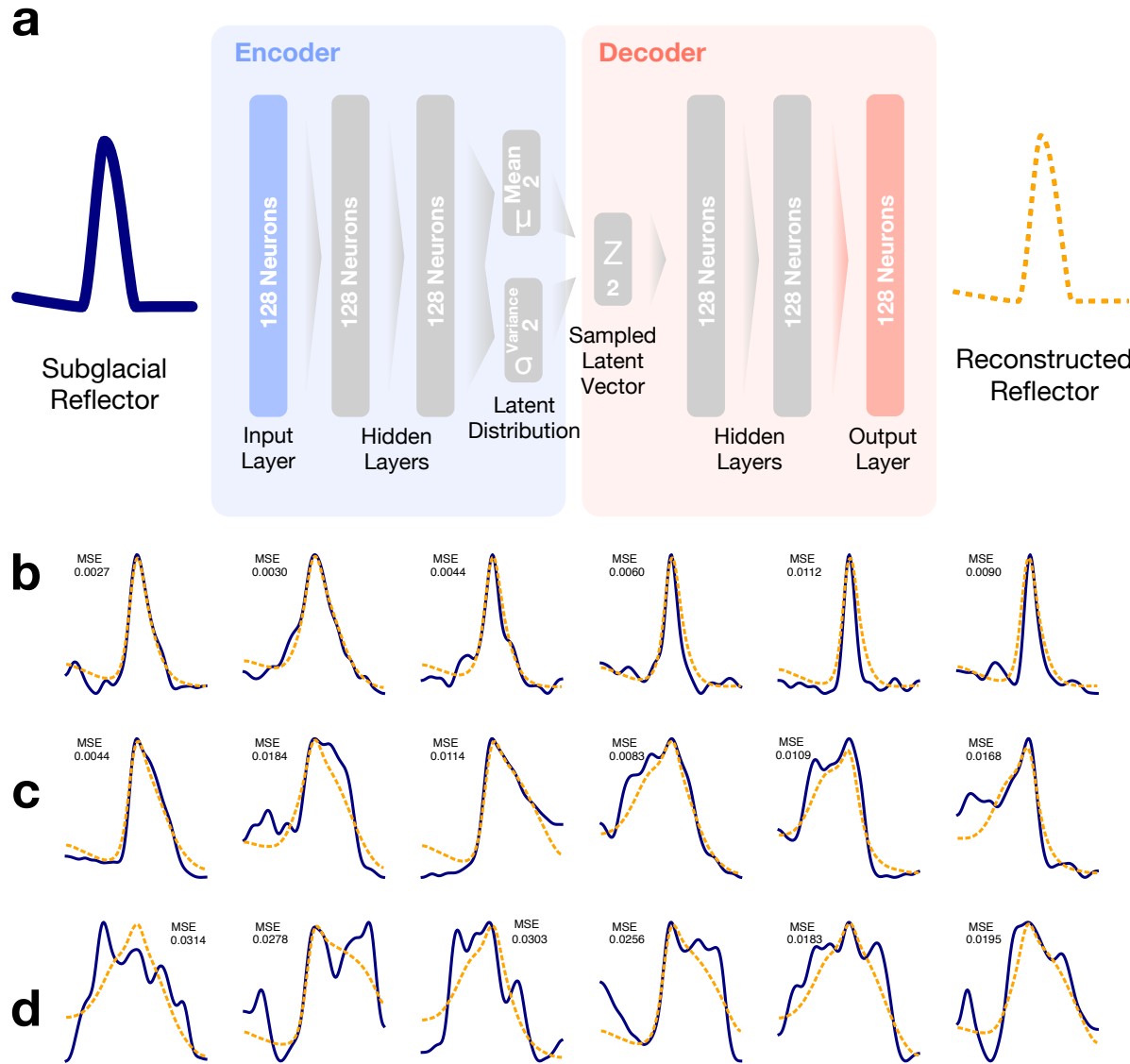

**Figure 2.** Variational Auto-Encoder (VAE) and demonstrations of ice bottom reflector reconstruction. (a) VAE architecture, with both encoder and decoder consisting of two fully connected layers with 128 neurons, and bottleneck $1 \times 2$ latent space. (b-d) Illustration of data reconstruction using VAE: input raw reflectors (blue waveforms) and VAE reconstructed reflectors (orange waveforms), where the horizontal axis corresponds to time and the vertical axis corresponds to the normalized reflection power (ranging from 0 to 1). Reconstruct MSEs are labeled above the waveforms. (b) Symmetrical reflector features. (c) Asymmetrical reflector features. (d) complex reflector features, which result in higher reconstruction errors.

reflectors. Furthermore, we select typical reflectors with large reconstruction errors to demonstrate the large misfit conditions (Figure 2d). Notably, reflectors contained with high-frequency signals, multiple peaks, and severe oscillations are challenging to reconstruct, thus resulting in higher errors. These peculiar signal features deviate significantly from the majority of reflector features in the training set, rendering the features difficult to encode and decode through latent vectors. The reconstructions of multiple peak features are usually simplified to broader reflection shapes, whose trends are approximate to the smooth shape and average of the input features.

As shown in Figure 2a, the original 128-length reflector waveform features are transformed into a 2-length latent vector between the encoder and decoder of the VAE. The features of ice bottom reflectors are derived by the encoder part of the VAE to latent vectors consisting of two dimensionless scalars, $Z_1$ and $Z_2$, which can be regarded as vectors containing the original signal features. Therefore, the distance between vectors from two reflector samples in the latent space can be considered as an indicator of statistical feature similarity.

## 2.3   Clustering Analysis in Latent Space

After VAE training, we randomly select a subset with 2000 reflector samples from the intact dataset for clustering. Because the selection is uniformly random, the reflector samples are from different radar images captured in different regions, and thus the samples reveal different ice bottom conditions. The reflector samples are first encoded by VAE's encoder to 2-D vectors in latent space. Figure 3a shows the vector distributions of the samples in the latent space, in which each scattered point corresponds to an encoded reflector sample. Due to the application of $KL$ divergence in VAE's loss function, the vector distribution of these samples in the latent space composed of $Z_1$ and $Z_2$ is approximate to a 2-D Gaussian distribution. According to the character of VAE (Kingma and Welling, 2013), the distance between encoded vectors in the latent space is equivalent to the difference between the input reflector samples. By measuring the distances between the reflectors' latent vectors, we can estimate the difference in waveforms' features. Furthermore, the distance-based clustering in latent vectors can classify the ice bottom reflector feature with similar features.

To improve the clustering efficiency, it is advisable to reduce the amount of data used. However, in order to ensure the accuracy of clustering, the selected samples should have sufficient data density and match the same distribution as the experimental data. As illustrated in Figure 3a, using latent vectors from a randomly selected set of 2000 reflectors is sufficient for clustering. Therefore, we apply these samples for the next clustering analysis. We employ the K-means clustering algorithm (MacQueen, 1967), which based on the Euclidean distance estimation of the differences between data samples, as well as the characteristics in VAE's latent space. Initially, $K$ clustering centers are randomly assigned in 2-D space. The distance of each sample vector to the cluster center is computed, and the sample is assigned to the nearest cluster with the smallest distance. Then, all the cluster centers are updated to the spatial center of all the samples belonging to the corresponding cluster. This assign-update process is repeated until the cluster center becomes constant or the clustering result remains unchanged.

The number of clusters ($K$) is a preset parameter in the K-means algorithm, which must balance the tradeoff between implied feature classes and feature density in the data. On the one hand, $K$ should be sufficiently large to distinguish between different ice bottom conditions. On the other hand, $K$ should not be so large as to create unnecessary subclasses. To obtain optimal

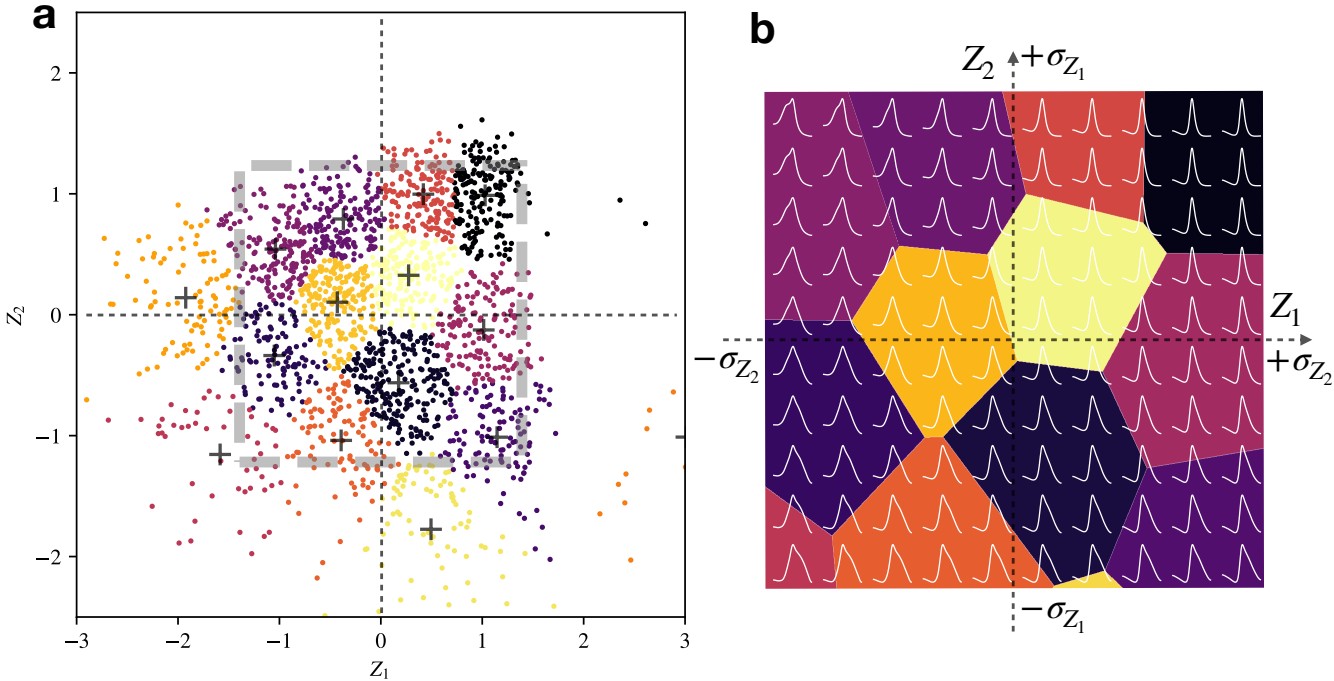

**Figure 3.** (a) Latent space distribution of 2000 randomly selected encoded reflector features, with each point representing an encoded reflector sample. The color of each point represents different clustering results (classes), and the black cross denotes the clustering center of each class. The gray dashed rectangle indicates the range of 2 standard deviations ($\pm\sigma_{Z_1}$ and $\pm\sigma_{Z_2}$) of the latent vectors. (b) Synthetic ice bottom reflectors reconstructed by virtual vectors, where the virtual vectors' range corresponds to the ranges of standard deviations ($\sigma_{Z_1}$ and $\sigma_{Z_2}$). Different colors divide the latent spaces corresponding to different clusters, in which the waveforms demonstrate the synthetic reflectors in different clusters. The Candidate cluster corresponding to subglacial lakes is shown in black color near the upper right corner.

clustering results, we first applied the elbow method to determine the appropriate value of $K$ (Figure S2). However, the elbow curve does not show a clear cutoff point, possibly due to the distribution of vectors in the latent space (Figure 3a) not displaying a distinct trend of multiple classes. Therefore, we tested various alternative values of $K$, and ultimately selected $K = 15$. The clustering in latent vectors separates the ice bottom reflector features with similar waveform features, as demonstrated by the
different colors of points in Figure 3a.

To visualize and trace the representative waveform features in different clusters as well as the different regions in latent space, we apply a set of virtual vectors to generate synthetic waveforms by VAE's decoder. The virtual vector set is generated by a grid with the same step length in the latent space, then the decoder generates the waveforms corresponding to the inputted vectors. The 2-D range of the virtual vector grid is assigned based on the standard deviation ($\sigma$) of $Z_1$ and $Z_2$, as shown
in the gray dashed rectangle in Figure 3a. The ranges in $Z_1$ and $Z_2$ are both divided into 10 intervals each. The synthetic

waveforms are shown in Figure 3b, as well as the corresponding area of clusters. The VAE's learning target involves waveform reconstruction. Consequently, we can equate the synthetic waveforms with the input reflector waveforms that are encoded as identical vectors in the latent space. These synthetic waveforms can serve as a direct reference for the initial cluster selection of input subglacial water reflections using conventional waveform methods, such as Hao et al. (2023).

## 2.4 Subglacial Lake Detection

We further investigate the geometry features of synthetic waveforms in different clusters. We initially identify one of the clusters corresponding to subglacial lakes (indicated by black clusters and the region in the upper right corner in Figure 3b). The waveforms within this cluster display symmetrical shapes and rapid signal attenuation near the waveform peak, similar to subglacial lake reflections previously identified in studies such as Schroeder et al. (2013); Hills et al. (2020); Hao et al. (2023). Subsequently, we map the distribution of these reflectors in radar images. The results show that these reflectors are continuously distributed in radar images, and the reflectors generally display flat, bright characteristics (Figure 4c). These continuous features are similar to the visual criteria used by glaciologists to identify subglacial lakes (Wolovick et al., 2013; Schroeder et al., 2013). Therefore, we further apply the results of the encoder-clustering as a candidate distribution of subglacial lakes.

In further applications of observational data, it has been observed that the signal-to-noise ratio of radar images from deep ice sheets is low due to the attenuation of radar signals. The interference of noise can occasionally cause odd clusters in the detection of candidate subglacial lakes (e.g., Figure 4c). Occasionally, subglacial lakes may be mistakenly identified as appearing in non-lake areas. Additionally, the complex conditions of the ice bottom can also cause interruptions in subglacial lake detection. To eliminate noise interference and extract continuous subglacial lakes or lakes, we limit the minimum width of subglacial lakes in observational detection. Detected subglacial lakes should contain a continuous ice bottom segmentation in subglacial water type with a width greater than 8 traces (corresponding to an average spatial distance of 112 m). Meanwhile, interruptions in continuous subglacial lakes, which are narrower than 8 traces, are considered noise interference and will be interpolated and filled into nearby subglacial lakes. During interpolation, it is ensured that the interpolated non-subglacial lakes in the continuous subglacial lakes are less than $25\%$ to avoid mistaken detection caused by abundant interpolation.

By implementing a threshold on the minimum width of subglacial lakes, we obtain a list of candidate lakes with larger widths, effectively minimizing noise interference. However, some of these candidates still exhibit weak and indistinct bottom reflector features that could not be conclusively identified as subglacial lakes. Therefore, we follow the conventional subglacial lake detection method based on englacial signal attenuation of bed reflectors (Wolovick et al., 2013; Hills et al., 2020). In this study, we apply a simplified process, using a linear threshold based on the average peak reflector echo power in different depths to reduce the reflector power anomalies. Values of peak echo power and depth are directly extracted from radar images for each reflector without ice surface correction, simplifying the approach. We calculate the best linear fit and standard deviation on the 2-D distribution of ice thickness (depth) and peak echo power of bottom reflectors from the radar images (Figure 5). Considering the reflector features analyzed during the VAE and clustering steps, and with uncorrected ice thickness applied, a lower linear threshold in average echo power (best fit $+1\sigma$, compared to the previous study (Wolovick et al., 2013)) is applied

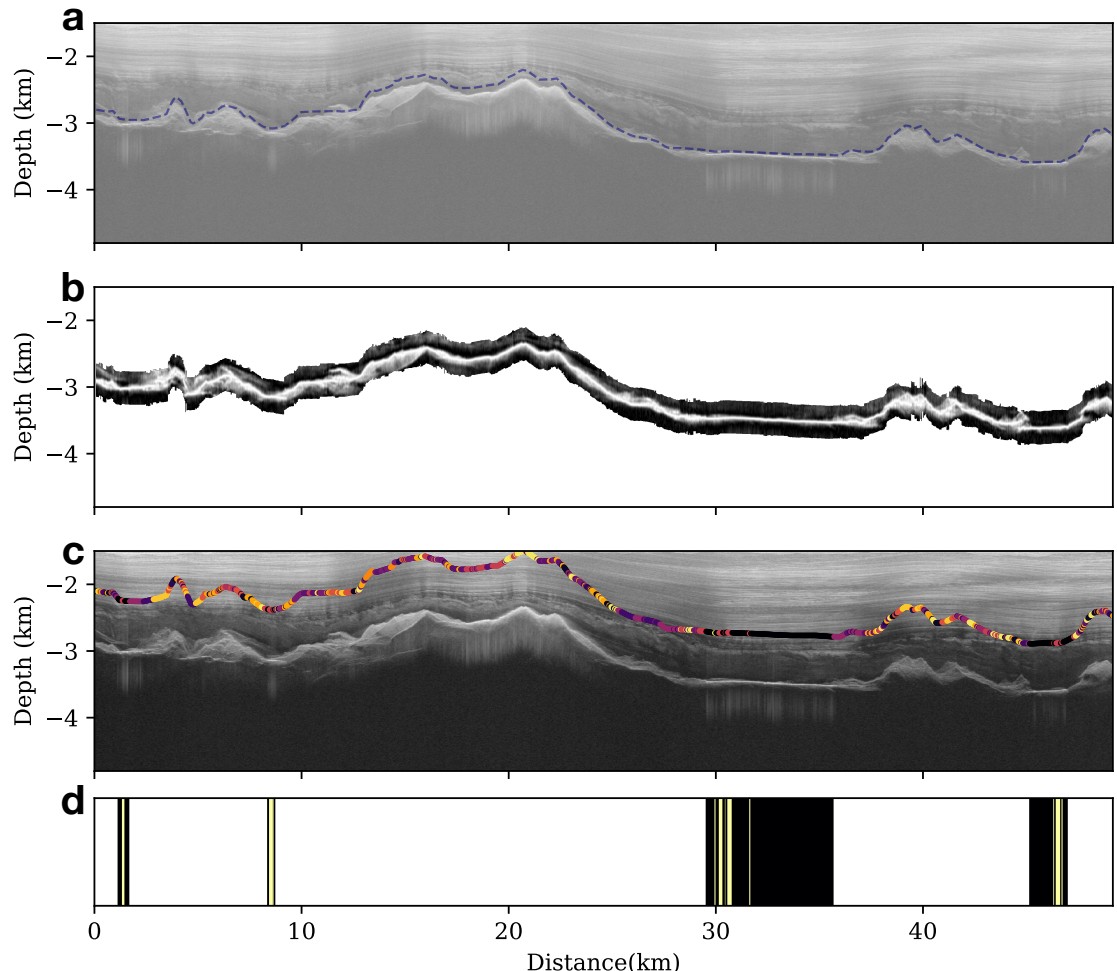

**Figure 4.** First example of subglacial lake detection includes two larger and two smaller subglacial lakes. (a) IPR image is shown with the blue dashed line indicating the positions of the ice bottom reflectors. (b) Separated, realigned, smoothed, denoised, and normalized ice bottom reflector features, which are applied as inputs to the encoder. (c) Results of the unsupervised clustering of the latent vectors obtained through encoding, where different colors correspond to classes in Figure 3. The black cluster corresponds to the candidate subglacial lakes. (d) The subglacial lakes detection based on continuous reflector features, where black blocks represent the raw detected subglacial lakes, the yellow blocks are occasional interruptions that are filled by interpolation, and white blocks correspond to the non-lake clusters.

to preserve potential subglacial lakes. The confirmed reflectors are represented as black points in Figure 5. The average echo power of the detected subglacial lakes in the filtered list should surpass the threshold at the corresponding average depth. Consequently, this final refinement excluded candidate lakes exhibiting weak and blurred reflector features.

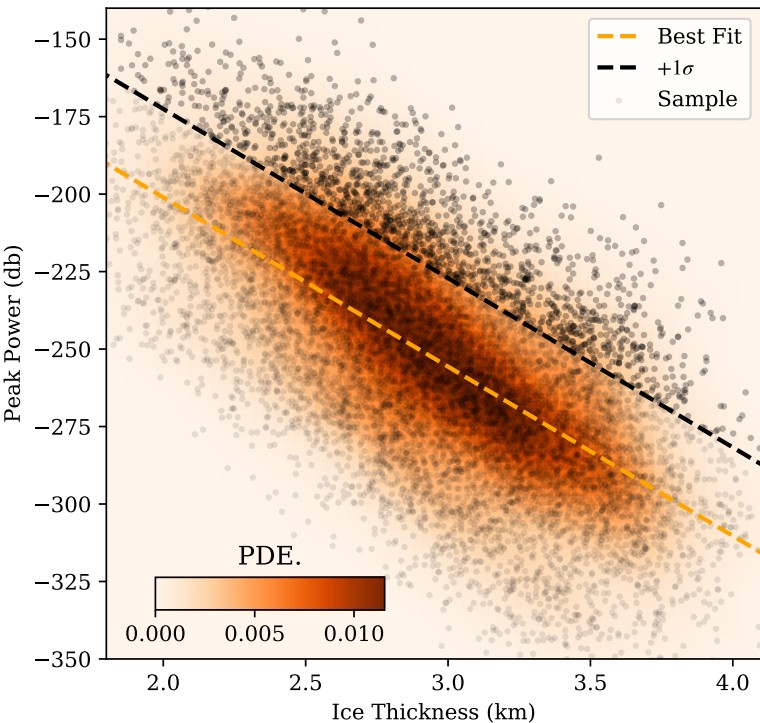

**Figure 5.** Distribution of ice bottom reflection peak power and ice thickness. The background colormap represents the probability density estimation (PDE.) of the data; orange dashed line represents the best linear fit; black dashed line denote the $+1\sigma$ cutoff threshold. Gray dots represent reflector samples, while black dots represent the detected samples for subglacial lakes.

## 3 Results

We apply the encode-cluster method to the IPR images in the AGAP-S dataset and trace the spatial distribution of subglacial lakes in the images. In this section, we first demonstrate subglacial lakes detected at different scales. Next, we compare the distribution of the detected subglacial lakes with known lake inventories, and discuss the newly detected subglacial lakes, as well as the known subglacial lakes missed by our method.

### 3.1 Subglacial Lakes in Different Scales

Figure 4 shows two large subglacial lake distributions and two smaller subglacial lakes located at the bottom of subglacial valleys. The two larger lakes on the right display high echo power as well as continuous and flat reflection features, which are relatively easy to detect visually. In contrast, the two smaller subglacial lakes on the left are easily overlooked due to

their relatively narrow widths containing insufficient continuous and flat reflection features. This example demonstrates the detection of two different types of subglacial lakes of varying widths from within a radar image. The geothermal and subglacial environments should be similar in the same radar image, which was continuously recorded in adjacent areas. Therefore, the detection of the two smaller subglacial lakes can be considered reliable based on the reflector encodes and the following clustering results. In addition to the examples shown in Figure 4, we provide further examples of subglacial lakes detection in Figures 6 and 7, where the workflows and sequences applied are identical to those shown in Figure 4.

Figure 6 shows the detection of a relatively small subglacial lake, which is located at the concave bottom of a subglacial valley. Despite its short length, this lake displays a flat and continuous reflection interface, with strong echo power and rapid attenuation characteristics, making it visually similar to the larger lakes in Figure 4 and Figure 7. The continuous reflection features of this type of smaller subglacial lakes are narrower and less prominent, which makes them easily overlooked in visual detection in previous studies.

Figure 7 presents a special example of a large continuous subglacial lake (at about 40 km along the transect) shown in a radar image. This subglacial lake has high returned power and flat reflection features that are visually easy to be detected. However, only part of the lake is detected by the encode-cluster method based on reflector features, and discontinuities are found within the lake (Figure 7d). Upon inspecting the radar image (Figure 7a), we observe that the left part of this subglacial lake (indicated by the white arrow in Figure 7c) displays different reflector features from the detected part of the lake. These inconsistent features visually have relatively thick and uniform reflection layer-liked features near the ice bottom interface, resembling frozen-on ice as described by Bell et al. (2011). Additionally, another discontinuity interrupts the detected subglacial lake distribution in the center, which also implies thick reflection layer features. Moreover, in other areas of the radar image, there are also other continuous clusters of subglacial reflector features (as the yellow arrow indicates in Figure 7c). By tracing these clusters, we note that these different classes of reflectors correspond to distinct uniform reflection layers with varying thicknesses. Due to the similar features of ice bottom reflections, we suggest that these continuous spatial distributions may relate to the ice flow dynamics and different stages of frozen-on ice (Bell et al., 2011).

## 3.2 Spatial Distribution of Detected Subglacial Lakes

We compile and integrate the identified subglacial lakes and lakes from the AGAP-S IPR images, and locate each detection within the spatial sampling range of each radar image provided by the database. Figure 8 presents the spatial distribution of subglacial lakes detected in the Gamburtsev Subglacial Mountains region, where the blue points represent subglacial lakes that have been confirmed by applying the peak reflection power filter to subglacial lake candidates detected by the encode-cluster method (light cyan points in Figure 8). Overall, the subglacial lakes are distributed in clusters with spatial continuity (e.g., the regional cluster near L1 and L3 area), but some isolated lakes are also detected, such as the L2 survey line. Densely distributed subglacial lakes in specific regions are usually identified in radar images by their obvious reflections, as illustrated in Figure 4 and 6. In these cases, most detections are validated through peak power filtering. However, certain candidates in densely distributed regions exhibit lower peak reflector echo power than the established thresholds. This discrepancy is primarily

attributed to the ambiguous and weak reflections, often associated with long-distance flat ice bottom shapes. An example of such candidates can be observed in the densely distributed light cyan points near the lower-left corner in Figure 8.

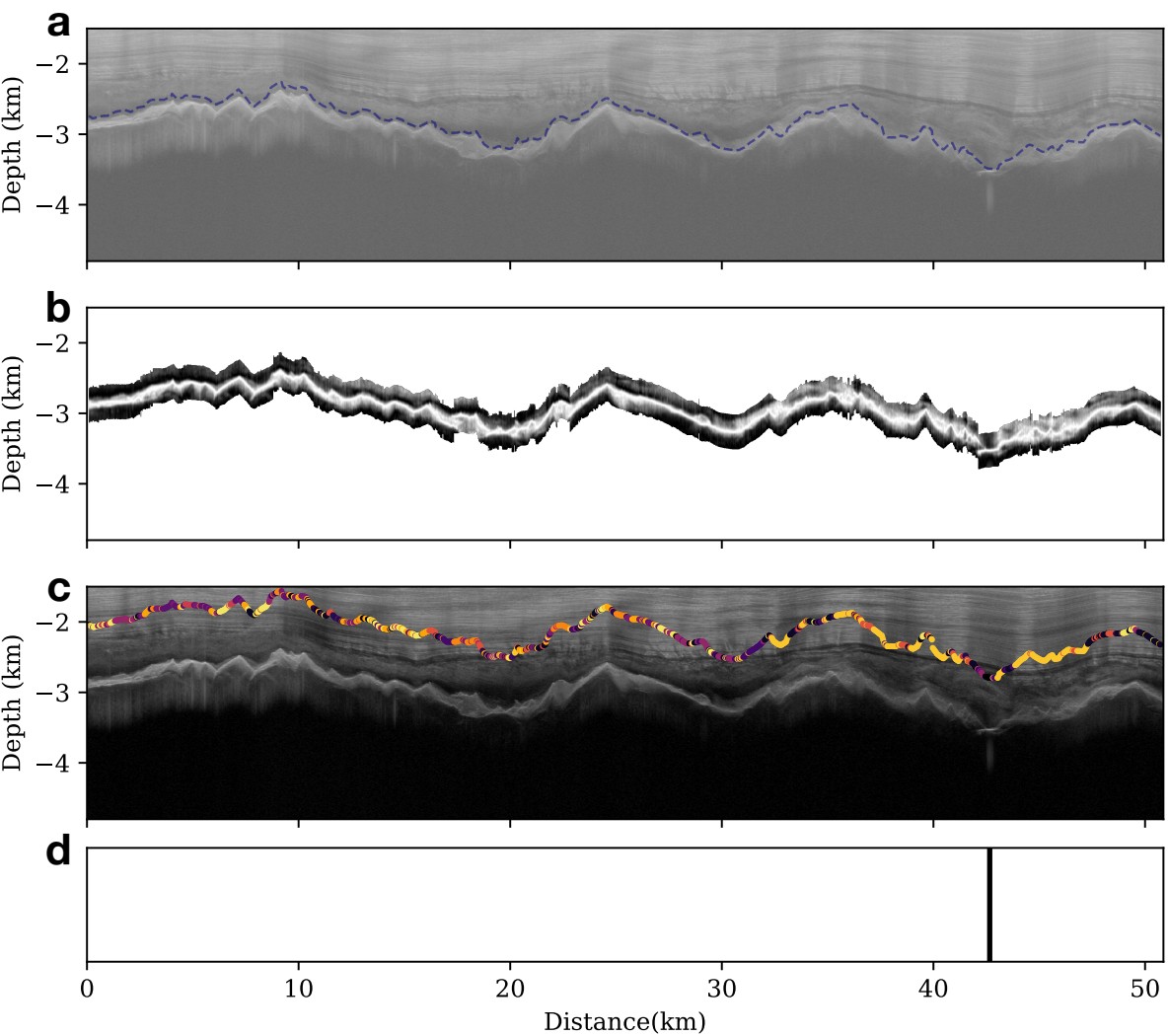

**Figure 6.** Second example of subglacial lake detection, which contains a relatively narrow subglacial lake. (a) Input radar image, where blue dashed line indicating the positions of the ice bottom reflectors. (b) Separated, realigned, smoothed, denoised, and normalized ice bottom reflector features, which are used as inputs to the encoder. (c) Results of the unsupervised clustering of the latent vectors obtained through encoding, where different colors correspond to classes in Figure 3. The black cluster corresponds to the candidate subglacial lakes.(d) The subglacial lakes detection based on continuous reflector features, where black blocks represent the raw detected subglacial lakes, the yellow blocks are occasional interruptions that are filled by interpolation, and white blocks correspond to the non-lake clusters.

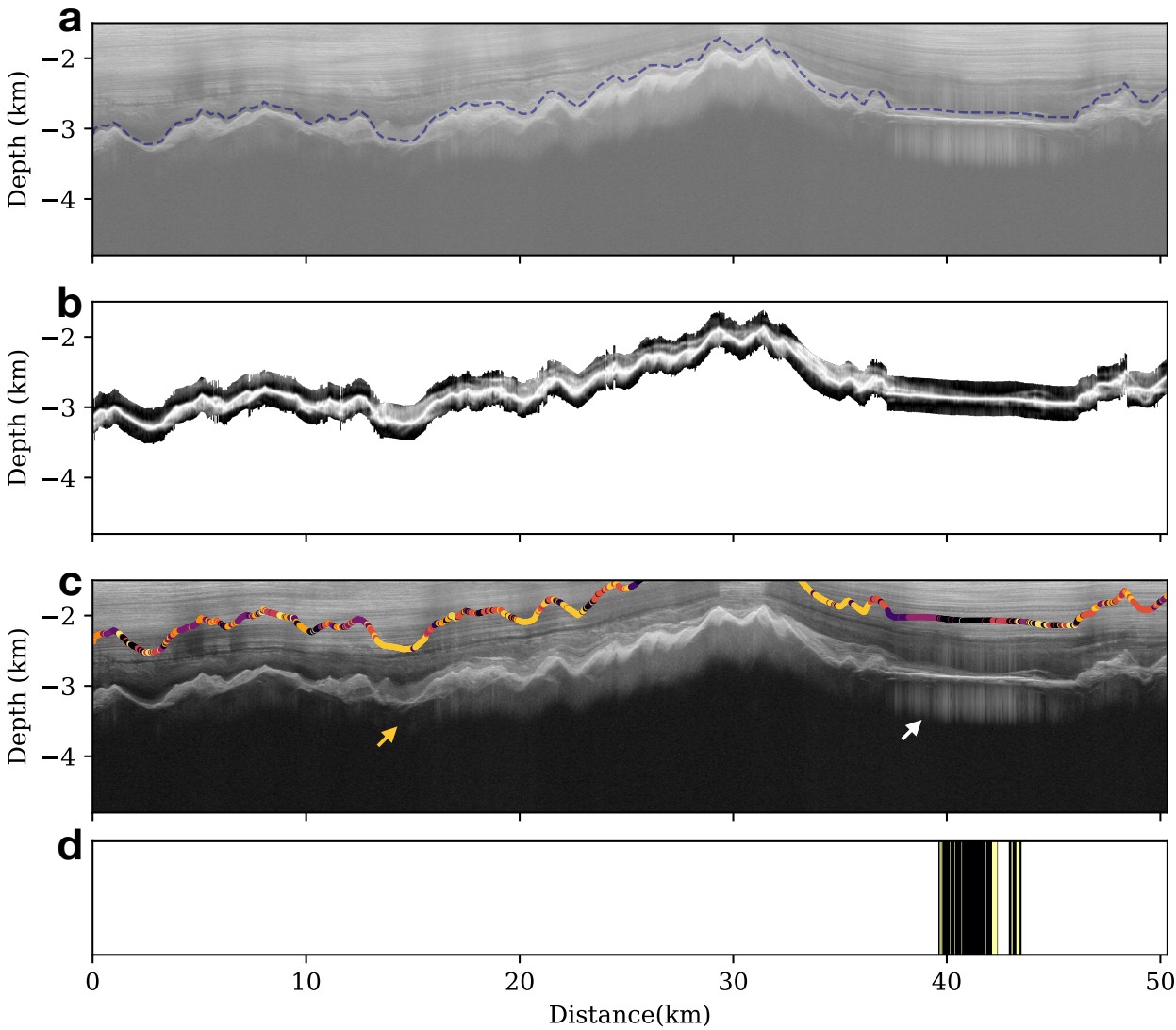

**Figure 7.** Third example of subglacial lake detection, which contains a subglacial lake. (a) Input radar image, where blue dashed line indicating the positions of the ice bottom reflectors. (b) Separated, realigned, smoothed, denoised, and normalized ice bottom reflector features, which are used as inputs to the encoder. (c) Results of the unsupervised clustering of the latent vectors obtained through encoding, where different colors correspond to classes in Figure 3. The black cluster corresponds to the candidate subglacial lakes. (d) The subglacial lakes detection based on continuous reflector features, where black blocks represent the raw detected subglacial lakes, the yellow blocks are occasional interruptions that are filled by interpolation, and white blocks correspond to the non-lake clusters. Yellow arrow indicates another continuous subglacial reflector class distribution, which may correspond to other symbolic subglacial conditions. White arrow indicates possible subglacial frozen-on ice condition.

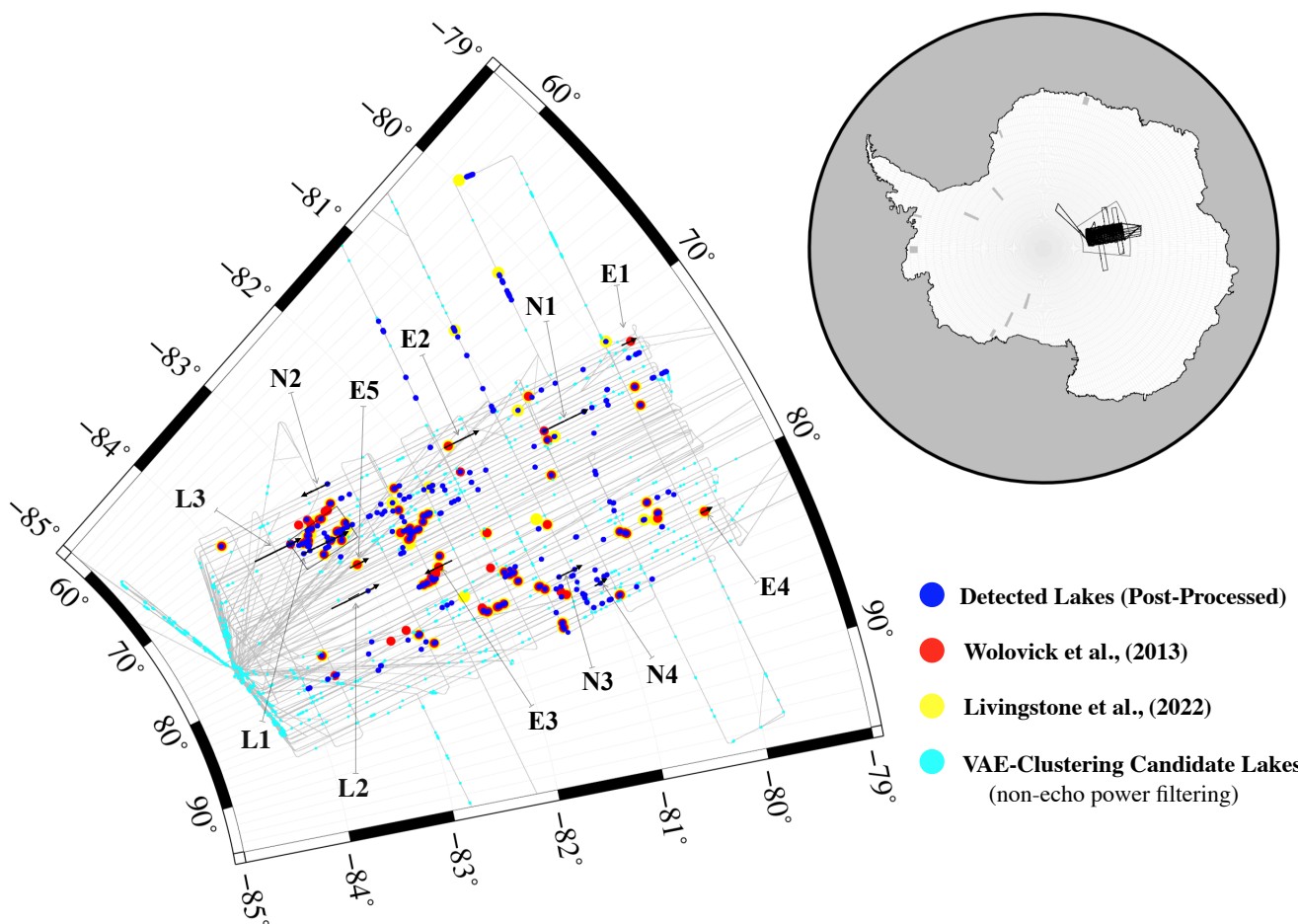

**Figure 8.** Distribution of airborne radar observation lines and detected subglacial lakes in the Gamburtsev Subglacial Mountains region. Blue points indicate the distribution of subglacial lakes detected in this study. Light cyan points mark all the lake candidates from the VAE-cluster method before echo power fitlering. Red and yellow points mark the subglacial lake distribution from the subglacial lake inventory from Wolovick et al. (2013) and Livingstone et al. (2022), respectively. Text labels with gray arrows indicate the position and directions of selected survey lines (shown in black arrows), where L1, L2, and L3 correspond to the detection example survey lines (Figure 4, 6 and 7 ), N1-4 corresponds to the newly detected lakes in Figure 9, E1-5 corresponds to the mismatch lakes with the known inventory in Figure 10. The inset map shows the location of the study area.

In addition to the subglacial lakes detected in this study, we compare the subglacial lakes detected in this study to the previously identified subglacial lake distributions, as shown by the red and yellow points in Figure 8, which correspond to the inventories of Wolovick et al. (2013) and Livingstone et al. (2022), respectively. The two larger subglacial lakes (shown

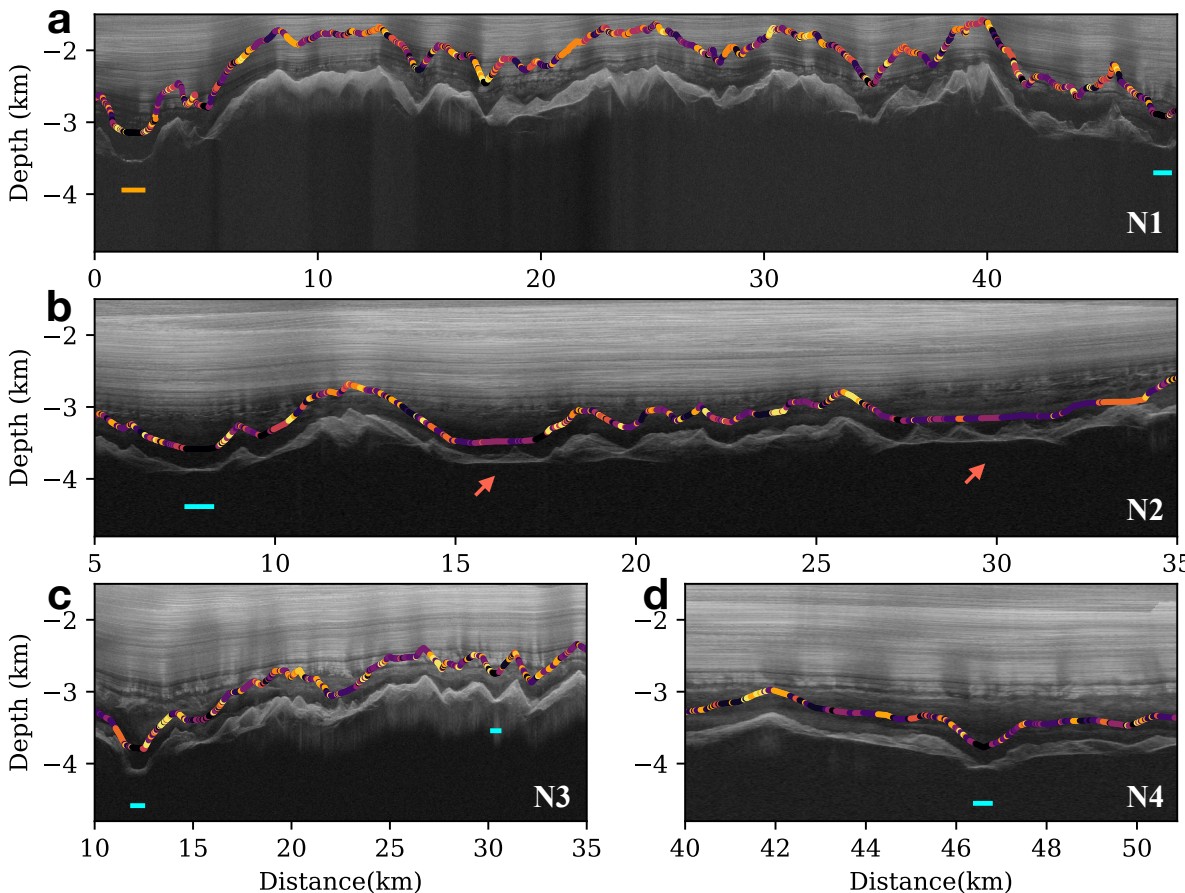

**Figure 9.** Newly detected subglacial lakes and lakes in the Gamburtsev Subglacial Mountains region, with text labels corresponding to the location distribution in Figure 8. Scatter points in different colors mark the encoded classes of the ice bottom reflectors, and the blue and orange lines indicate the detected range of subglacial lakes or lakes. (a) IPR image of the N1 region, containing a known subglacial lake (orange line on the left) and a newly detected subglacial lake (right). (b) Radar image segment of the N2 region, containing a newly labeled subglacial lake area; the two relatively flat ice bottom reflection segments indicated by red arrows may record the frozen-on ice. (c-d) Newly detected regional subglacial lakes with smaller sizes, from radar image segments from the N3 and N4 regions, respectively.

in Figure 4 and 7) correspond to the known subglacial lakes listed in the inventories (labeled as L1 and L3 in Figure 8). In contrast, the narrow subglacial lake shown in Figure 6 is not previously included in the inventories (labeled as L2 in Figure 8). Overall, the subglacial lake distribution detected in this study roughly overlaps with the known inventory, but there are also some mismatches, such as the lines labeled E1-E5 and N1-N4 in Figure 8. To further investigate the reasons for these discrepancies, we select the corresponding radar image segments from labeled regions and plot the segments in Figure 9 and 10, respectively.

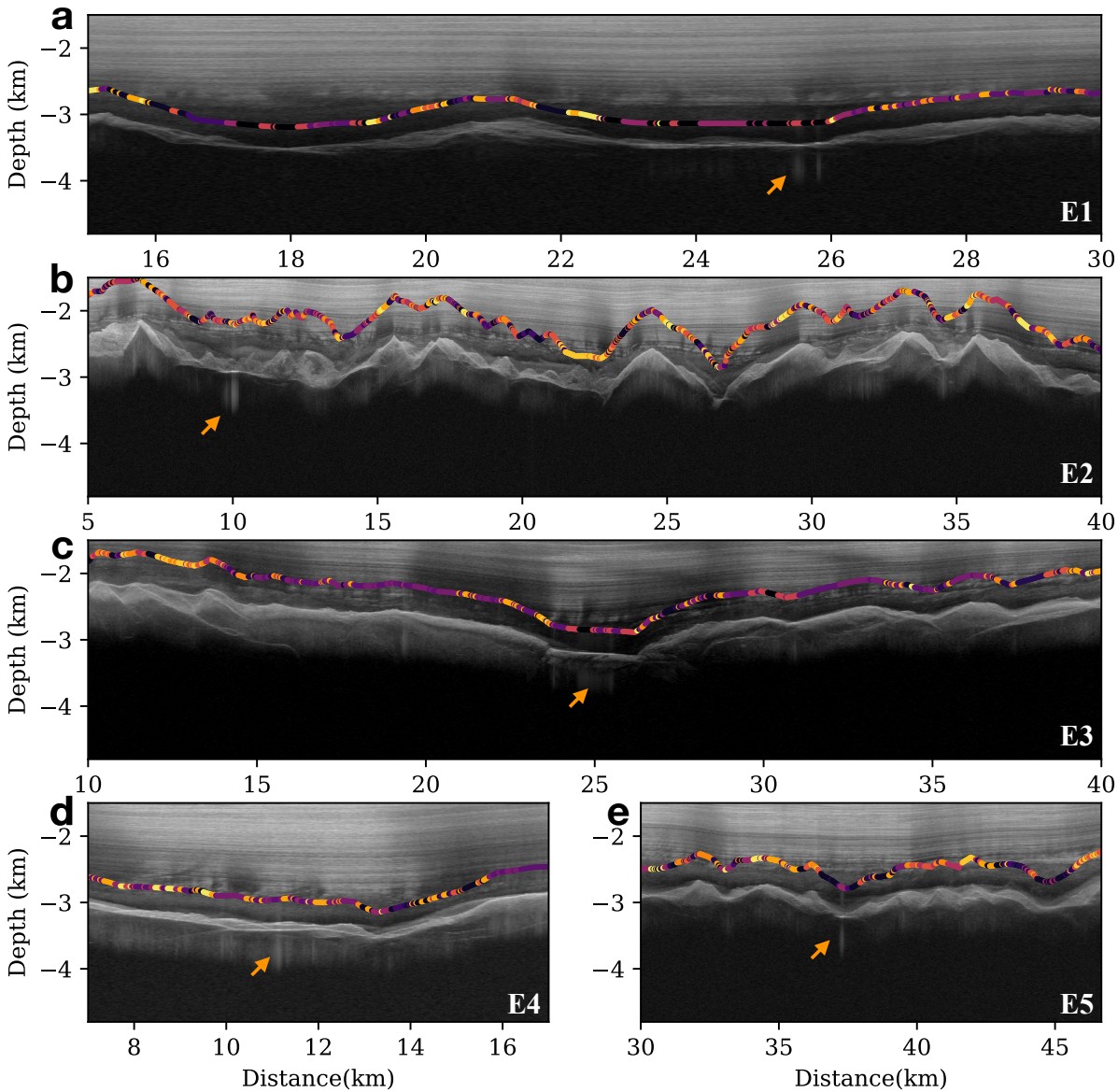

**Figure 10.** IPR image segments from the Gamburtsev Subglacial Mountains area, which mismatch with the identified lake inventory (Wolovick et al., 2013; Livingstone et al., 2022). The text labels correspond to the locations marked in Figure 8, and orange arrows mark the locations of the identified subglacial lakes from the inventories.

Figure 9 displays segments of radar images from the N1-N4 subregions, revealing multiple new subglacial lakes detected by the new method (indicated as blue lines in Figure 9). Figure 9a illustrates two detected subglacial lakes, one on the left (marked as orange line below) that was already included in previous subglacial lake inventories, and one on the right that is

newly detected by the new method. The ice bottom reflectors of both subglacial lakes have a similar visual appearance, with
sharp and narrow reflectors in the Z-axis. However, the lake on the right has a narrower width, which could make it easier to
overlook visually, potentially causing it to be neglected in previous studies.

The radar segment displayed in Figure 9b is from the N2 subregion near -83°S, 70°E in Figure 8, where a group of continuous
subglacial lakes has been detected and recorded in the known inventory (Figure 8N2). However, there is no previous detection
in this radar image from the known inventory. The new method detects subglacial lakes in about 7 km in Figure 9b. It is worth
noting that multiple reflectors with thick layer features (marked by red arrows) display in the 16 and 29 km simultaneously.
Considering the dense distribution of subglacial lakes nearby, these thicker reflection features are possibly formed by frozen-on
ice that complicates the shape of the near-basal reflection trace. Figures 9c and d show several smaller subglacial lakes, which
are similar to the narrow subglacial lake shown in example 2 in Figure 6. These small lakes may have originated from local
melting or subglacial rivers, corresponding to the regionally dense distribution of subglacial lakes near the L2, N3 and N4
regions in Figure 8.

Figure 10 presents subglacial lakes previously identified in the inventories, but which are not accurately detected by the
encode-cluster method and echo power filtering. The orange arrows in Figure 10 indicate the locations of previously identified
subglacial lakes. In Figure 10a, although there are multiple candidate lakes in the E1 subregion from the encode-clustering in
the radar segment, the average peak power is insufficient to confirm the subglacial lake in each lake candidate. The ice bottom
reflectors of this radar segment are visually different from other known subglacial lake features (e.g., Figures 4, 7 and 9).
In Figure 10b, the ice bottom reflectors near known subglacial lakes are classified as corresponding to other reflector classes
instead of lake reflectors. By inspecting the radar image, these reflectors display a thick layer near the ice bottom reflections,
which are similar to reflections in Figure 7c. We hypothesize that the subglacial lake in this segment may be associated with
a frozen-on ice condition, distinguishing it from the thick bottom reflectors observed in Figure 9b. In the latent space (Figure
3b), the clusters in 9b (purple) and Figure 7c (yellow) are adjacent. This adjacency suggests the presence of multiple clusters
potentially corresponding to distinct phases of frozen-on ice.

Similarly, the reflectors from previously identified subglacial lakes in the E4 and E5 subregions in Figures 10d and e are
also classified as other ice bottom reflection classes by the encode-cluster method. By observing the radar image segments
corresponding to these two subregions, the ice bottom reflectors that corresponded to previously identified subglacial lakes
also display thicker layer reflections, which do not match other subglacial lake features from other regions. The undetected
subglacial lake in Figure 10c, near 25 km, is similar to Figure 10a, where the average echo power is insufficient to confirm the
subglacial lakes. Besides, the encoded classes of ice bottom reflectors near the white arrow change to other classes, indicating
that the potential lakes here may consist of more complex bottom conditions.

Overall, the new method in this study can capture more candidate subglacial lakes with similar reflector features to the
previous lake inventories. Compared to the previously identified lake inventories, most of the newly detected subglacial lakes
or lakes in this study are smaller subglacial lakes, which are more possibly overlooked in manual visual inspections and easily
submerged by multi-trace averaging in detection windows. This automated method can promote updating the known lake in-
ventory with further investigation. Besides, the reflector waveform analysis can provide additional candidate clusters of similar

subglacial conditions. Further investigations, including drilling or modeling, are essential to elucidate the connection between reflection waveforms and distinct basal conditions. This exploration may potentially interpret the miss-detecting subglacial lakes (e.g., Figure 10) in known inventories.

## 4 Discussion

The subglacial analysis method proposed in this study is based on the shape of the ice bottom reflector features, which enables the full exploitation of ice bottom echo waveform information contained in the IPR observation data, providing a novel observational perspective for the study of the ice bottom beyond reflection power intensity and roughness. By contrast with conventional supervised learning methods, this study acquires no manual labels, thereby minimizing the potential for artificial misfit from training labels, and allowing for the application of this approach in surveying other potential subglacial conditions.

In the latent space clustering (Figure 3a), since there is no distinct inflection point in the elbow curves(Figure S2), it is necessary to specify the value of K for clustering the reflector samples. The selection of the K value in K-means directly impacts the area of each cluster in the latent space. A smaller K corresponds to larger clusters in latent space, while a larger K allows for more precise isolation of different reflection types. Consequently, different K values directly impact the detection of subglacial lakes. To identify an appropriate K value, we conducted multiple experiments with different K values (Figure S3 and S4) and ultimately selected K=15 in the final detection. Figure 11 illustrates the detected subglacial lake distributions under different K values. Figure 11a exhibits the boundaries between lake and non-lake clusters in the latent space. The black points in Figure 11a represent vectors from detected lakes in this study.Notably, some points fall outside the boundary of K=15 (white dashed curve) due to complementation through interpolations. When considering K values between 14-16 (as indicated by red, blue, and white dashed curves), the clustered areas corresponding to lakes exhibit relatively stable and lower misfit to the vectors from the final lake list. Figure 11b showcases the subglacial lake detections in a regional area from AGAP-S (from the black box in Figure 8). Compared with the known lake inventory (gray points), smaller K values lead to more erroneous detections (e.g., sparse yellow points when K=8), while larger K values might miss more lakes. When K=15, the detected lake distribution aligns well with the known lake inventory. Figures 11c and 11e demonstrate subglacial lake detections with different K values in same radar images. Similar to the spatial distribution in Figure 11b, smaller K values could result in false detections, such as the mistaken detection indicated by the yellow line near 30 km in Figure 11e (when K=8). Conversely, larger K values limit the detection range of subglacial lakes and introduce unexpected discontinuities. Overall, the subglacial lake range detected with K=15 correlates with visual observations. We further expanded the dataset by incorporating more data (5% of the waveforms from the dataset) into clustering and then traced the subglacial lake detections. The results (black lines in Figure 11c, e) showed negligible differences compared to the detections from the smaller dataset clustering (white lines).

The unsupervised clustering in the latent vectors relies on the feature difference in reflection waveforms, allowing analysis of reflectors without precise interpretations of basal radar reflectances and reducing dependence on model assumptions. However, subjective elements persist, such as the experiential selection of the K value and the lattice-liked boundaries observed in Figure 11. Within the latent space, the difference in reflector features can be measured based on the distance of corresponding vectors

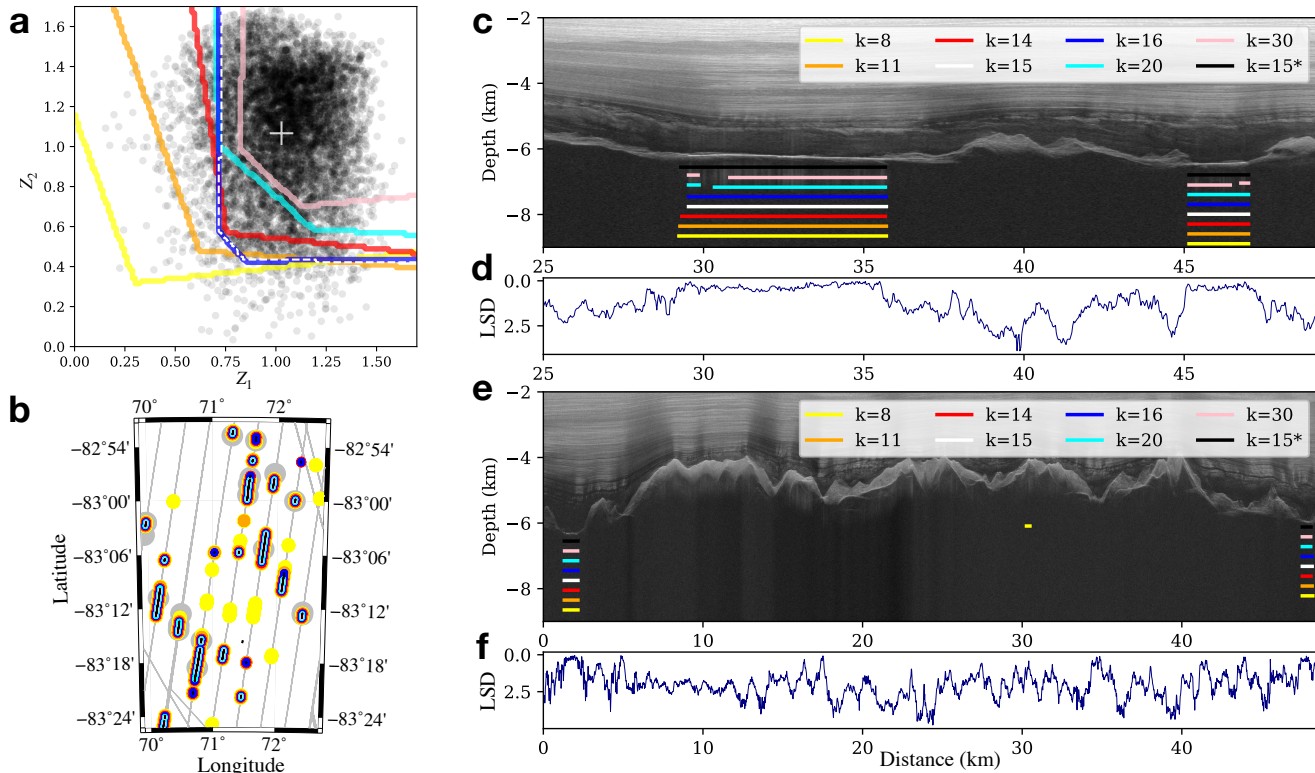

**Figure 11.** Differences in lake detections under different K values. (a) Sectional latent space. Curves in different colors indicate the boundaries between the lake and non-lake clusters when different K-values are applied in the clustering, with yellow for K=8, orange for K=11, red for K=14, white dashed line for K=15, blue for K=16 (partial overlapping with the white dashed line), cyan for K=20, and pink for K=30. Black points denote the reflector vectors from detected lakes in this study, and the white cross denotes the centroid of all lake vectors. (b) Regional spatial distributions of detected subglacial lakes in the map. The map area is truncated from the black box in Figure 8, where gray points represent the known lake inventory (Livingstone et al., 2022), and black lines indicate the detected ranges of subglacial lakes when K=15. Stacked points in various colors represent the detected lake distributions under different K values, corresponding to Figure (a). (c,e) Difference of subglacial lake detection under different K-values in radar image samples from Figure 4 and Figure 9a, where differently colored lines represent the detected lake ranges under different K values. Black lines denote the detection ranges when K=15, with 5% of the dataset applied in clustering. (d,f) Latent space distances (LSD) between bottom reflector vectors to the centroid of detected lake vectors, derived from Figure (c) and (e) respectively.

from the reflectors. Hence, latent space distance serves as a statistical similarity indicator for reflector features. Using the newly compiled subglacial lake list, we can trace vectors corresponding to lake reflectors within the newly detected lake ranges (depicted as black points in Figure 11a). These vectors contribute to a robust dataset, establishing a capable centroid of lake vectors in the latent space, denoted by the white cross in Figure 11a. The disparity between the lake centroid and each reflector can be quantified using latent space distance, serving as an index for the reflector's similarity to the lake echo feature.

Figure 11d and 11f demonstrate latent space distances (LSD) from the lake centroid, where the y-axes are reversed. Reflectors, which are similar to lake features, exhibit continuous and flat peaks (close to 0), while other reflectors display larger differences. Some regional reflectors also show brief high similarity (e.g., $\sim 4.5$km in Figure 11f), possibly corresponding to smaller water bodies or water tunnels, overlooked by the minimum lake width threshold. For future studies, with the detection of more lakes from different regions, a more precise centroid of lake vectors can be established. Moreover, an ample sample size will yield a more credible lake boundary in latent space and a reliable threshold for the similarity index based on latent space distance.

Given the potential flattening effect on the vector distribution in the latent space by the variational module in VAE (Figure 3a), we conduct a comparative analysis using an auto-encoder lacking the variational module (Figure S5). We compute the 2-D probability density for both distributions. In contrast to the VAE distribution (Figure S5a), the distribution derived from the auto-encoder without the variational module (Figure S5b) displays a more uniform trend and lacks discernible cluster patterns. Compared with the auto-encoder without the variational module, VAE provides a continuous latent space (Doersch, 2016), facilitating the direct tracing of waveforms from different clusters through synthetic waveforms generated from the latent space (Figure 3b).

In this study, the final subglacial lakes are obtained using radar echo power filtering, which is based on the linear relationship between reflection power and ice thickness (depth of ice bottom). However, this simple linear threshold filtering potentially excludes subglacial lakes with weaker echo power. To improve the detection of weaker subglacial lake signals, more precise filtering strategies that take into account the roughness and slope of the ice bottom may be beneficial.

Although the encode-cluster method provides an abstract classification for ice bottom reflections, the physical properties of the ice bottom reflection and the corresponding cluster still require further interpretation. The VAE encoder maps high-dimensional reflections to a vector that map to the reflector waveform feature. In the future, physical modeling and in-situ drilling may provide more direct relationships between the latent vectors and subglacial conditions, thereby enhancing the understanding of this subglacial lake detection method.

In addition to subglacial water bodies, other clusters of ice bottom reflections also exhibit some consistent patterns, as illustrated in Figures 7, 9b, and 10b. These resemblances may originate from similar subglacial conditions, particularly the thick layer-like reflections that could correspond to different stages of frozen-on ice. The encode-cluster method is capable of isolating these reflection clusters, offering potential reference for studying glacier dynamics. Geostatistical modeling based on subglacial topography (MacKie et al., 2020) may provide additional references for the reflector clusters in corresponding subglacial conditions. Furthermore, the novel interpretation of latent encoding and clustering could enhance conventional geostatistical analysis by directly utilizing the encoded or clustered results as input or reducing input data dimensions.

The detection method used in this study is based on deep learning, allowing for an automated analysis of data. Deep learning extractors, such as EisNet (Dong et al., 2021), developed in recent years can efficiently pick up the bed interface in radar images. By combining these two types of deep learning methods, an automatic method can be implemented to first extract the positions of the ice bottom and then analyze the features of the bottom reflector, which can further update the subglacial lake inventory by applying this combined deep learning method in the available IPR database. The data used in this study is focused on the

Gamburtsev Subglacial Mountains and can be extended to other database's radar image analyses covering, e.g., the Arctic, Antarctic, and Qinghai-Tibet Plateau. As such, has potential applications for analyzing and tracing spatiotemporal changes in global subglacial lakes and other ice bottom reflection features. Furthermore, this method based on vertical radar waveform also enables the single-trace waveform analysis from A-scope radar data, especially for early observations(Schroeder et al., 2022). The VAE-cluster method trained on Earth data can also provide a potential reference for analyzing ice bottom reflection from Mars' southern ice cap(Orosei et al., 2018) by the spacecraft radar measurements such as SHallow RADar (SHARAD, Seu et al., 2007) and Mars Advanced Radar for Subsurface and Ionosphere Sounding (MARSIS, Picardi et al., 2004).

## 5 Conclusions

We constructed a dataset of ice bottom reflection signals based on IPR data from the Gamburtsev Subglacial Mountains region in the CReSIS database. Using the VAE, we encoded and reconstructed the reflection signal features in the dataset. By applying K-means clustering to the encoded features, we separated the reflector features corresponding to subglacial lakes. By considering the relationship between the peak reflection power and ice thickness, we filtered subglacial lake candidates in this region. Compared with existing inventories, our method can effectively detect features of subglacial lakes and extract more smaller subglacial lakes. This method has potential applications in expanding the subglacial lake inventory and interpreting other subglacial conditions.

*Data availability.* The IPR images and bed echo markers used in this study were obtained from the CReSIS (Center for Remote Sensing of Ice Sheets) database at https://cresis.ku.edu/.

*Author contributions.* SD, LF and XT conceptualized the study. SD, LF and ZL implemented the methods and analysis. SD, LF, XT and XC interpreted the results. SD, LF and XT wrote the manuscript with corrections from ZL and XC.

*Competing interests.* The authors acknowledge there are no conflicts of interest recorded.

*Acknowledgements.* This study was supported by the National Natural Science Foundation of China (Nos. 42276257, 41941006, 41974044), the National Key Research and Development Program of China (No.2021YFC2801404) and "CUG Scholar" Scientific Research Funds at China University of Geosciences (Wuhan) (Project No.2022132). The authors would like to express their gratitude to Professor Dustin Schroeder from Stanford University and Professor Tong Hao from Tongji University for their valuable suggestions on this work. Additionally, the authors would like to extend their appreciation to the reviewers, Michael Wolovick, and Veronica Tollenaar, for their helpful comments and contributions that greatly improved this study.

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
