# Peer review of "Deep Clustering in Subglacial Radar Reflectance Reveals Subglacial Lakes"

_The Cryosphere, 2023_

## Referee Comment (RC1)

**Review of, "Deep Clustering in Radar Subglacial Reflector Reveals New Subglacial Lakes", by Sheng Dong et al., 2023**

Review by Michael Wolovick

**Overview**

In this manuscript, the authors apply Deep Learning (DL) techniques to the problem of identifying subglacial lakes using ice-penetrating radar data. They use a multi-step method consisting of first encoding the information contained in the shape of each vertical reflector trace into a lower-dimensional latent space, and then applying a clustering algorithm to that space in order to identify populations with similar trace shapes. Of these clusters, they identify the one with a narrow symmetric peak in reflection power as representing subglacial lakes, and they further refine the population of subglacial lakes by performing a simple linear attenuation correction to the bed reflection power. Thus, their final identification procedure can be viewed as containing two parts: one part, the encoding-clustering analysis, focuses on the shape of the reflector trace, looking for reflections that are narrow and symmetric, as would be expected for a specular interface. The second, the attenuation analysis, focuses on the strength of the reflector rather than the shape. The authors apply their method to the AGAP-S dataset from East Antarctica and compare their identified lakes with previously published lake compilations.

This manuscript is appropriate for publication in The Cryosphere. It represents a new method in the analysis of ice-penetrating radar data with machine learning techniques and a new method for the identification of subglacial water. However, before it can be accepted in final form, I think that the authors need to provide more justification and explanation around their choice to use a clustering algorithm and on their choice of a particular number of clusters to use in that algorithm. In the remainder of my review, I first explain my major concern about the clustering algorithm, and then I give detailed comments on the rest of the paper.

**Major Concern**

My biggest concern with the analysis in this paper is the decision to use a clustering algorithm, which splits the data into discrete non-overlapping clusters, and the arbitrary decision to use 15 clusters. The data presented in Figure 3a do not appear to display any inherent clustering on their own. Rather, the data points appear to vary continuously across the Z1,Z2 plane. This point is confirmed by the author's own elbow curve (supplemental figure 1), which does not display a clear cutoff, and this point is acknowledged by the authors themselves on lines 166-168. Thus, a mode of analysis that breaks the data up into discrete categories may not be appropriate, and the authors' decision to use 15 categories is arbitrary and unsupported.

Nonetheless, subsequent steps in the analysis protocol are dependent on the use of a clustering analysis earlier on. The ultimate end state of the analysis- a list of positively identified water bodies- requires that the data be split into discrete categories at *some* point. At some point a threshold must be applied to distinguish "water" from "not water", and if the clustering analysis is not used for this purpose, then some other means of setting a threshold must be

used.  Additionally, if I have interpreted the authors' reflection power analysis correctly, then the clustering analysis may also be needed at this stage as well, since I think that they are averaging reflection power values within contiguous reflectors identified through cluster analysis before analyzing reflection power (but note that their explanation of this part of the method was somewhat unclear, so I am not 100% confident that I have interpreted their procedure correctly; I discuss the need for more clarity around this method in the Detailed Comments section below). Averaging reflection power within contiguous similar reflectors is a reasonable methodological choice, since reflection power can be quite variable along-track.  Basically, the authors are using the clustering analysis to identify contiguous regions along the bed that have basically the same reflection trace, and they are defining each of those regions as "one reflector" with a single average reflection power for the purpose of reflection power analysis.

Thus, I face a dilemma:  on the one hand, I do not want to recommend that the authors remove the cluster analysis entirely, since doing so may necessitate downstream changes throughout their method, including changes to parts of the method that seem sensible; but on the other hand, the data do not seem to support the use of discrete clusters and the particular choice of 15 clusters is unsupported.  The arbitrary decision to use 15 total clusters can be regarded as an indirect means of setting the threshold separating "water" from "not water", since the size of each cluster varies inversely with the total number of clusters.  A low total number of clusters will increase the size of each individual cluster, thus increasing the diversity of reflectors in the "water" cluster, while a high total number of clusters will make the "water" cluster smaller. The authors state on line 168 that they have tested different values of K (the total number of clusters), but they do not show the results of those tests in the manuscript.

Therefore, as a minimum condition for publication, I think that the authors should show the results of some of these sensitivity tests.  How does the final population of water bodies depend on the choice of K?  What percentage of the identified water bodies are robust to the choice of K?  Perhaps the subpopulation of water bodies that emerge for multiple values of K could be considered a more robust identification of subglacial lakes.  (As an aside, if some values of K result in the clustering algorithm splitting the upper right "water corner" of Z1,Z2 space into two clusters, then that would be a valid argument for omitting those values of K; the K sensitivity test should only include values of K for which there is one unambiguous "water cluster").  Alternatively, if the authors can provide argumentation to justify their particular choice of K=15, then that could also satisfy my concerns, although the authors' own statement on lines 166-168 seems to indicate that they do not believe a particular value of K is supported by the data.  It may also be worth plotting the elbow diagram (supplemental figure 1) on log/log axes to see if a corner emerges when the data are plotted in that fashion.  Once the authors have either justified their particular choice of K, explored the sensitivity of their results to K, or both, then I think that this manuscript will make an excellent addition to *The Cryosphere*.

**Detailed Comments**

L14

Some of these references aren't really appropriate to use as a general background on subglacial water. Robin 1970 is about ice-penetrating radar, Siegert 2000 is about subglacial

lakes, and Pattyn 2010 is about the results of a specific model. Pattyn and Siegert could work, although they aren't necessarily the best citations for this purpose, but Robin 1970 is definitely the wrong reference to use here. Chapters 6 and 9 of (Cuffey and Paterson, 2010) could be cited here, although I understand that citing a textbook is a bit unsatisfying. Another important reference might be (Robin, 1955).

L21 "...in recent years…"
        I don't know if it is fair to describe the use of ice-penetrating radar for detecting the subsurface features of ice sheets as "recent". Maybe this sentence would be better as, "Ice-penetrating radar can be used to detect the subsurface features of ice sheets". Also, this would be a good place for the Robin (1970) reference, not L14. Another good reference might be (Robin et al., 1969) or (Bailey et al., 1964).

L50: "from the CReSIS"
        Should be "from CReSIS".

L52: "We then apply K-means clustering method"
        This sentence should be reworded in one of the following 3 ways: 1) "We then apply the K-means clustering method", 2) "We then apply a K-means clustering method", or 3) "We then apply K-means clustering methods". Wording (1) applies if there is only 1 version of the K-means clustering method, wording (2) applies if there are multiple versions of the method but you only use 1 of them, and wording 3) applies if you use multiple versions of the method.

L54: "We notice a cluster"
        "We identify a cluster" sounds better.

L61: "...to detect and label the other clusters…"
        "The" is unnecessary here.

L72: "The radar images also contain the positions of ice bottom reflectors, which were extracted by hybrid manual-automatic method (Wolovick et al., 2013)."
        Note that the bed picks produced by (Wolovick et al., 2013) are not the same bed picks included in the CReSIS data release. The AGAP-S data was processed in parallel at both the Lamont-Doherty Earth Observatory (LDEO) and CReSIS. The results of the LDEO processing are available at: https://pgg.ldeo.columbia.edu/data/agap-gambit. Though the original raw data is the same for both institutions, the code for SAR migration and bed picking was different, and of course different human operators provided the "manual" part of the manual-automatic bed picking. Both institutions used "hybrid manual-automatic" bed picking, but it might be better to cite a CReSIS source here if you are using the CReSIS version of the data.

L81: "Second, we apply the reflector position markers in the dataset to truncate the 1-D data within the ±200 sampling points near the reflector position for every single trace along Z-axis."

It might clarify things a bit to say that the reflector in this case is the bed. So maybe, "Second, we use the bed picks in the dataset to truncate the 1-D data within ±200 sampling points near the bed reflector position for every single vertical trace."

L88: Gaussian filter, normalization.
    What is the filter width? How are the data normalized?

L91: "1488600 1-D Z-axis (A-Scope) radar echo traces."
    Were you using the original 1.3 m along-track spacing of the data, or are you working with data that have been downsampled? I did not work with the CReSIS version of the AGAP-S data, but I know that other CReSIS data products are generally released at coarser horizontal resolution than this, and when processing the AGAP-S at LDEO, I downsampled the data by a factor of 10 in the along-track dimension. The original 1.3 m data should have a lot more than 1.4 million traces. If you are using downsampled data, then you should mention that.

L105-110: MSE
    What does MSE stand for?

L132: "...are challenging to be reconstructed…"
    Change to: "...are challenging to reconstruct…"

L166-168: "However, the elbow curve does not show a clear cutoff point, possibly due to the distribution of vectors in the latent space (Figure 3a) not displaying a distinct trend of multiple classes."
    The "elbow curve" for this method seems to be analogous to an L-curve for inverse problems. Could you please present the elbow curve on log-log axes in addition to the linear axes you used in your supplemental figure? The problem with linear axes for this purpose is that inverse power laws always appear L-shaped on linear axes, despite having no intrinsically preferable value.
    Additionally, when I look at Figure 3a, it appears to my eye that the data do not really have any clusters at all. Is that what you meant by "not displaying a distinct trend of multiple classes"? To my eye, it looks like the data are smoothly distributed within the central part of the latent space, with perhaps a greater number of outliers in the negative direction for both Z1 and Z2 than in the positive direction. It sounds as though the lack of visual clustering in Fig 3a is confirmed by the lack of a clear cutoff in the elbow curve. I discuss this issue at greater length in my "Major Concern" section above.

Figure 3b
    It might be easier to read and interpret this figure if you used a 10x10 grid of virtual waveforms instead of a 20x20 grid, and then made the individual waveforms twice as large. Additionally, it might be better to make the individual waveforms all black, and then overlay cluster boundaries as lines.

L185-192: Identifying subglacial water using clustering analysis

It seems as though your major reason for using clustering analysis was to get to this step, where you use your method to automatically identify water.  The basic argument you are making here seems to be that the upper right quadrant of Z1,Z2 space contains symmetrical sharp reflectors, and these reflectors are more likely to be water.  This argument is simple, robust, and I believe it.  But in addition to identifying points in this quadrant using a clustering analysis with an arbitrary number of clusters, it may help to make your analysis more robust if you also constructed alternate metrics to identify points in this quadrant.  For instance, you could select traces for which Z1 and Z2 are both more than 1sigma above the mean.  Or you could make a combined water index, I=Z1+Z2, and then select points with a high value of this index.  Alternatively, you could construct a not-water index by taking the euclidean distance from each point to the upper right corner (+2sigma,+2sigma).  These sorts of continuous water indices may help reduce the dependence on an arbitrary choice of K.

L199:  "Detected subglacial water bodies should contain a continuous ice bottom segmentation in subglacial water type with a width greater than 8 traces (corresponding to an average spatial distance of 10.4 m)."
        You should double-check the along-track spacing of the data product you use.  If the data have been downsampled from the original 1.3 m spacing, then your 8 trace threshold will correspond to a longer distance.

Figure 4c:
        This plot would definitely benefit from a continuous approach to reflector categorization.  The different colors here represent different categories, but it is hard to tell how close each category is to the water category.  By contrast, a "water index" would provide a continuous metric that could be displayed here.

L204-218, Figure 5: Reflection power analysis
        Did you correct the bed returned power for geometric spreading before doing this analysis?  Signal loss with depth comes from both attenuation within the ice and from simple geometric spreading with range.  The effect of geometric spreading can be calculated and removed.
        Additionally, I am curious whether Figure 5 shows the entire dataset, or only a subsample of the dataset?  When I did a similar analysis for the 2013 paper, I found many data points that were 3sigma or even 4sigma above the linear best-fit.  However, in this figure it looks like you would have perhaps 2 or 3 data points at a 3sigma level, and no data points at a 4sigma level.  Is the total sample size smaller here?  What exactly is being plotted in Figure 5?  Does each point represent a single trace, or does each point represent an along-track average of candidate water bodies?  I feel like this method needs a better explanation.
        After thinking about it for a bit, my guess is that you have done something like this:
1) Apply clustering analysis to the traces
2) Apply the 8-trace rule to generate contiguous reflectors that all belong to a particular cluster.
3) Compute average peak power for each contiguous reflector
4) Perform the attenuation analysis using this smaller dataset of horizontally averaged power data

Am I correct?  Is that the procedure that you followed?  If yes, then this should be explained in more clarity.  In particular, it should be clear that you used grouped adjacent reflectors according to their cluster, and that you have done this for all clusters, not just the water cluster.  If this is what you have done, then that also explains why you find far fewer high-reflectivity outliers than I did in the 2013 paper, since peak reflection power can be highly variable along-track and the averaging process will tend to reduce the amplitude of individual bright spots.

It also seems to me that the arbitrary choice to use 15 clusters will have a big impact at this stage, since it will determine the along-track length of contiguous regions that you average together into the analysis.  It would be interesting to see in the sensitivity analysis how changing the value of K affects this part of the analysis.

FIgures 4, 6, 7, 9, 10:  Radar results figures
These figures would all benefit from being zoomed in on the bed.  The vertical scale could be cropped between 2 km and 4km (or perhaps slightly below 4km, to accommodate the deep lake in Fig 9c).  In addition, the color scale of the echograms should be adjusted so that the lower limit is just a bit below the noise floor and the upper limit is closer to the brightest bed.  These changes would make it easier to follow along when the text goes into detail about specific features in these figures.

L260  "...but some are also sparsely detected."
This wording is awkward.  Perhaps, "...but some isolated points are also detected."

Figure 8: map figure
It is hard to tell what the text labels (L#, E#, N#) refer to.  Maybe you could move the text labels further away from their targets, and then add annotation arrows pointing from the text to the target?

Additionally, the main map should be bigger and the other elements of this figure should be smaller.  There is way too much empty white space in this figure.  The main map containing the central AGAP survey contains all of the important information in this figure.  Therefore, that main map should be as big as possible.  The other two items in the figure, the inset location map and the legend, can be placed in unused corners of the main map.

L255-268:  L#, E#, N#
What do L, E, and N stand for?  Are N1-N4 new subglacial lakes?

L 278:  "Considering the dense distribution of subglacial water bodies nearby, these thicker reflection features are possibly formed by frozen-on ice due to ice flow."
Freeze-on isn't caused by ice flow.  Freeze-on is caused by either conductive cooling or supercooling.  Perhaps a better way to phrase this sentence would be, "Considering the dense distribution of subglacial water bodies nearby, these thicker reflection features are possibly formed by frozen-on ice that complicates the shape of the near-basal reflection trace."

L281:  "...the sparse but regionally dense distribution…"

What exactly does "sparse but regionally dense" mean?

L309: "The unsupervised clustering analysis applied in the latent vectors relies on the implied feature difference of the reflection waveform, effectively excluding subjective and external factors in finding potential classifications of subglacial conditions, and reducing the dependence on model assumptions."

Except for the subjective choice to use 15 clusters. This choice has downstream effects in terms of determining the size of the "water" cluster (because average cluster size should vary inversely with the number of clusters), so this arbitrary choice indirectly determines how much variability in reflector shape you are willing to tolerate while still calling something "water". Additionally, the choice to use a 2D latent space instead of a higher dimensional space was also arbitrary. All methods require some degree of human choice on the part of the scientists employing the method.

It seems to me that the big advances achieved here are in 1) having a new method to quantify and classify the shape of the reflection waveform, and 2) using that method to help classify the physical setting of the ice sheet bed, particularly by helping to identify subglacial lakes. It is not really fair to say that you have excluded subjective and external factors, those factors simply enter into your analysis in a different way than they do in other analyses.

**References**

Bailey, J. T., Evans, S., and Robin, G. de Q.: Radio echo sounding of polar ice sheets, Nature, 204, 420–421, https://doi.org/10.1038/204420a0, 1964.

Cuffey, K. M. and Paterson, W. S. B.: The Physics of Glaciers, 4th ed., Butterworth-Heineman/Elsevier, Burlington, MA, 2010.

Robin, G. D. Q., Evans, S., and Bailey, J. T.: Interpretation of radio echo sounding in polar ice sheets, Philos. Trans. R. Soc. Lond. Math. Phys. Eng. Sci., 265, 437–505, https://doi.org/10.1098/rsta.1969.0063, 1969.

Robin, G. de Q.: Ice movement and temperature distribution in glaciers and ice sheets, J. Glaciol., 2, 523–532, 1955.

Wolovick, M. J., Bell, R. E., Creyts, T. T., and Frearson, N.: Identification and control of subglacial water networks under Dome A, Antarctica, J. Geophys. Res. Earth Surf., 118, 140–154, https://doi.org/10.1029/2012JF002555, 2013.

---

## Referee Comment (RC2)

**Review of "Deep Clustering in Radar Subglacial Reflector Reveals New Subglacial Lakes", Sheng Dong et al., The Cryosphere Discussions, 2023.**
Veronica Tollenaar

**General comments**

The paper discusses a subglacial lake detection method applied to a region near the center of the continent of Antarctica. With the available data, the problem can be seen as a positive and unlabeled problem, where some subglacial lakes have been outlined in earlier studies (positive labeled examples), while for the remaining area the presence or absence of subglacial lakes is unknown (unlabeled examples). The authors take an unsupervised learning approach to this problem, which is a valid choice.

The unsupervised learning consists of an auto-encoder, which basically reduces the dimensionality of the data, and a clustering, where one of the clusters is assumed to correspond to the presence of a subglacial lake. Although this approach is smart, novel, and has a high potential in delineating subglacial lakes, I see several weakly motivated choices in the methodology that I will also try to outline further through the specific comments per section.

My main issue is that the authors perform a clustering analysis on a (2-dimensionally) normally distributed set of samples. These samples are normally distributed through the applied loss function in the encoder. However, per definition, in this set of samples there is only a single cluster, otherwise the loss function should have allowed a certain number of gaussian distributions in the latent space. This caveat is also confirmed by the fact that there is no clear cutoff point in the elbow function to determine the number of clusters present in the data. In my view there are three potential approaches to adjust the manuscript to overcome these caveats in the methodology.

(i) The authors can illustrate quantitatively that the results are convincing, despite the conceptual problem with the methodology, making the study a pragmatic approach toward subglacial lake detection. With the absence of correctly labeled negative examples (i.e., the absence of subglacial lakes), traditional performance metrics such as precision and accuracy cannot be estimated. Nevertheless, a sensitivity estimate of the results, which is currently not part of the manuscript, can be included.

(ii) Instead of the clustering, the authors can identify where the currently known subglacial lakes are located in the latent space (i.e., plot these samples in Figure 3a). As "the distance between vectors in the latent space can serve as a statistical similarity indicator for reflector features" (Line 308-309), samples within a certain distance from the located latent-space vector of known subglacial lakes can be identified as subglacial lakes.

(iii) The authors could use another approach to deep clustering as discussed in various deep learning literature. The simplest solution would be to use an auto-encoder instead of a variational auto-encoder, despite obtaining a less meaningful latent space in the sense that the distance between latent vectors does not reflect a similarity. Nevertheless, it might appear that there are distinct clusters in the latent space.

I think that through adopting (a combination of) the above approaches, or by taking another approach that overcomes the illustrated problem, the study can significantly contribute to the development of an automated approach for the detection of subglacial lakes. This method will be essential to process the ever-growing amounts of data across the continent (and beyond) efficiently, and the authors already convey this message clearly through an elaborate discussion of their results and informative figures.

**Specific comments per section**

**Title and abstract**

Title: I think "Subglacial Radar Reflectance" sounds better than "Radar Subglacial Reflector". Also, apart from a very elaborate qualitative analysis of the results, there is no hard or independent evidence that the detected lakes are really lakes, let alone that they are "new", which implies that they were not there before (in time). Leaving the word "new" out of the title solves this issue. Otherwise, rephrasing toward something like "An automated method for subglacial lake detection based on deep clustering" could be nice, but it depends on the intention of the authors.

Line 3: It is confusing to read that you generate a dataset. Maybe better to rephrase as "In this study, we use available IPR images in the Gamburtsev Subglacial Mountains to extract one-dimensional reflector waveform features of the ice-bedrock interface."

Line 4: The method remains very mystical, maybe good to clarify that you apply a deep learning method to reduce the dimension of the data so that you can perform a cluster analysis.

**1 Introduction**

Line 13: The sentence does not read well. I would suggest: "Subglacial water, i.e., water between bedrock and ice sheet, is formed through a complex interplay.."

Line 15: Potentially also include the recent publication of Kazmierczak et al. in The Cryosphere:

E. Kazmierczak, S. Sun, V. Coulon, F. Pattyn, Subglacial hydrology modulates basal sliding response of the Antarctic ice sheet to climate forcing. The Cryosphere, **16**, 4537–4552 (2022).

Line 16-20: The importance of research in subglacial lakes is well outlined, but the order is a bit confusing. I would start with the ice sheet meltwater (following the previous sentence about ice flow and dynamics), then the history of climate change and ice sheet evolution, then the subglacial lake sediments, then the unique lacustrine ecosystems.

Line 21: Potentially write out the acronym of radar (radio detection and ranging).

Line 21: Potentially remove "in recent years", the next sentence refers to a publication of 1973.
Line 22: The sentence starting with "Subglacial water bodies" could fit better in the next paragraph, where these visual features are discussed again.

Line 23: I would swap around the subject and the object of these sentences so that it is easier for the reader to understand that here the authors are going to refer to other measurement techniques: "The thickness of the subglacial water layer and sediment characteristics at the bottom of lakes are also investigated with active seismic surveys (Paden et al., 2010; Arnold et al., 2020) and gravimetry and electromagnetic methods (Studinger et al, 2004, Key and Siegfried, 2017)."

Line 35: the "subjective factors" are not ruled out in this study: heavy postprocessing is applied and the results are discussed mainly in a qualitative way.

Line 36: the "absence of a complete interpretation of basal radar reflectance features" is also the case for the study: only a narrow window including the reflectance near the bedrock is considered, and the spatial context, i.e., along the bedrock, is only considered through a rather pragmatic postprocessing step that filters the results spatially. Deep learning is a powerful tool to consider these spatial relationships directly. If not adapting the methodology to actually rule out "subjective factors" and have a "complete interpretation of basal radar reflectance features", I would suggest a more elaborate and precise discussion of other methods, to illustrate more in detail in which aspects the proposed methodology is better.

Line 37: I would suggest an easier rephrasing: "In recent years, deep learning has been applied as a powerful tool to detect different features in IPR images, including bedrock interfaces, internal ice layers, snow accumulation layers". For the "radar semantic segmentation", that is an automated feature extraction in se, so I'd suggest to either refer to what is semantically segmented or remove.

Line 40: I am not sure if I understand the difference between this sentence and the previous: is the previous specifically about the detection of layers? If not, I would try to combine this sentence with the previous one and specify the subglacial features. For me it is not clear whether the subglacial features refer to anything under the surface or just features at the ice-bedrock interface.

Line 42: I would rephrase this sentence with: "Moreover, deep learning applied to IPR has also contributed to estimates of ice thickness (to enable data application in ice sheet studies.)", with the part in brackets potentially removed.

Line 46: Potentially include a reference to the dataset directly (see: https://data.cresis.ku.edu/#ACRDU)

Line 50: I think it is a bit confusing to use the wording "construct a dataset", it suggests that you collected the data in the field. I suggest the rephrasing: "In this study, we select IPR images in the region of the Gamburtsev Sublgacial Mountains from the CReSIS database. We crop these images around the ice bottom, to obtain a set of one-dimensional waveforms that capture the ice bottom reflectance characteristics. Using this data, we train …"

Line 52: The "time-domain waveform features" are confusing. Either introduce the time-domain aspect in an additional sentence (something like: "The radar is reflected most strongly by the bedrock beneath the ice sheet, resulting in a peak in the return signal received by the radar over time. Moreover, bedrock characteristics, such as roughness or the presence of water, influence the intensity and shape of the peak signal, to which we refer to as the waveform features of basal reflectors.")

Line 55: Do you mean the features that correspond to subglacial lakes?

Line 55: I would specify that this is a kind of post-processing step.

Line 58-60: What is the benefit of extracting reflectors with similar waveform characteristics as water bodies? How does that improve the efficiency and accuracy of the detection of subglacial lakes?

Line 61: Indeed, it is nice that you can characterize/cluster the subglacial features through this method.

**2 Data and Methods**

Figure 1: The Figure looks nice, and summarizes the workflow well, but there are several details that need to be adjusted: What is "Z-Scope"? What is "A-Scope"? "Ice Buttom" should be "Ice Bottom", "Reconstructed Reflector Feature" should be "Reconstructed Reflector" (as in "Ice Bottom Radar Reflector"). Both waveforms need axes with labels (time and power I guess). For the caption "(b) VAE reconstructs and encoding of the sampled ice bottom reflector features." should be changed to "(b) The VAE encodes and reconstructs the sampled ice bottom reflector." For the subpanel (c), the caption says "Supervised", while I think the authors mean "Unsupervised".

Line 69: This sentence about the lake inventories seems out of place. I think, together with the sentence "According to the lakes inventory…" on line 71, these sentences should be moved to the introduction in the paragraph that starts on line 50, so that paragraph 2.1 really focusses on the radar data.

Line 70: I miss a reference here: is it this dataset that's been used? https://data.bas.ac.uk/full-record.php?id=GB/NERC/BAS/PDC/01544

Line 74: "The radar data were acquired from L1B.." can be rephrased to "We use the L1B data product" to avoid confusion whether the data has been acquired by the authors.

Line 81: Is there a physical motivation for truncating the signal to this narrow range around the bedrock? When I see the radar images shown in the different Figures (e.g., Figure 4), I find it remarkable to see a distinct reflectance below the bedrock for each of the subglacial lakes that seems to be not captured anymore by choosing the narrow window.

Line 85: Assuming that the peak signal corresponds to a single point, I would guess the length of the truncated signal would be $64 + 1 + 64 = 129$, but it reads 128.

Line 88: Could you provide the bandwith/sigma of the gaussian kernel?

Line 89: How do you perform this normalization? Somehow I get the impression that all of the nearly 1,5 million (incredible number, congrats!) reflectance traces are normalized individually: or do you calculate a global mean and standard deviation and set these to 0 and 1? If normalized individually, I think this might be the cause of why you need to use the post-processing step where you use the peak power reflectance. I would advise to either (i) normalize all data with the statistics of the entire dataset as otherwise you're comparing different units to each other, or (ii) already implement the depth/power relationship while normalizing, or (iii), more experimental, normalize each individual waveform, but provide the peak power and the ice thickness as additional input to the VAE.

Line 97: What do you mean with the sentence starting with "And the.."? I think it deviates the attention from why you use the VAE: to reduce the dimension of your data.

Line 102: I think you use it to reduce the dimension of the reflector waveform features from the ice bottom, right? It is confusing to think that the goal is to reconstruct something that you already know.

Line 104: Your bottleneck consists of a two-dimensional latent distribution, enforced to follow a normal distribution through using the KL divergence in your loss function. I find the motivation for choosing to sample only two samples from your latent distribution just for visual representation weak. Another motivation can be that it is easier to perform the clustering in two dimensions, or that in other work it has been proven sufficient (for example in the referenced work of Li 2022).

Line 106: Conceptually I don't understand why the KL is used in the loss function: it forces the latent space to be normally distributed, which is essential when using VAE for generative purposes. However, as the authors want to perform a cluster analysis, I think there is a fundamental conflict. Clustering data that is normally distributed will not yield in clearly separable clusters. Or, differently put: the underlying assumption for clustering should be that there a different clusters, which, of course, can be each normally distributed, but through VAE the latent space is constructed as one single big cluster. The fact that there is no clear cutoff point of the elbow curve that the authors want to use to determine the number of clusters confirms that there are no separable clusters in the latent space. I have not read enough into the literature to know whether there are other examples of the approach that the authors take that still yield useful results – but a quick search indicated that there are fancy solutions for this mismatching of concepts, e.g., Lim et al., 2020. A simple solution would be to just use an Auto Encoder and perform the clustering on those results.

Lim, Kart-Leong, Xudong Jiang, and Chenyu Yi, Deep clustering with variational autoencoder. IEEE Signal Processing Letters, **27**, 231-235 (2020).

Line 122: Why do you stop training at epoch 10 if the training loss does not descend more after epoch 4? Can you report the generalization error? If the training loss does not decrease, but you continue training (epoch 5-10), you start to overfit to your training data.

Line 123: The word "evaluate" suggests a quantitative estimation, for example based on independent test data. Could you either provide this, or change to "illustrate"?

Figure 2: Could you provide axes and labels for all subpanels? Could you provide the MSE for all examples? Potentially the learning curve, and the generalization error could be included in this Figure.

Line 139: These vectors consist of two samples from the latent distribution, right?

Line 143: How does this subset vary from the validation subset mentioned in line 119? It seems like you are going to use these samples for clustering and not for "validate the encoder"?

Line 147: That gaussian distribution poses problems for the clustering (see earlier remark about line 106).

Line 148-153: This is almost philosophical, could you rephrase it with more direct wording?

Line 156: I do not directly see that 2000 reflectors are sufficient for clustering. From Figure 3, to me, the clusters seem rather arbitrary. Also, given that you have 1.5 million reflectors and you perform the dimension reduction to enable efficient clustering, I think the sample of 2000 is rather small (~0.1 % of all data). How long does it take to perform the clustering analysis?

Figure 3: The generative capacity of VAE is nice, and Figure 3b is a pretty visualization of this capacity. However, I do miss a link to the physical phenomenon, and therefore I would suggest to remove the subfigure or move it to Supplementary Materials.

Line 169: Here I miss evidence for the statement: what motivates the authors to conclude that there is an effective separation of bottom reflector features? And how do they correspond to different conditions?

Line 171-183: Similar to Figure 3b: a physical interpretation is lacking, and I would move this to Supplementary Materials.

Line 184: In this section the authors discuss how to detect subglacial lakes using the results of the clustering analysis. The main points discussed are related to post-processing steps, and I think this is not clearly reflected in the section title. Potential other titles could be "Subglacial lake detection" or "Post processing to detect subglacial lakes".

Line 192: I do not understand the conclusion here. I guess you want to say that one of the clusters seems to correspond to subglacial lakes, right? Another way to confirm this is to give statistics of to what clusters the waveforms at earlier detected subglacial lakes belong, e.g., 80% of known subglacial lakes have a bottom reflector that falls into cluster x.

Line 198: What do you mean by "based on experimental experience"? Is there a reference? A solution could be to remove that specification.

Line 203: What do you mean by "interpolation artifacts due to specific noise?"

Line 209: If I understood it well, before you used this peak echo power to normalize the data for the encoder. I wonder if this postprocessing step would still be necessary if don't apply this normalization earlier. That would potentially be something to investigate and report on.

Figure 4: For panel d, would it be possible to have the same colors as panel c? So black for the lake, and other colors corresponding to the different clusters that have been filtered out during the post processing?

Line 211: How did you calculate the best linear fit? Somehow, I get the impression that the orange dashed line should be steeper in Figure 5, but this might be an optical illusion.

Figure 5: Potentially only show the +1 sigma as that's the threshold you use, to avoid confusion.

**3 Results**

Line 229-230: If I understand it correct you are claiming that the results are reliable because the subglacial water bodies look like known subglacial waterbodies, right? Out of interest, what do you mean by the geothermal environment in adjacent areas?

Line 237-240: This statement is very similar to the statement in the previous paragraph. I think you do not need to convince the reader of the value of an automated method for detection, it is already clear that this is very valuable.

Line 241: I think it should be "(at about 40 km along the transect)" or so, it looks like the lake is ~3 km wide.

Line 241-253: Nice discussion of results.

Line 255-260: Somehow this paragraph makes me doubt that for the results in Figures 4, 6, and 7, the peak power post-processing step is not applied? Could you clarify that in the text?

Line 260: By "sparsely detected", do you mean that these are isolated lakes? Or just along a single IPR line?

Line 261: Normally it should be "compare to something": rephrase as "We compare the subglacial lakes detected in this study to the previously identified …"

Line 265: remove "which is newly detected", that is already clear from the first part of the sentence.

Line 277: Do you mean that the red arrows show lakes that have not been detected?

Line 278: In Figure 7c you associate the yellow cluster with frozen-on ice and ice flow dynamics. But in Figure 9 it looks like different shades of purple. Do you think multiple clusters do show this frozen-on ice? And are these clusters next to each other (it's hard to link the shades of purple in the Figures with the shades of purple in Figure 3a).

Line 280: I think the origin of the water bodies is very suggestive. What do you mean by the sparse but regionally dense distribution of subglacial water bodies?

Figure 8: I think the Figure is very essential for the study. It took a long time to understand the link between the regions and the labels, but I understand now that it is related to the thin black arrows. Potentially it would be nice to clarify that in the main text, as well as in the caption. Moreover, the two blue colors (blue and cyan), might be confusing, and the labelling can be "detected lakes (no post-processing)" and "lakes (post-processed)" or so, now it is not clear what is what exactly. Other questions that pop up when seeing the figure are: (i) in the region near "N3", going perpendicular to the radar lines, there is a clear line of lakes, does that correspond to a kind of channel in the subsurface topography? It could be interesting to overlay the detections on bed topography data, but that is probably out of scope for this study. (ii) There are a lot of "candidate lakes" on the southern part of the survey, it almost looks like an artifact, is that the case?

Line 287: What do you mean by "differ visually"?

Line 298-304: I think the conclusion is very bold, basically saying that the previous inventories are wrong in places where the authors do not detect lakes. I would be a bit more reserved and steer in the direction that this automated method is promising, and that further investigation is needed (as already suggested). Moreover, there is the remark about "multi-trace detection methods", but in some sense the applied post-processing of grouping 8 neighboring traces makes this method also a "multi-trace detection method", right? Or is this not applied for obtaining the map?

**4 Discussion**

Line 307: I understand what you mean by "all reflection information", but actually you crop the reflectance to contain only the signal of the bottom.

Line 308: I miss a sentence that states what has been done, something like "We encoded the waveforms to obtain two-dimensional vectors that conceptually summarize the waveform in the so-called latent space of an auto-encoder. The distance between vectors in the latent space…"

Line 328: What do you mean by this sentence? The clustering analysis can be used as input for other models?

Line 330: What do you mean by "an automated analysis data"? "automated analysis of the data"?

Line 336: "As such, the method has potential.."

Line 337: What do you mean by classifications for single-track radar data?

Line 339: Sorry for the noob question: does ice penetrating radar on Mars exist? Can you obtain those kinds of observations from space? And in general, DL methods are known to perform badly on out-of-distribution examples, so is it realistic to apply the method to data that is very dissimilar from airborne observations?

**5 Conclusions**

Concise, clear

**Data availability**

Will you share your clustered data, i.e., the data in Figure 4, 6-10? Will you share your code in a repository?

**Technical comments**

Line 21: introduce the acronym IPR here

Line 21: "subsurface feature**s"**

Line 37: remove (DL), acronym is not used often in the paper, and it complicates reading.

Line 42: "These deep learning-based approaches"

Line 66: remove "reduction", add "the" before variational auto=encoder

Line 116: "n" instead of "N"

Line 149: Brackets around Kingma and Welling 2013

Line 185: "different type's ice bottom" should be "different types of ice bottom"

Caption Figure 4: "Fist example" instead of "Example 1"

Caption Figure 4: "Results of the unsupervised clustering of the latent vectors"

Line 220: "dataset" instead of "database"

Line 243: Remove "This subglacial … return power", it repeats the previous sentence

Caption Figure 7: "continuous" instead of "continus"

Line 270: "Figure" instead of "Figures"

Line 335: "covering the Arctic" can be "covering, e.g., the Arctic"

Line 356: remove "A."

---

## Author Comment (AC2)

**Response to Veronica Tollenaar (RC2)**

**General comments**

The paper discusses a subglacial lake detection method applied to a region near the center of the continent of Antarctica. With the available data, the problem can be seen as a positive and unlabeled problem, where some subglacial lakes have been outlined in earlier studies (positive labeled examples), while for the remaining area the presence or absence of subglacial lakes is unknown (unlabeled examples). The authors take an unsupervised learning approach to this problem, which is a valid choice.

The unsupervised learning consists of an auto-encoder, which basically reduces the dimensionality of the data, and a clustering, where one of the clusters is assumed to correspond to the presence of a subglacial lake. Although this approach is smart, novel, and has a high potential in delineating subglacial lakes, I see several weakly motivated choices in the methodology that I will also try to outline further through the specific comments per section.

My main issue is that the authors perform a clustering analysis on a (2-dimensionally) normally distributed set of samples. These samples are normally distributed through the applied loss function in the encoder. However, per definition, in this set of samples there is only a single cluster, otherwise the loss function should have allowed a certain number of gaussian distributions in the latent space. This caveat is also confirmed by the fact that there is no clear cutoff point in the elbow function to determine the number of clusters present in the data. In my view there are three potential approaches to adjust the manuscript to overcome these caveats in the methodology.

(i) The authors can illustrate quantitatively that the results are convincing, despite the conceptual problem with the methodology, making the study a pragmatic approach toward subglacial lake detection. With the absence of correctly labeled negative examples (i.e., the absence of subglacial lakes), traditional performance metrics such as precision and accuracy cannot be estimated. Nevertheless, a sensitivity estimate of the results, which is currently not part of the manuscript, can be included.

(ii) Instead of the clustering, the authors can identify where the currently known subglacial lakes are located in the latent space (i.e., plot these samples in Figure 3a). As "the distance between vectors in the latent space can serve as a statistical similarity indicator for reflector features" (Line 308-309), samples within a certain distance from the located latent-space vector of known subglacial lakes can be identified as subglacial lakes.

(iii) The authors could use another approach to deep clustering as discussed in various deep learning literature. The simplest solution would be to use an auto-encoder instead of a variational auto-encoder, despite obtaining a less meaningful latent space in the sense that the distance between latent vectors does not reflect a similarity. Nevertheless, it might appear that there are distinct clusters in the latent space.

I think that through adopting (a combination of) the above approaches, or by taking another approach that overcomes the illustrated problem, the study can significantly contribute to the development of an automated approach for the detection of subglacial lakes. This method will be essential to process the ever-growing amounts of data across the continent (and beyond) efficiently, and the authors already convey this message clearly through an elaborate discussion of their results and informative figures.

**Reply:** Much appreciate your encouraging comments and valuable suggestions. We have updated the manuscript according to your concerns, as the following points:

(i) We have appended the clustered areas in latent space corresponding to subglacial lakes when different K value is applied in clustering analysis. We also traced the detected ranges of subglacial lakes in different K values applied.

(ii)We trained another auto-encoder which contains no variational module, and used the same reflector samples as Figure 3 to exhibit the latent space distribution. We have appended an additional comparison between VAE and Auto-Encoder on the same samples' latent space distributions and their probability density estimations.

For advice (ii), we agree that locating (mapping) the known subglacial lakes from the latest lakes list in latent space could provide a more reasonable indicator for the newly detected lakes. However, the known subglacial lakes list only implied the location of each lake. The absence of widths/ranges of known subglacial lakes makes only one trace of the reflector that can be extracted based on the longitude and latitude, which may also induce few known indicators that can be utilized in latent space and influence the detection. So, we consider this known lakes-supervised method to be a potential application for future study.

Thanks again for your detailed suggestions to benefit our work.

**Specific comments per section**

**Title and abstract**

Title: I think "Subglacial Radar Reflectance" sounds better than "Radar Subglacial Reflector". Also, apart from a very elaborate qualitative analysis of the results, there is no hard or independent evidence that the detected lakes are really lakes, let alone that they are "new", which implies that they were not there before (in time). Leaving the word "new" out of the title solves this issue. Otherwise, rephrasing toward something like "An automated method for subglacial lake detection based on deep clustering" could be nice, but it depends on the intention of the authors.

**Reply:** Thanks for your advice on the title. We have modified the title according to your advice.

Line 3: It is confusing to read that you generate a dataset. Maybe better to rephrase as "In this study, we use available IPR images in the Gamburtsev Subglacial Mountains to extract one- dimensional reflector waveform features of the ice-bedrock interface."

Line 4: The method remains very mystical, maybe good to clarify that you apply a deep learning method to reduce the dimension of the data so that you can perform a cluster analysis.

**Reply:** Thanks for the indications. According to your advice, we have modified and simplified these sentences.

**1 Introduction**

Line 13: The sentence does not read well. I would suggest: "Subglacial water, i.e., water between bedrock and ice sheet, is formed through a complex interplay.."

**Reply:** Done. Thanks.

Line 15: Potentially also include the recent publication of Kazmierczak et al. in The Cryosphere:

E. Kazmierczak, S. Sun, V. Coulon, F. Pattyn, Subglacial hydrology modulates basal sliding response of the Antarctic ice sheet to climate forcing. The Cryosphere, 16, 4537–4552 (2022).

**Reply:** We have added this citation.

Line 16-20: The importance of research in subglacial lakes is well outlined, but the order is a bit confusing. I would start with the ice sheet meltwater (following the previous sentence about ice flow and dynamics), then the history of climate change and ice sheet evolution, then the subglacial lake sediments, then the unique lacustrine ecosystems.

**Reply:** Thank you for your helpful advice, we have modified the order.

Line 21: Potentially write out the acronym of radar (radio detection and ranging).

Line 21: Potentially remove "in recent years", the next sentence refers to a publication of 1973.

**Reply:** Done. Thanks.

Line 22: The sentence starting with "Subglacial water bodies" could fit better in the next paragraph, where these visual features are discussed again.

**Reply:** We have moved this sentence to the next paragraph, thanks for your advice.

Line 23: I would swap around the subject and the object of these sentences so that it is easier for the reader to understand that here the authors are going to refer to other measurement techniques: "The thickness of the subglacial water layer and sediment characteristics at the bottom of lakes are also investigated with active seismic surveys

(Paden et al., 2010; Arnold et al., 2020) and gravimetry and electromagnetic methods (Studinger et al, 2004, Key and Siegfried, 2017)."

**Reply:** We have swapped these sentences to the improved version. Thanks a lot for advising.

Line 35: the "subjective factors" are not ruled out in this study: heavy postprocessing is applied and the results are discussed mainly in a qualitative way.

**Reply:** We have removed "subjective factors" in this sentence.

Line 36: the "absence of a complete interpretation of basal radar reflectance features" is also the case for the study: only a narrow window including the reflectance near the bedrock is considered, and the spatial context, i.e., along the bedrock, is only considered through a rather pragmatic postprocessing step that filters the results spatially. Deep learning is a powerful tool to consider these spatial relationships directly. If not adapting the methodology to actually rule out "subjective factors" and have a "complete interpretation of basal radar reflectance features", I would suggest a more elaborate and precise discussion of other methods, to illustrate more in detail in which aspects the proposed methodology is better.

**Reply:** Thanks for your suggestion. We have appended precise discussions of each conventional method, and have modified the context about the advantages.

Line 37: I would suggest an easier rephrasing: "In recent years, deep learning has been applied as a powerful tool to detect different features in IPR images, including bedrock interfaces, internal ice layers, snow accumulation layers". For the "radar semantic segmentation", that is an automated feature extraction in se, so I'd suggest to either refer to what is semantically segmented or remove.

**Reply:** We have updated these sentences to according to your suggestion.

Line 40: I am not sure if I understand the difference between this sentence and the previous: is the previous specifically about the detection of layers? If not, I would try to combine this sentence with the previous one and specify the subglacial features. For me it is not clear whether the subglacial features refer to anything under the surface or just features at the ice- bedrock interface.

Line 42: I would rephrase this sentence with: "Moreover, deep learning applied to IPR has also contributed to estimates of ice thickness (to enable data application in ice sheet studies.)", with the part in brackets potentially removed.

**Reply:** We have combined and simplified these sentences according to your helpful advice.

Line 46: Potentially include a reference to the dataset directly (see: https://data.cresis.ku.edu/#ACRDU)

**Reply:** Done. Thanks.

Line 50: I think it is a bit confusing to use the wording "construct a dataset", it suggests that you collected the data in the field. I suggest the rephrasing: "In this study, we select IPR images in the region of the Gamburtsev Sublgacial Mountains from the CReSIS database. We crop these images around the ice bottom, to obtain a set of one-dimensional waveforms that capture the ice bottom reflectance characteristics. Using this data, we train ..."

**Reply:** Thank you for your suggestion, these modified sentences read much better.

Line 52: The "time-domain waveform features" are confusing. Either introduce the time-domain aspect in an additional sentence (something like: "The radar is reflected most strongly by the bedrock beneath the ice sheet, resulting in a peak in the return signal received by the radar over time. Moreover, bedrock characteristics, such as roughness or the presence of water, influence the intensity and shape of the peak signal, to which we refer to as the waveform features of basal reflectors.")

**Reply:** Thanks for your indication. Here we modified the "time-domain waveform features" to "one-dimensional waveform features" to contain the continuous logic with the previous sentence.

Line 55: Do you mean the features that correspond to subglacial lakes? Line 55: I would specify that this is a kind of post-processing step.

**Reply:** We have added the specific subglacial lake feature in this sentence. Thanks.

Line 58-60: What is the benefit of extracting reflectors with similar waveform characteristics as water bodies? How does that improve the efficiency and accuracy of the detection of subglacial lakes?

**Reply:** We have separated this sentence into two parts and introduced the benefits of efficiency and accuracy separately. Thanks for the indication.

Line 61: Indeed, it is nice that you can characterize/cluster the subglacial features through this method.

**Reply:** Thanks.

**2 Data and Methods**

Figure 1: The Figure looks nice, and summarizes the workflow well, but there are several details that need to be adjusted: What is "Z-Scope"? What is "A-Scope"? "Ice Buttom" should be "Ice Bottom", "Reconstructed Reflector Feature" should be "Reconstructed Reflector" (as in "Ice Bottom Radar Reflector"). Both waveforms need axes with labels (time and power I guess). For the caption "(b) VAE reconstructs and encoding of the sampled ice bottom reflector features." should be changed to "(b) The VAE encodes and reconstructs the sampled ice bottom reflector." For the subpanel (c), the caption says "Supervised", while I think the authors mean "Unsupervised".

**Reply:** We have modified both the figure and caption according to your detailed indication and helpful suggestions. Thanks a lot.

Line 69: This sentence about the lake inventories seems out of place. I think, together with the sentence "According to the lakes inventory..." on line 71, these sentences should be moved to the introduction in the paragraph that starts on line 50, so that paragraph 2.1 really focusses on the radar data.

**Reply:** We have moved this sentence to the introduction (line 50). This modified version is indeed better. Thank you for the suggestion.

Line 70: I miss a reference here: is it this dataset that's been used? https://data.bas.ac.uk/full-record.php?id=GB/NERC/BAS/PDC/01544

**Reply:** We have added this link in this sentence, thanks for indication.

Line 74: "The radar data were acquired from L1B.." can be rephrased to "We use the L1B data product" to avoid confusion whether the data has been acquired by the authors.

**Reply:** We have modified this sentence.

Line 81: Is there a physical motivation for truncating the signal to this narrow range around the bedrock? When I see the radar images shown in the different Figures (e.g., Figure 4), I find it remarkable to see a distinct reflectance below the bedrock for each of the subglacial lakes that seems to be not captured anymore by choosing the narrow window.

**Reply:** The selection of time window width for truncating the signal is applied based on the experience. We did notice there are some distinct reflectances below the subglacial lake interface reflections, but some subglacial lakes from the known inventory (e.g., the left lake in Figure 9a) do not contain this specific feature. Therefore, we apply a narrower time window to reduce the sensitivity of this additional reflectance. We have appended more description of the motivation for the window width chosen here.

Line 85: Assuming that the peak signal corresponds to a single point, I would guess the length of the truncated signal would be 64 + 1 + 64 = 129, but it reads 128.

**Reply:** Thanks for the indication. We have modified the range to "-64 to +63". The length of 64 is utilized in the raw programming code, in which the index starts from zeros.

Line 88: Could you provide the bandwith/sigma of the gaussian kernel?

**Reply:** We have appended more details about the gaussian kernel. Thanks for the indication.

Line 89: How do you perform this normalization? Somehow I get the impression that all of the nearly 1,5 million (incredible number, congrats!) reflectance traces are normalized individually: or do you calculate a global mean and standard deviation and set these to 0 and 1? If normalized individually, I think this might be the cause of why you need to use the post- processing step where you use the peak power reflectance. I would advise to

either (i) normalize all data with the statistics of the entire dataset as otherwise you're comparing different units to each other, or (ii) already implement the depth/power relationship while normalizing, or (iii), more experimental, normalize each individual waveform, but provide the peak power and the ice thickness as additional input to the VAE.

**Reply:** Yes, the normalization is applied in every single waveform trace. In the early phase of our method concept design, we considered the strategy of all data normalization as you mentioned. However, the VAE failed to learn the waveform in this situation. The potential reason is 2*1 bottleneck was too small to reconstruct the waveform feature consisting of both the waveform shape and dynamic ranges. Thus, we applied single-trace normalization here to simply feature by excluding the dynamic ranges of echo power. According to your indication, we have appended more details about the normalization and its corresponding function in reducing features.

Line 97: What do you mean with the sentence starting with "And the.."? I think it deviates the attention from why you use the VAE: to reduce the dimension of your data.

**Reply:** We agree that this sentence is redundant here and have removed it. We used this sentence to explain the specific feature of VAE, but it seems useless in this paragraph.

Line 102: I think you use it to reduce the dimension of the reflector waveform features from the ice bottom, right? It is confusing to think that the goal is to reconstruct something that you already know.

**Reply:** We have modified this sentence to match the final goal of our VAE application. Thanks a lot for your indication and advice.

Line 104: Your bottleneck consists of a two-dimensional latent distribution, enforced to follow a normal distribution through using the KL divergence in your loss function. I find the motivation for choosing to sample only two samples from your latent distribution just for visual representation weak. Another motivation can be that it is easier to perform the clustering in two dimensions, or that in other work it has been proven sufficient (for example in the referenced work of Li 2022).

**Reply:** Thanks for your advice for updating the motivation of the 2-D latent space application, which indeed we think was weak before. We have appended more descriptions here according to your suggestion.

Line 106: Conceptually I don't understand why the KL is used in the loss function: it forces the latent space to be normally distributed, which is essential when using VAE for generative purposes. However, as the authors want to perform a cluster analysis, I think there is a fundamental conflict. Clustering data that is normally distributed will not yield in clearly separable clusters. Or, differently put: the underlying assumption for clustering should be that there a different clusters, which, of course, can be each normally distributed, but through VAE the latent space is constructed as one single big cluster. The fact that there is no clear cutoff point of the elbow curve that the authors want to use to

determine the number of clusters confirms that there are no separable clusters in the latent space. I have not read enough into the literature to know whether there are other examples of the approach that the authors take that still yield useful results – but a quick search indicated that there are fancy solutions for this mismatching of concepts, e.g., Lim et al., 2020. A simple solution would be to just use an Auto Encoder and perform the clustering on those results.

Lim, Kart-Leong, Xudong Jiang, and Chenyu Yi, Deep clustering with variational autoencoder. IEEE Signal Processing Letters, 27, 231-235 (2020).

**Reply:** We agree that the context here indeed confusing due to the conflicted motivation when using KL in loss function but applied clustering later in the dimension-reduced latent space. The goal of the VAE application in this study is to obtain a continuous-presenting latent space so that we can generate synthetic reflector waveforms (as shown in Figure 3b). After clustering, we can directly choose the cluster that covers the latent space corresponding to the subglacial lakes' feature. This goal is also the motivation we would like to exhibit Figure 3b. Thanks for your suggestion. We tested the auto-encoder without variational models and KL in the loss function. The distributions of the same data samples in different auto-encoders are shown below:

[Figure]

Similar to VAE's distribution(a), the result of auto-encoder(b) does not show a distinct trend of the cluster in the latent space distribution. These two distributions indicate that the waveforms may contain no potential clusters by the feature presenting. We have appended this comparison to supplyment Figures, and have modified the confusing description which may cause conflicted goals in the VAE application.

Line 122: Why do you stop training at epoch 10 if the training loss does not descend more after epoch 4? Can you report the generalization error? If the training loss does not decrease, but you continue training (epoch 5-10), you start to overfit to your training data.

**Reply:** Yes, because we noticed the potential overfitting, the final model we applied in the encode and generalization is from epoch 4. We have appended the training loss curve until epoch 4 of both the training and validating datasets in Supplemental Figure S1(unfortunately, we lost the loss information between epochs 5-10). Thus, we additionally repeat the training of VAE using the same dataset until epoch 10 to demonstrate the potential overfitting as shown right-side. We have modified the descriptions of the epochs in training. Thanks for indication

[Figure]

Line 123: The word "evaluate" suggests a quantitative estimation, for example based on independent test data. Could you either provide this, or change to "illustrate"?

**Reply:** We have changed the word "evaluate" to "illustrate" here. Thanks for the indication.

Figure 2: Could you provide axes and labels for all subpanels? Could you provide the MSE for all examples? Potentially the learning curve, and the generalization error could be included in this Figure.

**Reply:** Thanks for your suggestion. Because of the dynamic normalization, in all the subpanels' vertical axis is unified to 0-1. Here, we would like to better exhibit the waveform difference between the raw and reconstructor, so we simplified the axes and label. According to your suggestion, we have appended more description to the caption of Figure 2 and detailed the MSE value for all examples (Due to the missing label of raw data, we have replaced the examples in the raw version with similar waveforms).

Line 139: These vectors consist of two samples from the latent distribution, right?

**Reply:** Yes. We have appended 'from two reflector samples' in this sentence.

Line 143: How does this subset vary from the validation subset mentioned in line 119? It seems like you are going to use these samples for clustering and not for "validate the encoder"?

**Reply:** Yes, we use these samples for clustering, instead of validating the encoder. We are sorry for the mistake and have amended this sentence.

Line 147: That gaussian distribution poses problems for the clustering (see earlier remark about line 106).

**Reply:** We have appended more descriptions about the gaussian distribution and the motivation of the clustering applied.

Line 148-153: This is almost philosophical, could you rephrase it with more direct wording?

**Reply:** We have modified this sentence. Thanks for indicating.

Line 156: I do not directly see that 2000 reflectors are sufficient for clustering. From Figure 3, to me, the clusters seem rather arbitrary. Also, given that you have 1.5 million reflectors and you perform the dimension reduction to enable efficient clustering, I think the sample of 2000 is rather small (~0.1 % of all data). How long does it take to perform the clustering analysis?

**Reply:** Thanks for your indication. The clustering analysis takes about 20s. We have used a larger dataset of 0.1 % of all data for clustering according to your advice. However, because of the difference between the data applied, the region of each cluster show a slight difference in latent space, which could impact all the following results and requires huge works on replotting figures and map. Therefore, we consider appending an additional comparison of the detected range of subglacial lakes in same radar image in Figure 11. We agree that using a larger amount of samples can provide more reasonable clustering results. We have appended more description about the comparison in detection ranges and potential improvement in the discussion.

Figure 3: The generative capacity of VAE is nice, and Figure 3b is a pretty visualization of this capacity. However, I do miss a link to the physical phenomenon, and therefore I would suggest to remove the subfigure or move it to Supplementary Materials.

**Reply:** Thanks for your advice. The purpose of this subfigure is to demonstrate the shapes of waveforms corresponding to the vectors in different clusters of latent space. Based on these reconstructed waveforms, we can select the cluster of reflectors which visually similar to the ice-water interface. Thus, we think this subfigure is relatively necessary here. This subfigure also provides a potential reference for the reflector waveform in the subglacial lake cluster (in black color). We have appended more descriptions about the purpose of this subfigure. We have also appended additional color blocks in the background to demonstrate the boundaries between different clusters.

Line 169: Here I miss evidence for the statement: what motivates the authors to conclude that there is an effective separation of bottom reflector features? And how do they correspond to different conditions?

**Reply:** We agree that the statement is missing here. We have modified the conclusion of the effective separation of bottom reflector features. We have removed this arbitrary conclusion and modified the sentence.

Line 171-183: Similar to Figure 3b: a physical interpretation is lacking, and I would move this to Supplementary Materials.

**Reply:** Thanks for the advice. We have modified Figure 3b, and have appended an additional background color block to indicate the boundary between different clusters. We have also modified the descriptions here.

Line 184: In this section the authors discuss how to detect subglacial lakes using the results of the clustering analysis. The main points discussed are related to post-processing steps, and I think this is not clearly reflected in the section title. Potential other titles could be "Subglacial lake detection" or "Post processing to detect subglacial lakes".

**Reply:** Thanks for your advice, we have modified the title of this section to "Subglacial lake detection".

Line 192: I do not understand the conclusion here. I guess you want to say that one of the clusters seems to correspond to subglacial lakes, right? Another way to confirm this is to give statistics of to what clusters the waveforms at earlier detected subglacial lakes belong, e.g., 80% of known subglacial lakes have a bottom reflector that falls into cluster x.

**Reply:** Sorry, maybe we missed the position of these sentences due to the mismatching line numbers. Maybe the paragraph that causes your confusion is the first paragraph in this section. We agree with your opinion here. However, as we mentioned in the reply of major concern, only points of location subglacial lakes are provided by the inventory, which causes difficulties in tracing the reflectors corresponding to the known lakes. According to your advice, we have modified these primary conclusions.

Line 198: What do you mean by "based on experimental experience"? Is there a reference? A solution could be to remove that specification.

**Reply:** Thanks for your advice, we have removed this sentence. The experimental experience was from the final result analysis after this step. We filtered the small subglacial lakes with a threshold on the lake range and compared the result with the known lake inventory. After multiple attempts, we finally chose this value.

Line 203: What do you mean by "interpolation artifacts due to specific noise?"

**Reply:** Thanks for your indication of this redundant description. We have modified and simplified this sentence to "mistaken detection caused by abundant interpolation"

Line 209: If I understood it well, before you used this peak echo power to normalize the data for the encoder. I wonder if this postprocessing step would still be necessary if don't apply this normalization earlier. That would potentially be something to investigate and report on.

**Reply:** We agree that there was potential content that needed to be reported. We did apply the raw signals without normalizations in the VAE training. However, the VAE failed to reconstruct the input signals. The potential reason is the raw signals before normalizations contain more features (especially for the peak echo power), so it is relatively more difficult to reconstruct by an auto-encoder with a smaller size of latent space (bottleneck). Therefore, we applied power normalization for all the reflector waveforms before VAE training. We have applied more descriptions about the motivation of normalization in the VAE section, as well as the peak echo power postprocessing. Thanks for your advice.

Figure 4: For panel d, would it be possible to have the same colors as panel c? So black for the lake, and other colors corresponding to the different clusters that have been filtered out during the post processing?

**Reply:** Thanks for the advice. We have modified these figures by changing the colormap on panel d.

Line 211: How did you calculate the best linear fit? Somehow, I get the impression that the orange dashed line should be steeper in Figure 5, but this might be an optical illusion.

**Reply:** We used LinearRegression module from scikit-learn toolkit in Python. We did notice the mismatch of the steep on fitting, which we considered as the algorithm difference between linear fitting and probability density estimation.

Figure 5: Potentially only show the +1 sigma as that's the threshold you use, to avoid confusion.

**Reply:** Thanks for your advice, We have modified this figure and removed the dashed line of +2 sigma.

**3 Results**

Line 229-230: If I understand it correct you are claiming that the results are reliable because the subglacial water bodies look like known subglacial waterbodies, right? Out of interest, what do you mean by the geothermal environment in adjacent areas?

**Reply:** Thanks for your indications. In this sentence, we would like to describe that the geothermal and subglacial environments should be similar in the same radar image, which was continuously recorded in adjacent areas. We have modified this sentence to be more readable.

Line 237-240: This statement is very similar to the statement in the previous paragraph. I think you do not need to convince the reader of the value of an automated method for detection, it is already clear that this is very valuable.

**Reply:** We have removed this redundant description about the automated advantages. Thanks for your advice.

Line 241: I think it should be "(at about 40 km along the transect)" or so, it looks like the lake is ~3 km wide.

**Reply:** Thanks for the suggestion. It did look better after modification.

Line 241-253: Nice discussion of results.

**Reply:** Thanks.

Line 255-260: Somehow this paragraph makes me doubt that for the results in Figures 4, 6, and 7, the peak power post-processing step is not applied? Could you clarify that in the text?

**Reply:** The post-processing of the peak power threshold helps to obtain the weak reflections. Results in Figures 4, 6, and 7 show strong reflections and are therefore validated in this post-processing. We have appended more context about the post-processing step. Thanks for your advice.

Line 260: By "sparsely detected", do you mean that these are isolated lakes? Or just along a single IPR line?

**Reply:** Yes, we have modified this description.

Line 261: Normally it should be "compare to something": rephrase as "We compare the subglacial lakes detected in this study to the previously identified ..."

**Reply:** Thanks for the indication. We have amended that.

Line 265: remove "which is newly detected", that is already clear from the first part of the sentence.

**Reply:** Done.

Line 277: Do you mean that the red arrows show lakes that have not been detected?

**Reply:** In this context, the red arrows indicate other continuous reflector features within the same cluster, though they do not correspond to the subglacial lakes cluster. We noticed the context near this sentence may cause confusion. Thus, we have modified this part, separated this sentence into a new paragraph, and added more description about this radar image.

Line 278: In Figure 7c you associate the yellow cluster with frozen-on ice and ice flow dynamics. But in Figure 9 it looks like different shades of purple. Do you think multiple clusters do show this frozen-on ice? And are these clusters next to each other (it's hard to link the shades of purple in the Figures with the shades of purple in Figure 3a).

**Reply:** Yes. We consider that different clusters (which appears continuously in radar reflectors) may correspond to different phases or situation of frozen-on ice. However, the relations still need further studies and field observations. We have appended a color block in Figure 3b to demonstrate the adjacent relation of different clusters. From the Figure 3b, we notice these clusters are next to each other.

Line 280: I think the origin of the water bodies is very suggestive. What do you mean by the sparse but regionally dense distribution of subglacial water bodies?

**Reply:** We have removed this confusing description of the "sparse but" and modified that to "the regionally dense distribution of subglacial water bodies". Thanks for the indication.

Figure 8: I think the Figure is very essential for the study. It took a long time to understand the link between the regions and the labels, but I understand now that it is related to the thin black arrows. Potentially it would be nice to clarify that in the main text, as well as in the caption. Moreover, the two blue colors (blue and cyan), might be confusing, and the labelling can be "detected lakes (no post-processing)" and "lakes (post-processed)" or so, now it is not clear what is what exactly. Other questions that pop up when seeing the figure are: (i) in the region near "N3", going perpendicular to the radar lines, there is a clear line of lakes, does that correspond to a kind of channel in the subsurface topography? It could be interesting to overlay the detections on bed topography data, but that is probably out of scope for this study. (ii) There are a lot of "candidate lakes" on the southern part of the survey, it almost looks like an artifact, is that the case?

**Reply:** We have appended more descriptions to the main text about the markers used in the map. According to your advice, we have modified this map in both color usage and caption. For question (i), we agree that will be an interesting illustration by tracing the nearby subglacial lakes in radar images and comparing them with bed topography data. There will be the next studies after our arranging of the new subglacial lake list. For question (ii), the candidate lakes on the southern part are invalidated by the echo power filtering. Flatten topography with weak reflections is exhibited in the radar image in this region, which mismatches the features of subglacial lakes. We have appended more descriptions of this abnormality.

Line 287: What do you mean by "differ visually"?

**Reply:** It should be "visually different from...". We are sorry for this confusing description.

Line 298-304: I think the conclusion is very bold, basically saying that the previous inventories are wrong in places where the authors do not detect lakes. I would be a bit more reserved and steer in the direction that this automated method is promising, and that further investigation is needed (as already suggested). Moreover, there is the remark about "multi-trace detection methods", but in some sense the applied post-processing of grouping 8 neighboring traces makes this method also a "multi-trace detection method", right? Or is this not applied for obtaining the map?

**Reply:** Thanks for the indication. We agree that this conclusion is too bold. We have modified these sentence, and appended more context about the automated method application in updating the lake inventory. According to your advice about "multi-trace detection methods", we have modified the sentence and appended more details about the "multi-trace method".

**4 Discussion**

Line 307: I understand what you mean by "all reflection information", but actually you crop the reflectance to contain only the signal of the bottom.

**Reply:** Thanks for your indication. We agree that the reflectance was cropped. Therefore, we have modified "all reflection information" to "ice bottom echo waveform information" according to your advice.

Line 308: I miss a sentence that states what has been done, something like "We encoded the waveforms to obtain two-dimensional vectors that conceptually summarize the waveform in the so-called latent space of an auto-encoder. The distance between vectors in the latent space..."

**Reply:** Much appreciate your helpful supplement. We have added this sentence here, which greatly improves the context.

Line 328: What do you mean by this sentence? The clustering analysis can be used as input for other models?

**Reply:** We agree that the usage of "data" may cause confusion. Similar to the subglacial lakes, reflectors classified in the same clusters by the analysis may correspond to different subglacial environments. Although the relations between cluster and environment are still waiting for further study. We think this primary clustering analysis can reduce the data complexity for other models.

Line 330: What do you mean by "an automated analysis data"? "automated analysis of the data"?

**Reply:** We have modified this context. Thank you for the indication.

Line 336: "As such, the method has potential.."

**Reply:** Thanks for the indication, we have amended this sentence.

Line 337: What do you mean by classifications for single-track radar data?

**Reply:** The "classifications" here means "analysis", and "single-track radar data" means the reflection waveform from single-trace radar observations. We have modified the description and appended a citation here. Thanks for the indication.

Line 339: Sorry for the noob question: does ice penetrating radar on Mars exist? Can you obtain those kinds of observations from space? And in general, DL methods are known to perform badly on out-of-distribution examples, so is it realistic to apply the method to data that is very dissimilar from airborne observations?

**Reply:** Thanks for the indication. We have modified this sentence to "provide a potential reference for analyzing ...", and have appended more missing citations here. There are public data on radar-sounding observation from Mars, such as the SHARAD[1] and MARSIS[2]. Some observation tracks from orbit have covered Mars' southern ice cap[3]. Studies (e.g.,[4]) have discussed the detection of candidate martian subglacial water

bodies. We agree with the potential challenges in transferring the model on out-of-distribution examples, thus we have modified the discussion here.

[1] Seu, R., Phillips, R. J., Biccari, D., Orosei, R., Masdea, A., Picardi, G., ... & Nunes, D. C. (2007). SHARAD sounding radar on the Mars Reconnaissance Orbiter. Journal of Geophysical Research: Planets, 112(E5).

[2] Picardi, G., Biccari, D., Seu, R., Plaut, J., Johnson, W. T. K., Jordan, R. L., ... & Zampolini, E. (2004, August). MARSIS: Mars advanced radar for subsurface and ionosphere sounding. In Mars express: The scientific payload (Vol. 1240, pp. 51-69). (https://pds-geosciences.wustl.edu/missions/mars_express/marsis.htm)

[3] Orosei, R., Lauro, S. E., Pettinelli, E., Cicchetti, A. N. D. R. E. A., Coradini, M., Cosciotti, B., ... & Seu, R. (2018). Radar evidence of subglacial liquid water on Mars. Science, 361(6401), 490-493.

[4] Carrer, L., & Bruzzone, L. (2021). A novel approach to the detection and imaging of candidate martian subglacial water bodies by radar sounder data. IEEE Transactions on Geoscience and Remote Sensing, 60, 1-15.

**5 Conclusions**

Concise, clear

**Reply:** Thanks.

**Data availability**

Will you share your clustered data, i.e., the data in Figure 4, 6-10? Will you share your code in a repository?

**Reply:** Thanks for the suggestion. We are still arranging and packing the code and results. We will update the open-source information about both the data and code in this section before the final publication.

**Technical comments**

Line 21: introduce the acronym IPR here

**Reply:** Done.

Line 21: "subsurface features"

**Reply:** Done.

Line 37: remove (DL), acronym is not used often in the paper, and it complicates reading. Line 42: "These deep learning-based approaches"

**Reply:** Modified.

Line 66: remove "reduction", add "the" before variational auto=encoder

**Reply:** Done.

Line 116: "n" instead of "N"

**Reply:** Done.

Line 149: Brackets around Kingma and Welling 2013

**Reply:** Done.

Line 185: "different type's ice bottom" should be "different types of ice bottom"

**Reply:** Done.

Caption Figure 4: "Fist example" instead of "Example 1"

**Reply:** Modified

Caption Figure 4: "Results of the unsupervised clustering of the latent vectors"

**Reply:** Done.

Line 220: "dataset" instead of "database"

**Reply:** Done.

Line 243: Remove "This subglacial ... return power", it repeats the previous sentence

**Reply:** Done.

Caption Figure 7: "continuous" instead of "continus"

**Reply:** Done.

Line 270: "Figure" instead of "Figures"

**Reply:** Done.

Line 335: "covering the Arctic" can be "covering, e.g., the Arctic"

**Reply:** Done.

Line 356: remove "A."

**Reply:** Done. Thanks a lot for indicating the technical issues above.

---

## Author Response (AR1)

**Response to Huw Horgan (Public justification)**

Thank you for your detailed reviewer responses. At this stage I encourage you to implement the changes you have detailed in your response. I am unclear whether you intend to implement the methodology changes suggested by Reviewer 1 (RC1) in their comment "L185-192: Identifying subglacial water using clustering analysis". Please implement these changes or justify why you do not consider this necessary.

> **Reply:** Thanks a lot for your encouragement. We have updated the responses in the revised version (in the following content). According to your indication of RC1's comment on Lines 185-192, we have appended more details about the changes and implements according to this comment. Following the suggestion for the combined water index, we approached a new experiment and provided primary results for the index based on the distance between the centroid of lake vectors and onsite reflectors (shown in Figures 11d and 11f). More discussions of both the index and potential work in the future have been appended in the Discussion section (Line 333-348). We are grateful for your kind indication, as well as the reviewer's helpful suggestion, which improved our manuscript.

Please also make changes or further justify your response to suggestion (ii) provided by Reviewer 2 (RC2) "ii) Instead of the clustering, the authors can identify where the currently known subglacial lakes are located in the latent space (i.e., plot these samples in Figure 3a)." I appreciate your response but the reviewer suggestion is a sensible one and I'm not sure your one trace justification is adequate justification not to follow the advice given.

> **Reply:** Thanks for your feedback on our previous response to RC2's comments about identifying the lakes based on the known lakes' reflectors instead of using the clustering. We agree that this suggestion is sensible and needs more claims in response. In the revised version, we have implemented the measurement of the reflector's difference in latent space (examples shown in Figures 11d and 11f) and plotted the vectors from lake samples in latent space (Figure 11a). Considering the detailed distributions of known lakes, such as the precise lake width in each radar image, were not provided in the latest inventories, we acquired the lake ranges based on the clustering results in this stage. Besides, according to lake distributions in the map and radar images, clustering is effective in separating the lakes' reflectors from non-lake features. Thus, we suggest that clustering at least be applied in detecting a primary subglacial list, and the latent space measurement is a potential future study when more lakes have been entered into the list, as demonstrated in the discussion section (Line 333-348).

Finally, thank you again for your high quality submission and responses. I look forward to receiving your revised manuscript.

> **Reply:** Thank you very much for your comment and help with our manuscript!

**Response to Michael Wolovick (RC1)**

**Overview**

In this manuscript, the authors apply Deep Learning (DL) techniques to the problem of identifying subglacial lakes using ice-penetrating radar data. They use a multi-step method consisting of first encoding the information contained in the shape of each vertical reflector trace into a lower-dimensional latent space, and then applying a clustering algorithm to that space in order to identify populations with similar trace shapes. Of these clusters, they identify the one with a narrow symmetric peak in reflection power as representing subglacial lakes, and they further refine the population of subglacial lakes by performing a simple linear attenuation correction to the bed reflection power. Thus, their final identification procedure can be viewed as containing two parts: one part, the encoding-clustering analysis, focuses on the shape of the reflector trace, looking for reflections that are narrow and symmetric, as would be expected for a specular interface. The second, the attenuation analysis, focuses on the strength of the reflector rather than the shape. The authors apply their method to the AGAP-S dataset from East Antarctica and compare their identified lakes with previously published lake compilations.

This manuscript is appropriate for publication in The Cryosphere. It represents a new method in the analysis of ice-penetrating radar data with machine learning techniques and a new method for the identification of subglacial water. However, before it can be accepted in final form, I think that the authors need to provide more justification and explanation around their choice to use a clustering algorithm and on their choice of a particular number of clusters to use in that algorithm. In the remainder of my review, I first explain my major concern about the clustering algorithm, and then I give detailed comments on the rest of the paper.

> **Reply:** We greatly appreciate your helpful comments and detailed advice regarding our manuscript.

**Major Concern**

My biggest concern with the analysis in this paper is the decision to use a clustering algorithm, which splits the data into discrete non-overlapping clusters, and the arbitrary decision to use 15 clusters. The data presented in Figure 3a do not appear to display any inherent clustering on their own. Rather, the data points appear to vary continuously across the Z1,Z2 plane. This point is confirmed by the author's own elbow curve (supplemental figure 1), which does not display a clear cutoff, and this point is acknowledged by the authors themselves on lines 166-168. Thus, a mode of analysis that breaks the data up into discrete categories may not be appropriate, and the authors' decision to use 15 categories is arbitrary and unsupported. Nonetheless, subsequent steps in the analysis protocol are dependent on the use of a clustering analysis earlier on. The ultimate end state of the analysis- a list of positively identified water bodies- requires

that the data be split into discrete categories at some point. At some point a threshold must be applied to distinguish "water" from "not water", and if the clustering analysis is not used for this purpose, then some other means of setting a threshold must be used. Additionally, if I have interpreted the authors' reflection power analysis correctly, then the clustering analysis may also be needed at this stage as well, since I think that they are averaging reflection power values within contiguous reflectors identified through cluster analysis before analyzing reflection power (but note that their explanation of this part of the method was somewhat unclear, so I am not 100% confident that I have interpreted their procedure correctly; I discuss the need for more clarity around this method in the Detailed Comments section below). Averaging reflection power within contiguous similar reflectors is a reasonable methodological choice, since reflection power can be quite variable along-track. Basically, the authors are using the clustering analysis to identify contiguous regions along the bed that have basically the same reflection trace, and they are defining each of those regions as "one reflector" with a single average reflection power for the purpose of reflection power analysis.

Thus, I face a dilemma: on the one hand, I do not want to recommend that the authors remove the cluster analysis entirely, since doing so may necessitate downstream changes throughout their method, including changes to parts of the method that seem sensible; but on the other hand, the data do not seem to support the use of discrete clusters and the particular choice of 15 clusters is unsupported. The arbitrary decision to use 15 total clusters can be regarded as an indirect means of setting the threshold separating "water" from "not water", since the size of each cluster varies inversely with the total number of clusters. A low total number of clusters will increase the size of each individual cluster, thus increasing the diversity of reflectors in the "water" cluster, while a high total number of clusters will make the "water" cluster smaller. The authors state on line 168 that they have tested different values of K (the total number of clusters), but they do not show the results of those tests in the manuscript.

Therefore, as a minimum condition for publication, I think that the authors should show the results of some of these sensitivity tests. How does the final population of water bodies depend on the choice of K? What percentage of the identified water bodies are robust to the choice of K? Perhaps the subpopulation of water bodies that emerge for multiple values of K could be considered a more robust identification of subglacial lakes. (As an aside, if some values of K result in the clustering algorithm splitting the upper right "water corner" of Z1,Z2 space into two clusters, then that would be a valid argument for omitting those values of K; the K sensitivity test should only include values of K for which there is one unambiguous "water cluster"). Alternatively, if the authors can provide argumentation to justify their particular choice of K=15, then that could also satisfy my concerns, although the authors' own statement on lines 166-168 seems to indicate that they do not believe a particular value of K is supported by the data. It may also be worth plotting the elbow diagram (supplemental figure 1) on log/log axes to see if a corner emerges when the data are plotted in that fashion. Once the authors have either justified

their particular choice of K, explored the sensitivity of their results to K, or both, then I think that this manuscript will make an excellent addition to The Cryosphere.

> **Reply:** Reply: Much appreciate your thoughtful and detailed feedback on our method. We agree that the application of 15 clusters on latent space analysis is arbitrary and unsupported. This number of 15 was selected and applied according to multiple attempts in different K values and their results of subglacial detections and the lake distributions. We finally applied K=15 by comparing the lake distributions with known inventories in maps and visual discriminations in radar images . Indeed, we were still seeking a reasonable method to validate and evaluate the K number when we submitted the primary manuscript. Thus, we would like to acknowledge your helpful advice.
>
> According to your concerns and advice, we have modified the manuscript by the following points:
>
> (1) We have plotted the elbow curve on log/log axes, and have updated this diagram on Figure S1(S2 in the revised version). We agree that although the log/log elbow curve does not have a significant cut-off point, this external plotting could provide an additional reference.
>
> (2) We have appended more content in the discussion section about both the cluster ranges in latent space and subglacial lake detections when different K values are applied (Line 313-332 and Figure 11). The result has shown the sensitivity between detections and K values. We also appended more discussion on the results from different K-values applied.
>
> (3) We measured the difference between reflection waveforms in latent space and provided a similarity index by the latent space distance from the centroid of detected lakes' reflector vectors (Figure 11d and f). We have appended more discussions about the potential applications of the index (Line 333-348).
>
> We agree that introducing reflection power into clustering is a potential study based on the primary 2-D clustering in this study. And we also agree that there will be a dilemma on the method's step, as you mentioned. Besides, the 3-D clustering by adding the reflection power as additional parameters was difficult to display in the 2-D image. Thus, we considered implementing this conception in the next studies. Thank you very much for sharing your helpful suggestions and feedback.

**Detailed Comments**

1. L14 Some of these references aren't really appropriate to use as a general background on subglacial water. Robin 1970 is about ice-penetrating radar, Siegert 2000 is about subglacial lakes, and Pattyn 2010 is about the results of a specific model. Pattyn and Siegert could work, although they aren't necessarily the best citations for this purpose, but Robin 1970 is definitely the wrong reference to use here. Chapters 6 and 9 of (Cuffey

and Paterson, 2010) could be cited here, although I understand that citing a textbook is a bit unsatisfying. Another important reference might be (Robin, 1955).

> **Reply:** We have modified and updated these wrong references (Line 14). Thanks a lot for your indications and suggestions.

2. L21 "...in recent years..."

I don't know if it is fair to describe the use of ice-penetrating radar for detecting the subsurface features of ice sheets as "recent". Maybe this sentence would be better as, "Ice-penetrating radar can be used to detect the subsurface features of ice sheets". Also, this would be a good place for the Robin (1970) reference, not L14. Another good reference might be (Robin et al., 1969) or (Bailey et al., 1964).

> **Reply:** We have removed "...in recent years…", modified this sentence and added more citations (Line 21). Thanks for your kind advice on enriching our introduction.

3. L50: "from the CReSIS"

Should be "from CReSIS".

> **Reply:** We have removed "the"(Line 49). Thanks for your indication.

4. L52: "We then apply K-means clustering method"

This sentence should be reworded in one of the following 3 ways: 1) "We then apply theK-means clustering method", 2) "We then apply a K-means clustering method", or 3) "We then apply K-means clustering methods". Wording (1) applies if there is only 1 version of the K-means clustering method, wording (2) applies if there are multiple versions of the method but you only use 1 of them, and wording 3) applies if you use multiple versions of the method.

> **Reply:** Much appreciate your detailed advice on improving our expression. According to our updated version, we have applied multiple K-means by using different K values. Thus, we reworded this sentence according to (3). We have updated the sentence here (Line 51-53).

5. L54: "We notice a cluster"

"We identify a cluster" sounds better.

> **Reply:** Thanks for your indication, we have updated that (Line 53).

6. L61: "...to detect and label the other clusters..." "The" is unnecessary here.

> **Reply:** We have removed "the" here (Line 59), Thanks.

7. L72: "The radar images also contain the positions of ice bottom reflectors, which were extracted by hybrid manual-automatic method (Wolovick et al., 2013)."

Note that the bed picks produced by (Wolovick et al., 2013) are not the same bed picks included in the CReSIS data release. The AGAP-S data was processed in parallel at both

the Lamont-Doherty Earth Observatory (LDEO) and CReSIS. The results of the LDEO processing are available at: https://pgg.ldeo.columbia.edu/data/agap-gambit. Though the original raw data is the same for both institutions, the code for SAR migration and bed picking was different, and of course different human operators provided the "manual" part of the manual-automatic bed picking. Both institutions used "hybrid manual-automatic" bed picking, but it might be better to cite a CReSIS source here if you are using the CReSIS version of the data.

> **Reply:** Thanks a lot for the detailed interpretation of the data. We received the difference in data, and have updated the citation to CReSIS in this part (Line 74-75).

8. L81: "Second, we apply the reflector position markers in the dataset to truncate the 1-D data within the ±200 sampling points near the reflector position for every single trace along Z-axis."

It might clarify things a bit to say that the reflector in this case is the bed. So maybe, "Second, we use the bed picks in the dataset to truncate the 1-D data within ±200 sampling points near the bed reflector position for every single vertical trace."

> **Reply:** Much appreciate your advice. It looks much better after modification (Line 78-79).

9. L88: Gaussian filter, normalization.

What is the filter width? How are the data normalized?

> **Reply:** Thanks for your indications. The filter kernel sigma=4, and all traces of reflectors are normalized into 0-1 individually. We have added the details about the Gaussian filter and normalization here (Line 87-89).

10. L91: "1488600 1-D Z-axis (A-Scope) radar echo traces."

Were you using the original 1.3 m along-track spacing of the data, or are you working with data that have been downsampled? I did not work with the CReSIS version of the AGAP-S data, but I know that other CReSIS data products are generally released at coarser horizontal resolution than this, and when processing the AGAP-S at LDEO, I downsampled the data by a factor of 10 in the along-track dimension. The original 1.3 m data should have a lot more than 1.4 million traces. If you are using downsampled data, then you should mention that.

> **Reply:** Much appreciate your indication. We checked the manual of the database and confirmed the data has been downsampled as you mentioned. We have amended and updated the introduction of the data product according to the database's manual (Line 71-74).

11. L105-110: MSE

What does MSE stand for?

**Reply:** MSE represents "mean squared error". We have fixed that (Line 109). Thanks for the indication.

12. L132: "...are challenging to be reconstructed..." Change to: "...are challenging to reconstruct..."

**Reply:** Thanks for your suggestion, we have modified this sentence (Line 133).

13. L166-168: "However, the elbow curve does not show a clear cutoff point, possibly due to the distribution of vectors in the latent space (Figure 3a) not displaying a distinct trend of multiple classes."

The "elbow curve" for this method seems to be analogous to an L-curve for inverse problems. Could you please present the elbow curve on log-log axes in addition to the linear axes you used in your supplemental figure? The problem with linear axes for this purpose is that inverse power laws always appear L-shaped on linear axes, despite having no intrinsically preferable value.

Additionally, when I look at Figure 3a, it appears to my eye that the data do not really have any clusters at all. Is that what you meant by "not displaying a distinct trend of multiple classes"? To my eye, it looks like the data are smoothly distributed within the central part of the latent space, with perhaps a greater number of outliers in the negative direction for both Z1 and Z2 than in the positive direction. It sounds as though the lack of visual clustering in Fig 3a is confirmed by the lack of a clear cutoff in the elbow curve. I discuss this issue at greater length in my "Major Concern" section above.

**Reply:** Thanks for your suggestions. We have updated the Elbow curves and contain a log/log axes version (Figure S2b). The distribution in Figure 3a indeed shows no clustering trend according to the points scattered. To emphasize the distribution in latent space, we estimated the probability distribution of vectors in Figure S5a, as shown below. Furthermore, we appended an additional test using auto-encoder without a variational module but however gained a similarly smooth and flat distribution in latent space (Figure S5b). We have appended more content

about the failed elbow curve, K value selection, and latent space distributions in the new paragraphs in the Discussion section (Line 313-333 and Line 349-355).

14. Figure 3b

It might be easier to read and interpret this figure if you used a 10x10 grid of virtual waveforms instead of a 20x20 grid, and then made the individual waveforms twice as large. Additionally, it might be better to make the individual waveforms all black, and then overlay cluster boundaries as lines.

> **Reply:** We have modified the figure according to your advice (Figure 3b). Considering the black color corresponding to the subglacial lake in the clustering colormap, which may fuse the waveforms with the stacked cluster boundaries, we plotted the waveforms in white instead.

15. L185-192: Identifying subglacial water using clustering analysis

It seems as though your major reason for using clustering analysis was to get to this step, where you use your method to automatically identify water. The basic argument you are making here seems to be that the upper right quadrant of Z1,Z2 space contains symmetrical sharp reflectors, and these reflectors are more likely to be water. This argument is simple, robust, and I believe it. But in addition to identifying points in this quadrant using a clustering analysis with an arbitrary number of clusters, it may help to make your analysis more robust if you also constructed alternate metrics to identify points in this quadrant. For instance, you could select traces for which Z1 and Z2 are both more than 1sigma above the mean. Or you could make a combined water index, I=Z1+Z2, and then select points with a high value of this index. Alternatively, you could construct a not-water index by taking the euclidean distance from each point to the upper right corner (+2sigma,+2sigma). These sorts of continuous water indices may help reduce the dependence on an arbitrary choice of K.

> **Reply:** We consider your suggestions highly reasonable, as this approach provides a quantitative assessment of feature similarity. However, certain issues may exist when applying based on the current method, primarily related to the order of steps. In my opinion, the new strategy potentially consists of the following steps:
>
> (1) Determine the centroid in the latent space corresponding to subglacial lakes, based on the detected subglacial lake list.
>
> (2) Assess the certainty of subglacial lakes by calculating the distance/index from the centroid, and apply the judgment to all reflectors.
>
> (3) Detect subglacial lake and record the subglacial lake boundaries based on the extracted reflector locations.
>
> Notably, Step (1) requires the existing distributions of subglacial lakes (requires start/end point to utilize all reflectors for encoding), which can be obtained by

isolating corresponding subglacial basal reflections using known subglacial lake ranges (if the ranges of the lake given). However, the known subglacial lake catalogs typically provide only the location of the lakes, resulting in only single data point (single trace waveform from radar image) available for each regional lake. Conversely, Step (3) can conveniently provide a reasonable range of subglacial lake, allowing us to determine the average centroid and boundaries for Step (1).

This new strategy can be implemented based on the distributions and ranges of subglacial lakes provided by the current clustering method. Thus, we consider that this new strategy will be a potential improvement to our current method. Especially, when widely applying the encoding method in updating the known lake inventory, the new strategy could provide valuable estimations of subglacial reflectors from different regions.

According to your suggestions and the steps above, we preliminarily implemented the "water" index by calculating the distance between the centroid of detected lakes' vectors and reflectors in radar images. Results have been added to Figures 11d and 11f. These water indices indeed can provide a direct identification of the potential lakes. We have appended more discussions on the index and potential applications into the Discussion section (Lines 339-348). Much appreciate your suggestions.

16. L199: "Detected subglacial water bodies should contain a continuous ice bottom segmentation in subglacial water type with a width greater than 8 traces (corresponding to an average spatial distance of 10.4 m)."

You should double-check the along-track spacing of the data product you use. If the data have been downsampled from the original 1.3 m spacing, then your 8 trace threshold will correspond to a longer distance.

Reply: Thanks for your indications. After the updating according to the manual, we have modified the spatial distance to 112 m (Line 196).

17. Figure 4c:

This plot would definitely benefit from a continuous approach to reflector categorization.

The different colors here represent different categories, but it is hard to tell how close each category is to the water category. By contrast, a "water index" would provide a continuous metric that could be displayed here.

Reply: Thanks for the indication. We have updated Figure 3b to demonstrate the adjacent relation of different clusters, which provides references for this Figure. Considering the sequence of steps we discussed above, we provided the "water index" in the additional Figure 11c-f using two radar images from the examples and demonstrated the relationship between indices and different K values applied in the clustering.

18. L204-218, Figure 5: Reflection power analysis

Did you correct the bed returned power for geometric spreading before doing this analysis? Signal loss with depth comes from both attenuation within the ice and from simple geometric spreading with range. The effect of geometric spreading can be calculated and removed.

Additionally, I am curious whether Figure 5 shows the entire dataset, or only a subsample of the dataset? When I did a similar analysis for the 2013 paper, I found many data points that were 3sigma or even 4sigma above the linear best-fit. However, in this figure it looks like you would have perhaps 2 or 3 data points at a 3sigma level, and no data points at a 4sigma level. Is the total sample size smaller here? What exactly is being plotted in Figure 5? Does each point represent a single trace, or does each point represent an along-track average of candidate water bodies? I feel like this method needs a better explanation.

After thinking about it for a bit, my guess is that you have done something like this:

1) Apply clustering analysis to the traces

2) Apply the 8-trace rule to generate contiguous reflectors that all belong to a particular cluster.

3) Compute average peak power for each contiguous reflector

4) Perform the attenuation analysis using this smaller dataset of horizontally averaged power data

Am I correct? Is that the procedure that you followed? If yes, then this should be explained in more clarity. In particular, it should be clear that you used grouped adjacent reflectors according to their cluster, and that you have done this for all clusters, not just the water cluster. If this is what you have done, then that also explains why you find far fewer high-reflectivity outliers than I did in the 2013 paper, since peak reflection power can be highly variable along-track and the averaging process will tend to reduce the amplitude of individual bright spots.

It also seems to me that the arbitrary choice to use 15 clusters will have a big impact at this stage, since it will determine the along-track length of contiguous regions that you average together into the analysis. It would be interesting to see in the sensitivity analysis how changing the value of K affects this part of the analysis.

> **Reply:** The plotted data points were randomly chosen from all the AGAP-S radar images because the full plotting of the sample points will fill the figure into black or other pure color selected. Thus, we reduce the number of data points used in this figure, which perhaps has also reduced the number of sparse points above +3/4 sigma above the linear best fit. We were also confused by the difference between this figure and your 2013 paper, but now we guess that maybe the difference in these figures is caused by the different preprocessing of different raw data sources as you mentioned above. In this figure, we did not correct the bed

returned to power for geometric spreading, because the data download from CReSIS seems to contain no enough information and parameters for correction (actually we are not sure if they were already corrected in preprocessing). Thus we applied +1 sigma, which lowers the threshold in your 2013 paper to ensure a larger tolerance to the uncorrected signal.

The reflection power is directly extracted from the radar image data after $10*log\_10(X)$ processing. So the sparse sample points are applied without averaging in clusters or windows. We apply the average return power as follows: When this threshold line was applied in the subglacial lake detection, the average return power of each detected lake range is calculated. If the average return power is higher than the depth-corresponding threshold, the lake will be collected and recorded in the list.

Considering the simplified implementation of echo power filtering we applied, we have modified the descriptions by emphasizing the simplified processes in the filtering (Line 203-206). We also updated the motivation for using a lower threshold to reduce the effect of uncorrected ice thickness (Line 208-210). Thanks a lot for the detailed interpretation and indication.

19. Figures 4, 6, 7, 9, 10: Radar results figures

These figures would all benefit from being zoomed in on the bed. The vertical scale

could be cropped between 2 km and 4km (or perhaps slightly below 4km, to accommodate the deep lake in Fig 9c). In addition, the color scale of the echograms should be adjusted so that the lower limit is just a bit below the noise floor and the upper limit is closer to the brightest bed. These changes would make it easier to follow along when the text goes into detail about specific features in these figures.

**Reply:** Much appreciate your suggestion for enhancing the display. We have updated the color scales and vertical scales in these figures. The modified version indeed highlights the feature of subglacial lakes in radar images.

20. L260 "...but some are also sparsely detected."

This wording is awkward. Perhaps, "...but some isolated points are also detected."

**Reply:** Thanks for your advice. We have modified that following your guidance (Line 251).

21. Figure 8: map figure

It is hard to tell what the text labels (L#, E#, N#) refer to. Maybe you could move the text labels further away from their targets, and then add annotation arrows pointing from the text to the target?

Additionally, the main map should be bigger and the other elements of this figure should be smaller. There is way too much empty white space in this figure. The main map

containing the central AGAP survey contains all of the important information in this figure. Therefore, that main map should be as big as possible. The other two items in the figure, the inset location map and the legend, can be placed in unused corners of the main map.

> **Reply:** Thanks a lot for your advice. We have modified the map according to your advice: We have moved the text labels further away from their targets and the main region of survey lines and related them with their targets by gray arrows. Besides, we have enlarged the main map, adjusted the ratio of other elements, and moved the inset location map and legend to the left sides.

22. L255-268: L#, E#, N#

What do L, E, and N stand for? Are N1-N4 new subglacial lakes?

> **Reply:** Yes, N1-N4 are the text labels that indicate the position of IPR survey lines containing new subglacial lakes detected (shown in Figure 9). Besides, L1-L3 labels denote the locations of survey lines in Figures 4, 6, and 7. E1-5 corresponds to the mismatch lakes with the known inventory in Figure 10. Thanks for your indications for the potential weak statements of text labels in the map figure. We have appended more details to Figure 8's caption about the meaning of L#, E#, N#.

23. L 278: "Considering the dense distribution of subglacial water bodies nearby, these thicker reflection features are possibly formed by frozen-on ice due to ice flow."

Freeze-on isn't caused by ice flow. Freeze-on is caused by either conductive cooling or supercooling. Perhaps a better way to phrase this sentence would be, "Considering the dense distribution of subglacial water bodies nearby, these thicker reflection features are possibly formed by frozen-on ice that complicates the shape of the near-basal reflection trace."

> **Reply:** Thanks for your indication, we have amended this sentence (Line 276-277).

24. L281: "...the sparse but regionally dense distribution..."

What exactly does "sparse but regionally dense" mean?

> **Reply:** Here we would like to indicate the regionally dense distributed subglacial lakes. We have removed "sparse" and modified the sentence (Line 279).

25. L309: "The unsupervised clustering analysis applied in the latent vectors relies on the implied feature difference of the reflection waveform, effectively excluding subjective and external factors in finding potential classifications of subglacial conditions, and reducing the dependence on model assumptions."

Except for the subjective choice to use 15 clusters. This choice has downstream effects in terms of determining the size of the "water" cluster (because average cluster size should vary inversely with the number of clusters), so this arbitrary choice indirectly determines how much variability in reflector shape you are willing to tolerate while still

calling something "water". Additionally, the choice to use a 2D latent space instead of a higher dimensional space was also arbitrary. All methods require some degree of human choice on the part of the scientists employing the method.

It seems to me that the big advances achieved here are in 1) having a new method to quantify and classify the shape of the reflection waveform, and 2) using that method to help classify the physical setting of the ice sheet bed, particularly by helping to identify subglacial lakes. It is not really fair to say that you have excluded subjective and external factors, those factors simply enter into your analysis in a different way than they do in other analyses.

> **Reply:** We agree that the subjective and external factors still exist in the method. We have removed this sentence and modified the discussions in this paragraph according to your suggestion (Line 308-312). Besides, we have appended more discussions about the subjective factors that are present in our method in this stage (Line 334-336), as the motivation for the discussions about the potential implementation of the water index. Much appreciate your indications and suggestions.

**Response to Veronica Tollenaar (RC2)**

**General comments**

The paper discusses a subglacial lake detection method applied to a region near the center of the continent of Antarctica. With the available data, the problem can be seen as a positive and unlabeled problem, where some subglacial lakes have been outlined in earlier studies (positive labeled examples), while for the remaining area the presence or absence of subglacial lakes is unknown (unlabeled examples). The authors take an unsupervised learning approach to this problem, which is a valid choice.

The unsupervised learning consists of an auto-encoder, which basically reduces the dimensionality of the data, and a clustering, where one of the clusters is assumed to correspond to the presence of a subglacial lake. Although this approach is smart, novel, and has a high potential in delineating subglacial lakes, I see several weakly motivated choices in the methodology that I will also try to outline further through the specific comments per section.

My main issue is that the authors perform a clustering analysis on a (2-dimensionally) normally distributed set of samples. These samples are normally distributed through the applied loss function in the encoder. However, per definition, in this set of samples there is only a single cluster, otherwise the loss function should have allowed a certain number of gaussian distributions in the latent space. This caveat is also confirmed by the fact that there is no clear cutoff point in the elbow function to determine the number of clusters present in the data. In my view there are three potential approaches to adjust the manuscript to overcome these caveats in the methodology.

(i) The authors can illustrate quantitatively that the results are convincing, despite the conceptual problem with the methodology, making the study a pragmatic approach toward subglacial lake detection. With the absence of correctly labeled negative examples (i.e., the absence of subglacial lakes), traditional performance metrics such as precision and accuracy cannot be estimated. Nevertheless, a sensitivity estimate of the results, which is currently not part of the manuscript, can be included.

(ii) Instead of the clustering, the authors can identify where the currently known subglacial lakes are located in the latent space (i.e., plot these samples in Figure 3a). As "the distance between vectors in the latent space can serve as a statistical similarity indicator for reflector features" (Line 308-309), samples within a certain distance from the located latent-space vector of known subglacial lakes can be identified as subglacial lakes.

(iii) The authors could use another approach to deep clustering as discussed in various deep learning literature. The simplest solution would be to use an auto-encoder instead of a variational auto-encoder, despite obtaining a less meaningful latent space in the sense that the distance between latent vectors does not reflect a similarity. Nevertheless, it might appear that there are distinct clusters in the latent space.

I think that through adopting (a combination of) the above approaches, or by taking another approach that overcomes the illustrated problem, the study can significantly contribute to the development of an automated approach for the detection of subglacial lakes. This method will be essential to process the ever-growing amounts of data across the continent (and beyond) efficiently, and the authors already convey this message clearly through an elaborate discussion of their results and informative figures.

> **Reply:** Much appreciate your encouraging comments and valuable suggestions. We have updated the manuscript according to your concerns, as the following points:
>
> (i) We have appended the clustered areas in latent space corresponding to subglacial lakes when different K value is applied in clustering analysis. We also traced the detected ranges of subglacial lakes in different K values applied.
>
> (ii) We traced the distribution of vectors from all detected lakes in latent space (Figure 11a). Based on the distribution, we further measured the difference between the onsite reflectors and the centroid of lake vectors. We have appended two examples for the measurement in Figure 11c-e.
>
> (iii) We trained another auto-encoder which contains no variational module, and used the same reflector samples as Figure 3 to exhibit the latent space distribution. We have appended an additional comparison between VAE and Auto-Encoder on the same samples' latent space distributions and their probability density estimations.
>
> For the potential approach (ii), we greatly agree that locating the known subglacial lakes by vectors in latent space could provide a more reasonable identification. To obtain the vectors of known subglacial lakes, the precise distributions of lakes in each radar image are required, which however were not provided from the lake inventory. On the other hand, the detection via the clustering (e.g., Figure 4,6,7) includes the required ranges of lake distributions. Thus, we made use of the detections from this study to locate the lakes' vectors in latent space. Based on the centroid of lake vectors, the distances that indicate reflector similarity were implemented. Due to the cluster is necessary to obtain the primary lake ranges, we discussed the results in measurement in the Discussion section as a potential approach for future studies especially after more precise ranges of lakes are collected.
>
> Thanks again for your detailed feedback and suggestions to benefit our work.

**Specific comments per section**

**Title and abstract**

1. Title: I think "Subglacial Radar Reflectance" sounds better than "Radar Subglacial Reflector". Also, apart from a very elaborate qualitative analysis of the results, there is no hard or independent evidence that the detected lakes are really lakes, let alone that they are "new", which implies that they were not there before (in time). Leaving the word "new" out of the title solves this issue. Otherwise, rephrasing toward something like "An automated method for subglacial lake detection based on deep clustering" could be nice, but it depends on the intention of the authors.

> **Reply:** Thanks for your nice advice on the title. We have modified the title to "Deep Clustering in Subglacial Radar Reflectance Reveals Subglacial Lakes" according to your advice.

2. Line 3: It is confusing to read that you generate a dataset. Maybe better to rephrase as "In this study, we use available IPR images in the Gamburtsev Subglacial Mountains to extract one-dimensional reflector waveform features of the ice-bedrock interface."

Line 4: The method remains very mystical, maybe good to clarify that you apply a deep learning method to reduce the dimension of the data so that you can perform a cluster analysis.

> **Reply:** Thanks for the indications. According to your advice, we have modified and simplified these sentences. (Line 3-4)

**1 Introduction**

3. Line 13: The sentence does not read well. I would suggest: "Subglacial water, i.e., water between bedrock and ice sheet, is formed through a complex interplay.."

> **Reply:** We have updated this sentence according to your kind suggestion (Line 13). Thanks.

4. Line 15: Potentially also include the recent publication of Kazmierczak et al. in The Cryosphere:

E. Kazmierczak, S. Sun, V. Coulon, F. Pattyn, Subglacial hydrology modulates basal sliding response of the Antarctic ice sheet to climate forcing. The Cryosphere, 16, 4537–4552 (2022).

> **Reply:** We have added this citation (Line 15). Thanks.

5. Line 16-20: The importance of research in subglacial lakes is well outlined, but the order is a bit confusing. I would start with the ice sheet meltwater (following the previous sentence about ice flow and dynamics), then the history of climate change and ice sheet evolution, then the subglacial lake sediments, then the unique lacustrine ecosystems.

> **Reply:** Thank you for your helpful advice, we have modified the order and remerged the sentence(Line 16-20).

6. Line 21: Potentially write out the acronym of radar (radio detection and ranging).

Line 21: Potentially remove "in recent years", the next sentence refers to a publication of 1973.

> **Reply:** We have updated the content according to your helpful suggestions (Line 21). Thanks.

7. Line 22: The sentence starting with "Subglacial water bodies" could fit better in the next paragraph, where these visual features are discussed again.

> **Reply:** We have moved this sentence to the next paragraph (Line 26), thanks for your advice.

8. Line 23: I would swap around the subject and the object of these sentences so that it is easier for the reader to understand that here the authors are going to refer to other measurement techniques: "The thickness of the subglacial water layer and sediment characteristics at the bottom of lakes are also investigated with active seismic surveys (Paden et al., 2010; Arnold et al., 2020) and gravimetry and electromagnetic methods (Studinger et al, 2004, Key and Siegfried, 2017)."

> **Reply:** We have swapped these sentences to the modified version (Line 22-24). Thanks a lot for advising.

9. Line 35: the "subjective factors" are not ruled out in this study: heavy postprocessing is applied and the results are discussed mainly in a qualitative way.

> **Reply:** We have removed "subjective factors" and other related content in this sentence (Line 34). Thanks for your advice.

10. Line 36: the "absence of a complete interpretation of basal radar reflectance features" is also the case for the study: only a narrow window including the reflectance near the bedrock is considered, and the spatial context, i.e., along the bedrock, is only considered through a rather pragmatic postprocessing step that filters the results spatially. Deep learning is a powerful tool to consider these spatial relationships directly. If not adapting the methodology to actually rule out "subjective factors" and have a "complete interpretation of basal radar reflectance features", I would suggest a more elaborate and precise discussion of other methods, to illustrate more in detail in which aspects the proposed methodology is better.

> **Reply:**  Thanks for your indication and suggestion. We agree that "the complete interpretation of basal radar reflectance features" is also absent in this study. Therefore, we removed this expression in this sentence and added an additional sentence following to explain that our methods can analyze the reflections when interpretations are absent.
>
> *"In the past decades, IPR surveys have collected large amounts of radar images, which enable the analysis of basal radar reflectance features even if the interpretation of basal radar reflectance features is absent."* (Line35-37)

Besides, we have appended more precise discussions in the Discussion section to illustrate the advantages of the method in this study (Line 308-312).

11. Line 37: I would suggest an easier rephrasing: "In recent years, deep learning has been applied as a powerful tool to detect different features in IPR images, including bedrock interfaces, internal ice layers, snow accumulation layers". For the "radar semantic segmentation", that is an automated feature extraction in se, so I'd suggest to either refer to what is semantically segmented or remove.

**Reply:** We have updated these sentences to according to your suggestion (Line 37 and 40). Thanks for your kind suggestion.

12. Line 40: I am not sure if I understand the difference between this sentence and the previous: is the previous specifically about the detection of layers? If not, I would try to combine this sentence with the previous one and specify the subglacial features. For me it is not clear whether the subglacial features refer to anything under the surface or just features at the ice-bedrock interface.

**Reply:** Thanks for your indications. The main detection target in this sentence is the subglacial target, especially for identifying the subglacial waters, which is different from the previous sentence. We noticed confusing overlapped content here, and we have simplified and merged these two sentences in this part (Line 40).

13. Line 42: I would rephrase this sentence with: "Moreover, deep learning applied to IPR has also contributed to estimates of ice thickness (to enable data application in ice sheet studies.)", with the part in brackets potentially removed.

**Reply:** We have combined and simplified these sentences according to your helpful advice (Line 40-41). Thanks.

14. Line 46: Potentially include a reference to the dataset directly (see: https://data.cresis.ku.edu/#ACRDU)

**Reply:** Thanks for your advice. We have appended this link to the manuscript.(Line 42)

15. Line 50: I think it is a bit confusing to use the wording "construct a dataset", it suggests that you collected the data in the field. I suggest the rephrasing: "In this study, we select IPR images in the region of the Gamburtsev Sublgacial Mountains from the CReSIS database. We crop these images around the ice bottom, to obtain a set of one-dimensional waveforms that capture the ice bottom reflectance characteristics. Using this data, we train ..."

**Reply:** We have modified the sentence following your suggestions (Line 47-51). Thank you for your suggestion, these modified sentences read much better.

16. Line 52: The "time-domain waveform features" are confusing. Either introduce the time-domain aspect in an additional sentence (something like: "The radar is reflected

most strongly by the bedrock beneath the ice sheet, resulting in a peak in the return signal received by the radar over time. Moreover, bedrock characteristics, such as roughness or the presence of water, influence the intensity and shape of the peak signal, to which we refer to as the waveform features of basal reflectors.")

> **Reply:** Thanks for your indication. Here we modified the "time-domain waveform features" to "one-dimensional waveform features" to contain the continuous with the previous sentence. (Line 51)

17. Line 55: Do you mean the features that correspond to subglacial lakes? Line 55: I would specify that this is a kind of post-processing step.

> **Reply:** Yes, we would like to explain the features of subglacial lakes here. We have added the specific subglacial lake feature in this sentence (Line 54).Thanks.

18. Line 58-60: What is the benefit of extracting reflectors with similar waveform characteristics as water bodies? How does that improve the efficiency and accuracy of the detection of subglacial lakes?

> **Reply:** We have separated this sentence into two parts and introduced the benefits of efficiency and accuracy separately :
>
> *"This automated method can improve the efficiency of the detection of subglacial lakes. By collecting and verifying the waveform characteristics of subglacial reflectors, the accuracy of subglacial lakes can also be improved." (Line 57-59)*
>
> Thanks for the indication.

19. Line 61: Indeed, it is nice that you can characterize/cluster the subglacial features through this method.

> **Reply:** Thanks.

**2 Data and Methods**

20. Figure 1: The Figure looks nice, and summarizes the workflow well, but there are several details that need to be adjusted: What is "Z-Scope"? What is "A-Scope"? "Ice Buttom" should be "Ice Bottom", "Reconstructed Reflector Feature" should be "Reconstructed Reflector" (as in "Ice Bottom Radar Reflector"). Both waveforms need axes with labels (time and power I guess). For the caption "(b) VAE reconstructs and encoding of the sampled ice bottom reflector features." should be changed to "(b) The VAE encodes and reconstructs the sampled ice bottom reflector." For the subpanel (c), the caption says "Supervised", while I think the authors mean "Unsupervised".

> **Reply:** We noticed there were too many useless concepts (e.g., "Z-Scope"/"A-Scope") in this Figure. We have modified both the figure and caption according to your detailed indication and helpful suggestions(Figure 1).Thanks a lot.

21. Line 69: This sentence about the lake inventories seems out of place. I think, together with the sentence "According to the lakes inventory..." on line 71, these sentences should be moved to the introduction in the paragraph that starts on line 50, so that paragraph 2.1 really focusses on the radar data.

**Reply:** We have moved this sentence to the introduction (Line 47). This modified version is indeed better. Thank you for the suggestion.

Line 70: I miss a reference here: is it this dataset that's been used? https://data.bas.ac.uk/full-record.php?id=GB/NERC/BAS/PDC/01544

**Reply:** We have added this link in this sentence (Line 68), thanks for indication.

22. Line 74: "The radar data were acquired from L1B.." can be rephrased to "We use the L1B data product" to avoid confusion whether the data has been acquired by the authors.

**Reply:** We have modified this sentence(Line 71-73). Thanks.

23. Line 81: Is there a physical motivation for truncating the signal to this narrow range around the bedrock? When I see the radar images shown in the different Figures (e.g., Figure 4), I find it remarkable to see a distinct reflectance below the bedrock for each of the subglacial lakes that seems to be not captured anymore by choosing the narrow window.

**Reply:** The motivation for truncating the signals in the narrow windows is to directly isolate the single main waveform of bedrock reflection (such as the waveforms shown in Figure 2b-d). We did notice there are some distinct reflectances below the subglacial lake interface reflections, but some subglacial lakes from the known inventory (e.g., the left lake in Figure 9a) do not contain this specific feature. Therefore, we apply a narrower time window to reduce the influence of this additional reflectance feature. We have appended more description of the motivation for the window width chosen here (Line 80-83).

24. Line 85: Assuming that the peak signal corresponds to a single point, I would guess the length of the truncated signal would be 64 + 1 + 64 = 129, but it reads 128.

**Reply:** Thanks for the indication. We have modified the range to "-64 to +63" (Line 84). The length of 64 is utilized in the raw programming code, in which the index starts from zeros.

25. Line 88: Could you provide the bandwidth/sigma of the gaussian kernel?

**Reply:** We have appended more details about the gaussian kernel. Thanks for the indication (Line 88).

26. Line 89: How do you perform this normalization? Somehow I get the impression that all of the nearly 1,5 million (incredible number, congrats!) reflectance traces are normalized individually: or do you calculate a global mean and standard deviation and set these to 0 and 1? If normalized individually, I think this might be the cause of why you

need to use the post- processing step where you use the peak power reflectance. I would advise to either (i) normalize all data with the statistics of the entire dataset as otherwise you're comparing different units to each other, or (ii) already implement the depth/power relationship while normalizing, or (iii), more experimental, normalize each individual waveform, but provide the peak power and the ice thickness as additional input to the VAE.

> **Reply:** The normalization is applied in every single waveform trace. In the early phase of our method concept design, we considered the strategy of all data normalization as you mentioned. However, the VAE failed to learn the waveform in this situation. The potential reason is 2*1 bottleneck was too small to reconstruct the waveform feature consisting of both the waveform shape and dynamic ranges. Thus, we applied single-trace normalization here to simply feature by excluding the dynamic ranges of echo power. According to your indication, we have appended more details about the normalization and its corresponding function in reducing features (Line 90–91). Thanks.

27. Line 97: What do you mean with the sentence starting with "And the.."? I think it deviates the attention from why you use the VAE: to reduce the dimension of your data.

> **Reply:** We agree that this sentence is redundant here and have removed it. We used this sentence to explain the specific feature of VAE, but it seems useless in this paragraph. Thank you for this suggestion.

28. Line 102: I think you use it to reduce the dimension of the reflector waveform features from the ice bottom, right? It is confusing to think that the goal is to reconstruct something that you already know.

> **Reply:** We have modified this sentence to match the final goal of our VAE application (Line 103). Thanks a lot for your indication and advice.

29. Line 104: Your bottleneck consists of a two-dimensional latent distribution, enforced to follow a normal distribution through using the KL divergence in your loss function. I find the motivation for choosing to sample only two samples from your latent distribution just for visual representation weak. Another motivation can be that it is easier to perform the clustering in two dimensions, or that in other work it has been proven sufficient (for example in the referenced work of Li 2022).

> **Reply:** Thanks for your advice for updating the motivation of the 2-D latent space application, which indeed we think was weak before. We have appended more descriptions here according to your suggestion(Line 104-106).

30. Line 106: Conceptually I don't understand why the KL is used in the loss function: it forces the latent space to be normally distributed, which is essential when using VAE for generative purposes. However, as the authors want to perform a cluster analysis, I think there is a fundamental conflict. Clustering data that is normally distributed will not yield in clearly separable clusters. Or, differently put: the underlying assumption for clustering

should be that there a different clusters, which, of course, can be each normally distributed, but through VAE the latent space is constructed as one single big cluster. The fact that there is no clear cutoff point of the elbow curve that the authors want to use to determine the number of clusters confirms that there are no separable clusters in the latent space. I have not read enough into the literature to know whether there are other examples of the approach that the authors take that still yield useful results – but a quick search indicated that there are fancy solutions for this mismatching of concepts, e.g., Lim et al., 2020. A simple solution would be to just use an Auto Encoder and perform the clustering on those results.

Lim, Kart-Leong, Xudong Jiang, and Chenyu Yi, Deep clustering with variational autoencoder. IEEE Signal Processing Letters, 27, 231-235 (2020).

**Reply:** We agree that the context here may be confusing due to the conflicted motivation when using KL in loss function but applied clustering later in the dimension-reduced latent space. The goal of the VAE application in this study is to obtain a continuous-presenting latent space so that we can generate synthetic reflector waveforms (as shown in Figure 3b). After clustering, we can directly choose the cluster in latent space corresponding to the subglacial lakes' feature. This goal is also the motivation we would like to exhibit Figure 3b. Thanks for your suggestion. We tested the auto-encoder without variational models and KL in the loss function. The distributions of the same data samples in different auto-encoders are shown below (also in Figure S5):

[Figure]

Similar to VAE's distribution(a), the result of auto-encoder(b) does not show a distinct trend of the cluster in the latent space distribution. These two distributions indicate that the waveforms may contain no potential clusters by the feature

presenting. We have appended this comparison to Figures S5, and have appended additional discussions on that (Line 349-355)

31. Line 122: Why do you stop training at epoch 10 if the training loss does not descend more after epoch 4? Can you report the generalization error? If the training loss does not decrease, but you continue training (epoch 5-10), you start to overfit to your training data.

**Reply:** Because we noticed the potential overfitting, the final model we applied in encoding and generalization is from epoch 4. We have appended the training loss curve until epoch 4 of both the training and validating datasets in Supplemental Figure S1 (unfortunately, the loss changes between epochs 5-10 were released by the program after training). Due to the random initial weight in dense layers, we cannot access both the loss in later epochs and the generalization error for the trained network. We are sorry for the absence. As a potential solution, we additionally repeat the training of VAE using the same dataset until epoch 10 to demonstrate the potential overfitting as shown right-side. We have modified the descriptions of the epochs in training. Besides, generalization errors (MSEs) have been appended for each sample in Figure 2b-d for reference. Thanks for the indication and suggestion.

[Figure]

32. Line 123: The word "evaluate" suggests a quantitative estimation, for example based on independent test data. Could you either provide this, or change to "illustrate"?

**Reply:** We have changed the word "evaluate" to "illustrate" (Line 124). Thanks for the indication.

33. Figure 2: Could you provide axes and labels for all subpanels? Could you provide the MSE for all examples? Potentially the learning curve, and the generalization error could be included in this Figure.

**Reply:** Thanks for your suggestion. We have appended the generated MSE above each waveform. Because of the single-trace normalization in the dynamic range, all the subpanels' vertical axis is unified to 0-1. In these subfigures, we would like to better exhibit the waveform difference between the raw and reconstructor, so we simplified the axes and label. According to your suggestion, we have appended more description to the caption of Figure 2 and detailed the MSE value for all examples. Because the raw waveform in the previous version was randomly

selected, we cannot trace the selections for the same waveform. Therefore, we have replaced the examples in the previous version with similar waveforms, to better calculate their generalization error (MSE).

34. Line 139: These vectors consist of two samples from the latent distribution, right?

Reply: Yes. We have appended 'from two reflector samples' in this sentence (Line 141). Thanks for your indication.

35. Line 143: How does this subset vary from the validation subset mentioned in line 119? It seems like you are going to use these samples for clustering and not for "validate the encoder"?

Reply: Yes, we use these samples for clustering, instead of validating the encoder. We are sorry for the mistake and have amended this sentence (Line144). Thanks.

36. Line 147: That gaussian distribution poses problems for the clustering (see earlier remark about line 106).

Reply: We agree with the potential problems in clustering, and have applied additional test according to you earlier suggestion (Figure S5 and Line 349-355). Thanks for your kind indication.

37. Line 148-153: This is almost philosophical, could you rephrase it with more direct wording?

Reply: We have modified this sentence and replaced them as:

*"By measuring the distances between the reflectors' latent vectors, we can estimate the difference in waveforms' features. Furthermore, the distance-based clustering in latent vectors can classify the ice bottom reflector feature with similar features."* (Line 151-153)

Thanks for the helpful feedback.

38. Line 156: I do not directly see that 2000 reflectors are sufficient for clustering. From Figure 3, to me, the clusters seem rather arbitrary. Also, given that you have 1.5 million reflectors and you perform the dimension reduction to enable efficient clustering, I think the sample of 2000 is rather small (~0.1 % of all data). How long does it take to perform the clustering analysis?

Reply: Thanks for your indication and suggestion. The clustering analysis takes about 20s. We agree that using a larger amount of samples can provide more reasonable clustering results. We have used larger datasets (5%) for testing clustering results according to your advice. However, the Gaussian-like distribution of samples in latent space (Figure 3a and Figure S5) could cause an unstable clustering result. We noticed that the region of each time of cluster attempt in different randomly selected data shows a slight difference in latent space, which could impact all the following results and require additional work on

updating results and figures. Besides, keeping the reflectors sample set constant enable direct comparisons on the detected range when different K value is applied in clusterings. We have appended an additional comparison of the detected range of subglacial lakes in latent space, radar images, and a regional map for different K values in Figure 11.

Therefore, we consider keeping a constant reflectors sample set in this study and discussing the variations in cluster results when the K value changes, rather than changing the data amounts in clustering samples. We have appended more descriptions about the comparison in detection ranges in the discussion (Line 313-330). Besides, we also compared the detected ranges of lakes when more data was applied in clustering and K=15 with 2000 samples applied (Black lines in Figure 11c,e, Line 330-333).

39. Figure 3: The generative capacity of VAE is nice, and Figure 3b is a pretty visualization of this capacity. However, I do miss a link to the physical phenomenon, and therefore I would suggest to remove the subfigure or move it to Supplementary Materials.

**Reply:** Thanks for your feedback and advice. The purpose of this subfigure is to demonstrate the shapes of waveforms corresponding to different clusters of latent space. We agree that the physical phenomenon and the generated waveforms lack links. Based on these reconstructed waveforms, we can access the waveforms corresponding to different clusters. These reconstructed waveforms are also helpful for verifying the reflection waveform from the ice-water interface with other methods, such as Hao et al., (2023). Thus, we considered this subfigure could be retained for the readers to access the representative waveform for reflectors in different clusters as a potential reference.  Besides, this subfigure could also provide direct demonstrations for other clusters, such as the potential frozen-on-ice cluster in the following context. Thus, we consider to retain this subfigure. We have appended more descriptions about the purpose of this subfigure (Line 177-179). We have also modified this figure with additional color blocks in the background to better demonstrate the boundaries between different clusters (Figure 3).

Hao, T., Jing, L., Liu, J., Wang, D., Feng, T., Zhao, A., and Li, R.: Automatic Detection of Subglacial Water Bodies in the AGAP Region, East Antarctica, Based on Short-Time Fourier Transform, Remote Sensing, 15, 363, 2023.

40. Line 169: Here I miss evidence for the statement: what motivates the authors to conclude that there is an effective separation of bottom reflector features? And how do they correspond to different conditions?

**Reply:** We agree that the statement is missing here, as well as the evidence of "effective" separation. We are sorry for the inaccurate conclusion in this sentence. We have modified this sentence to "*The clustering in latent vectors separates the ice bottom reflector features with similar waveform features*"(Line 168-170). Thanks a lot for your helpful feedback.

41. Line 171-183: Similar to Figure 3b: a physical interpretation is lacking, and I would move this to Supplementary Materials.

**Reply:** Thanks for your feedback and advice. We agree that this paragraph is redundant especially when lacking physical interpretation. We have removed this paragraph and modified the paragraph above to simplify the expression.

42. Line 184: In this section the authors discuss how to detect subglacial lakes using the results of the clustering analysis. The main points discussed are related to post-processing steps, and I think this is not clearly reflected in the section title. Potential other titles could be "Subglacial lake detection" or "Post processing to detect subglacial lakes".

**Reply:** Thanks for your advice, we have modified the title of this section to "Subglacial Lake Detection".

43. Line 182: I do not understand the conclusion here. I guess you want to say that one of the clusters seems to correspond to subglacial lakes, right? Another way to confirm this is to give statistics of to what clusters the waveforms at earlier detected subglacial lakes belong, e.g., 80% of known subglacial lakes have a bottom reflector that falls into cluster x.

**Reply:** Thanks for your suggestion. Your conclusion is correct. Following your helpful suggestion, we have modified the sentence to "*We initially identify one of the clusters corresponding to subglacial lakes.*"(Line 183) and have modified descriptions in this paragraph (Line 183-186). Besides, the other way to confirm the lakes cluster you mentioned is wonderful. The statistics from the known lakes can make the cluster selection more dependable. However, in this phase, the known lakes in the known inventories only contain the location information, as shown as arrows in Figure 10 and gray point Figure 11b, which limits the number of reflector samples. Only the reflector waveforms from the lakes' center points are available, which may reduce the accuracy of lake identification. We still needed a primary screening to obtain the precise range of each lake and to contribute better data statistics.

In this study, we consider the data catalog to be still limited, thus we directly trace the distribution of clusters in maps and radar images. The result shows the overlapped distribution with the known lakes (e.g., Figures 4, 6, 7, and 8), which we think can substitute the statistics when the precise ranges of lakes are absent.

Therefore, we think this solution is better especially when applied in the next subglacial lake detection in large coverage, and the previous studies (such as this

study) have provided a primary catalog of encoded reflector's wave. Following this pathway of the method, the known catalog could also label a reference region in latent space for a more precise measurement of the similarity of lakes' reflectors.

Much appreciate your creative suggestion.

44. Line 198: What do you mean by "based on experimental experience"? Is there a reference? A solution could be to remove that specification.

**Reply:** Thanks for your advice, we have removed this sentence. The experimental experience was from the final result analysis after this step. We filtered the small subglacial lakes with a threshold on the lake range and compared the result with the known lake inventory. After multiple attempts, we finally chose this value.

45. Line 203: What do you mean by "interpolation artifacts due to specific noise?"

**Reply:** The "interpolation artifacts due to specific noise" actually meant the mistaken interpolation due to the specific distribution of noise. For example, if two noise-caused mistaken detected lake points appear in the head and tailer of a bed reflector region, the interpolation could fulfill the internal part and mistakenly identify the whole line to a subglacial lake. This condition should be avoided, thus we applied an additional check here. According to your question, we noticed it may be a trifling detail for the algorithm. We have modified and simplified this sentence to "*mistaken detection caused by abundant interpolation*"(Line 199). Thanks for your question and indication of this redundant description.

46. Line 209: If I understood it well, before you used this peak echo power to normalize the data for the encoder. I wonder if this postprocessing step would still be necessary if don't apply this normalization earlier. That would potentially be something to investigate and report on.

**Reply:** We agree that there was potential content that needed to be reported. We did apply the raw signals without normalizations in the VAE training. However, the VAE failed to reconstruct the input signals. The potential reason is the raw signals before normalizations contain more features (especially for the peak echo power), so it is relatively more difficult to reconstruct by an auto-encoder with a smaller size of latent space (bottleneck). We are also considering the higher dimensions in the bottleneck could pass more features for complex waveform reconstruction from the latent space. However, 3-D latent space may hinder 2-D plotting and challenge the clustering. Thus, we kept the 2-D latent space in this study to investigate waveform shape only. Therefore, we applied power normalization for all the reflector waveforms before VAE training.

According to your suggestion and feedback, We have applied more descriptions about the motivation of normalization in the section "ice bottom reflector"(Line 88-91), as well as the peak echo power post-processing. Thanks a lot.

47. Figure 4: For panel d, would it be possible to have the same colors as panel c? So black for the lake, and other colors corresponding to the different clusters that have been filtered out during the post processing?

> **Reply:** Thanks for the advice. We have modified figures 4, 6, and 7 by changing the colormap to the same as panel d, where black is for the lake, yellow is for the interpolated interruptions, and white corresponds to the non-lake clusters.

48. Line 211: How did you calculate the best linear fit? Somehow, I get the impression that the orange dashed line should be steeper in Figure 5, but this might be an optical illusion.

> **Reply:** We used LinearRegression module from scikit-learn toolkit in Python. We did notice the mismatch of the steep on fitting, which we considered as the algorithm difference between linear fitting and probability density estimation.

49. Figure 5: Potentially only show the +1 sigma as that's the threshold you use, to avoid confusion.

> **Reply:** Thanks for your advice, We have modified this figure and removed the dashed line of +2 sigma (Figure 5).

**3 Results**

50. Line 229-230: If I understand it correct you are claiming that the results are reliable because the subglacial water bodies look like known subglacial waterbodies, right? Out of interest, what do you mean by the geothermal environment in adjacent areas?

> **Reply:** Thanks for your indications. In this sentence, we would like to describe that the geothermal and subglacial environments should be similar in the same radar image, which was continuously recorded in adjacent areas. We have modified this sentence to be more readable:
>
> *"The geothermal and subglacial environments should be similar in the same radar image, which was continuously recorded in adjacent areas." (Line 223-224)*

51. Line 237-240: This statement is very similar to the statement in the previous paragraph. I think you do not need to convince the reader of the value of an automated method for detection, it is already clear that this is very valuable.

> **Reply:** We have removed this redundant description about the automated advantages. Thanks for your advice.

52. Line 241: I think it should be "(at about 40 km along the transect)" or so, it looks like the lake is ~3 km wide.

> **Reply:** We have modified this sentence (Line 233). Thanks for the suggestion. It did look better after modification.

53. Line 241-253: Nice discussion of results.

**Reply:** Thanks.

54. Line 255-260: Somehow this paragraph makes me doubt that for the results in Figures 4, 6, and 7, the peak power post-processing step is not applied? Could you clarify that in the text?

**Reply:** Thanks for your feedback. Results in Figures 4, 6, and 7 show strong reflections and are therefore validated in this post-processing. Most of the failed detections in the post-processing step correspond to long-distance distributed ambiguous and weak reflections. Because of the densely distributed subglacial lakes and strong reflection (obvious feature, such as the ice bottom interfaces in panel a in Figures 4, 6, and 7) in the regions near the survey line of Figures 4, 6, and 7 (L1-3 in Figure 8 map). Most of the subglacial lakes pass in the peak power post-processing step. There is no weak reflection that fails in the peak power post-processing step in the survey line of Figures 4, 6, and 7. According to your question, we have modified the content about the post-processing step (Line 210-212) and appended more content in this paragraph (Line 251-253).

55. Line 260: By "sparsely detected", do you mean that these are isolated lakes? Or just along a single IPR line?

**Reply:** Yes, we have modified this description to 'isolated lakes' according to your suggestion(Line 251). Thanks.

56. Line 261: Normally it should be "compare to something": rephrase as "We compare the subglacial lakes detected in this study to the previously identified ..."

**Reply:** Thanks for the indication. We have amended that.(Line 258)

57. Line 265: remove "which is newly detected", that is already clear from the first part of the sentence.

**Reply:** Thanks for the advice. We have removed that.

58. Line 277: Do you mean that the red arrows show lakes that have not been detected?

**Reply:** Here, the red arrows indicate other continuous reflector features within the same cluster, though they do not correspond to the subglacial lakes cluster. We noticed the context near this sentence may cause confusion. Thus, we have modified this part, separated this sentence into a new paragraph, and added more description about this radar image (Line 275).

59. Line 278: In Figure 7c you associate the yellow cluster with frozen-on ice and ice flow dynamics. But in Figure 9 it looks like different shades of purple. Do you think multiple clusters do show this frozen-on ice? And are these clusters next to each other (it's hard to link the shades of purple in the Figures with the shades of purple in Figure 3a).

**Reply:** Yes. We consider that different clusters (which appears continuously in radar reflectors) may correspond to different phases or situation of frozen-on ice.

However, the relations still need further studies and field observations. We have appended color blocks in Figure 3b to demonstrate the adjacent relation of different clusters. From the Figure 3b, we notice these clusters are next to each other. According to your suggestion, we have appended more discussions here(Line 288-291). Thanks.

60. Line 280: I think the origin of the water bodies is very suggestive. What do you mean by the sparse but regionally dense distribution of subglacial water bodies?

**Reply:** We have removed this confusing description of the "sparse but" and modified that to "the regionally dense distribution of subglacial water bodies"(Line 279). Thanks for the indication.

61. Figure 8: I think the Figure is very essential for the study. It took a long time to understand the link between the regions and the labels, but I understand now that it is related to the thin black arrows. Potentially it would be nice to clarify that in the main text, as well as in the caption. Moreover, the two blue colors (blue and cyan), might be confusing, and the labelling can be "detected lakes (no post-processing)" and "lakes (post-processed)" or so, now it is not clear what is what exactly. Other questions that pop up when seeing the figure are: (i) in the region near "N3", going perpendicular to the radar lines, there is a clear line of lakes, does that correspond to a kind of channel in the subsurface topography? It could be interesting to overlay the detections on bed topography data, but that is probably out of scope for this study. (ii) There are a lot of "candidate lakes" on the southern part of the survey, it almost looks like an artifact, is that the case?

**Reply:** We have appended more descriptions to the main text about the markers used in the map. According to your advice, we have modified this map in both color configuration and caption, such as using "light cyan" to replace "cyan" and using "Detected Lakes (Post-Processed)" and "VAE-Clustering Candidate Lakes (non-echo power filtering)". For question (i), we agree that will be an interesting illustration by tracing the nearby subglacial lakes in radar images and comparing them with bed topography data. There will be the next studies after our arranging of the new subglacial lake list. For question (ii), the candidate lakes on the southern part are invalidated by the echo power filtering. Flatten topography with weak reflections is exhibited in the radar image in this region, which mismatches the features of subglacial lakes. We have appended more descriptions of this abnormality in the first paragraph in this section(Line 253-256). Thanks for your feedback and sharing!

62. Line 287: What do you mean by "differ visually"?

**Reply:** It should be "visually different from..."(Line 285). We are sorry for this confusing description and have fixed that.

63. Line 298-304: I think the conclusion is very bold, basically saying that the previous inventories are wrong in places where the authors do not detect lakes. I would be a bit

more reserved and steer in the direction that this automated method is promising, and that further investigation is needed (as already suggested). Moreover, there is the remark about "multi-trace detection methods", but in some sense the applied post-processing of grouping 8 neighboring traces makes this method also a "multi-trace detection method", right? Or is this not applied for obtaining the map?

> **Reply:** Thanks for the indication. We agree that this conclusion is too bold. We have modified these sentence, and appended more context about the automated method application in updating the lake inventory (Line 300-302). According to your advice about "multi-trace detection methods", we have modified the sentence and simplified the sentence about the "multi-trace averaging"(Line 302).

**4 Discussion**

64. Line 307: I understand what you mean by "all reflection information", but actually you crop the reflectance to contain only the signal of the bottom.

> **Reply:** Thanks for your indication. We agree that the reflectance was cropped. Therefore, we have modified "all reflection information" to "ice bottom echo waveform information" (Line 309) according to your advice.

65. Line 308: I miss a sentence that states what has been done, something like "We encoded the waveforms to obtain two-dimensional vectors that conceptually summarize the waveform in the so-called latent space of an auto-encoder. The distance between vectors in the latent space..."

> **Reply:** Thanks for your feedback. We have appended more discussions about the latent space, where we also added a similar description *"Within the latent space, the difference in reflector features can be measured based on the distance of corresponding vectors from the reflectors. Hence, latent space distance serves as a statistical similarity indicator for reflector features."* (Line 336-337)

66. Line 328: What do you mean by this sentence? The clustering analysis can be used as input for other models?

> **Reply:** Yes, we consider that the clustering analysis or latent space measurements can be used as input for other models to reduce the data dimensions. Thanks for your feedback, we have updated this sentence (370-371).

67. Line 330: What do you mean by "an automated analysis data"? "automated analysis of the data"?

> **Reply:** It should be "automated analysis of the data". We have modified this description (Line 372). Thank you for your kind indication.

68. Line 336: "As such, the method has potential.."

> **Reply:** Thanks for the indication, we have amended this sentence (Line 378).

69. Line 337: What do you mean by classifications for single-track radar data?

> **Reply:** The "classifications" here means "analysis", and "single-track radar data" means the reflection waveform from single-trace radar observations. We have modified the description and appended a citation here (Line 380). Thanks for your indication.

70. Line 339: Sorry for the noob question: does ice penetrating radar on Mars exist? Can you obtain those kinds of observations from space? And in general, DL methods are known to perform badly on out-of-distribution examples, so is it realistic to apply the method to data that is very dissimilar from airborne observations?

> **Reply:** Thanks for the indication. We have modified this sentence to "provide a potential reference for analyzing ..."(Line 370), and have appended more missing citations here. There are public data on radar-sounding observation from Mars, such as the SHARAD[1] and MARSIS[2]. Some observation tracks from orbit have covered Mars' southern ice cap[3]. Studies (e.g.,[4]) have discussed the detection of candidate martian subglacial water bodies. We agree with the potential challenges in transferring the model on out-of-distribution examples, thus we have modified the discussion here (Line 381-384).
>
> [1] Seu, R., Phillips, R. J., Biccari, D., Orosei, R., Masdea, A., Picardi, G., ... & Nunes, D. C. (2007). SHARAD sounding radar on the Mars Reconnaissance Orbiter. Journal of Geophysical Research: Planets, 112(E5).
>
> [2] Picardi, G., Biccari, D., Seu, R., Plaut, J., Johnson, W. T. K., Jordan, R. L., ... & Zampolini, E. (2004, August). MARSIS: Mars advanced radar for subsurface and ionosphere sounding. In Mars express: The scientific payload (Vol. 1240, pp. 51-69). (https://pds-geosciences.wustl.edu/missions/mars_express/marsis.htm)
>
> [3] Orosei, R., Lauro, S. E., Pettinelli, E., Cicchetti, A. N. D. R. E. A., Coradini, M., Cosciotti, B., ... & Seu, R. (2018). Radar evidence of subglacial liquid water on Mars. Science, 361(6401), 490-493.
>
> [4] Carrer, L., & Bruzzone, L. (2021). A novel approach to the detection and imaging of candidate martian subglacial water bodies by radar sounder data. IEEE Transactions on Geoscience and Remote Sensing, 60, 1-15.

**5 Conclusions**

Concise, clear

> **Reply:** Thanks.

**Data availability**

71. Will you share your clustered data, i.e., the data in Figure 4, 6-10? Will you share your code in a repository?

> **Reply:** Thanks for the suggestion. We are still arranging and packing the code and results. We will update the open-source information about both the data and code in this section near the final publication.

**Technical comments**

72. Line 21: introduce the acronym IPR here

> **Reply:** Done.

73. Line 21: "subsurface features"

> **Reply:** Done.

74. Line 37: remove (DL), acronym is not used often in the paper, and it complicates reading. Line 42: "These deep learning-based approaches"

> **Reply:** Modified.

75. Line 66: remove "reduction", add "the" before variational auto=encoder

> **Reply:** Done.

76. Line 116: "n" instead of "N"

> **Reply:** Done.

77. Line 149: Brackets around Kingma and Welling 2013

> **Reply:** Done.

78. Line 185: "different type's ice bottom" should be "different types of ice bottom"

> **Reply:** Done.

79. Caption Figure 4: "Fist example" instead of "Example 1"

> **Reply:** Modified

80. Caption Figure 4: "Results of the unsupervised clustering of the latent vectors"

> **Reply:** Done.

81. Line 220: "dataset" instead of "database"

> **Reply:** Done.

82. Line 243: Remove "This subglacial ... return power", it repeats the previous sentence

> **Reply:** Done.

83. Caption Figure 7: "continuous" instead of "continus"

> **Reply:** Done.

84. Line 270: "Figure" instead of "Figures"

**Reply:** Done.

85. Line 335: "covering the Arctic" can be "covering, e.g., the Arctic"

**Reply:** Done.

86. Line 356: remove "A."

**Reply:** Done. Thanks a lot for indicating the technical issues above.

---

## Author Response (AR2)

**Response to Huw Horgan (Public justification, 2)**

Thank you for the submission of your revised manuscript. I have reviewed your submission and appreciate the time you have spent implementing the reviewer's recommendations.

It is my opinion that your manuscript still requires some changes prior to acceptance. You have included the relevant content but there are now some structural issues as much of the requested information has been placed in the discussion section while it is more suited to methods and results. Specifically I am referring to the method used to select the number of clusters used in your K-means cluster determination. You have done a good amount of work demonstrating your reasoning and the impact of your selection.

> **Reply:** Thanks a lot for your detailed comments, feedback, and suggestions! We appreciate your indications and the solutions to the structural issues, which improve our manuscript. We have moved some contents to the Methods and Result sections and remerged several paragraphs. Hope these changes can address your concerns. Thank you again for your invaluable suggestions!

Specific changes I think should be implemented include:

-Include the details on the number of clusters selection (currently paragraph 2 of your discussion) in your Data and Methods section.

> **Reply:** We have moved part of paragraph 2 of the Discussion section to the Data and Methods section 'Clustering Analysis in Latent Space' (Line 169-173).

- Include details on changing number of subset reflectors in cluster analysis in Data and Methods.

> **Reply:** We have moved the details on changing K number of subset reflectors (Figure S2 and S3) to the Data and Methods section 'Clustering Analysis in Latent Space' (Line 173).

-Shift the figures that show the sensitivity to number of clusters from discussion to results.

> **Reply:** We have added a new subsection '3.3 Detection Sensitivity with Number of Clusters' and shifted the figure and descriptions about sensitivity to number of clusters to this subsection (Line 313-338). We also separated the paragraph and remerged to suit the Results section.

-Include the K-means elbow analysis in the main body of the manuscript (Data and Methods).

> **Reply:** We have included the Elbow curves in the Data and Methods section.

-The presentation of the spatial distribution of latent space distance also seems better suited to Results.

> **Reply:** We agree with you. We have moved the sentences about the latent space distance to the Results section (Line 329-338). We also retained the second half of the part about latent space distance in the Discussion section to discuss the potential application (Line 345-350, as well as the limitation of the existing subjective elements).

Smaller technical changes:

1. L3 'in the' > 'from the'

> **Reply:** Done. Thanks for your advice.

2. L6 'are used' > 'are then used'

> **Reply:** Done. Thanks.

3. L7 (and elsewhere) 'known lakes inventory' > 'known-lakes inventory'

> **Reply:** Done. Thank your for your indication.

4. L23 Paden and Arnold references are not the correct references here.

> **Reply:** Thanks a lot for your indication. We have amended these incorrect references with "(Peters et al., 2008; Horgan et al., 2012)"(Line 23-24), we are sorry for the mistake.

5. L37-L46 consider combining these short paragraphs with sentence linking existing basal picks to deep learning suitability.

> **Reply:** Good suggestion. We have merged these two paragraphs, removed some inessential citations, and appended a sentence linking the deep learning to the large dataset (Line 37-45). Thanks.

6. L50 'Using this' > 'Using these'

> **Reply:** Done. Thanks for you indication.

7. L62 'We will introduce' > 'we introduce'

> **Reply:** Done. Thanks.

8. Figure 1. Should include the spatial and power filtering applied

> **Reply:** Thanks for your suggestion, we have updated both Figure 1 and its caption according to your suggestion.

9. L69 'lake inventory' > 'lake inventories' Correctly bracket references.

> **Reply:** Done. Thanks for your helpful indication.

10. L84 'which maintain' > 'which maintains'

**Reply:** Done. Thanks.

11. L85 'length of 128' > 'length of 128 samples'

   **Reply:** We have added 'samples' following your indication. Thanks.

12. L97 Include a short definition of 'latent space' so as not to lose the non-experts

   **Reply:** We have appended a sentence about a brief definition to 'latent space': "*Between the encoder and decoder, the latent space characters as the 'bottleneck' of the VAE, in which the input feature is depressed into the smallest size.*"(Line 97-98).Thanks for your helpful suggestion.

13. L106 'easier presenting' meaning unclear, please clarify

   **Reply:** We have modified this sentence with '*The 2-D latent space also facilitates the visualization of spatial distributions among the latent vectors in 2-D plots.*' (Line 105-106). Thanks for your feedback.

14. L120 'VAE' > to 'the VAE' (twice)

   **Reply:** Done. Thanks for the indication.

15. L127 'low dimensional bottleneck' briefly define

   **Reply:** We have replaced the 'bottleneck' with 'latent space' to match the preceding description (Line 127). Thanks.

16. L138 '2-length' > 2*1

   **Reply:** Done. Thanks for the suggestion.

17. L144 'intact dataset' meaning unclear, clarify

   **Reply:** We have modified 'intact dataset' with 'entire dataset' (Line 144). Thanks for your indication.

18. L154 '...amount of data used.' citation needed.

   **Reply:** Thanks for your indication. This was the conclusion we drew according to the experience. We noticed that this expression was indeed too subjective here. Thus, we have modified this sentence with "*In clustering, a redundant dataset can slow down the clustering calculation. Conversely, over-reduced datasets may lack essential features and lower the accuracy.*" (Line 154-144)

19. L158 'which based' > 'which is based'

   **Reply:** Done. Thanks.

20. L166 '(Figure S2)' present in main body.

   **Reply:** We have moved Figure S2 (Elbow Curves) to the main body (as Figure 4 in the revised version).

21. L168 'ultimately selected K = 15' justify.

**Reply:** Thanks for your indication, we noticed that 'ultimately' is not a precise expression here, and have amended that with 'finally' (Line 173).

22. L186 'continuously distributed' > 'spatially continuous'

**Reply:** Thanks a lot for your helpful advice that improves our manuscript. We have updated this sentence (Line 188-189).

23. Section 2.4 include subheadings 'Spatial continuity' 'Interpolation' 'Depth-dependent echo power filtering' or similar.

**Reply:** Thanks for your indication. We agree it lacked subheadings in this section and have appended the subheadings for these two paragraphs in this section (Line 193: 2.4.1 Spatial threshold and discontinuity interpolation and Line 203: 2.4.2 Depth-dependent echo power filtering).

24. L190 'In further applications of observational data' Meaning unclear, probably not needed. > 'It has been...'

**Reply:** Thanks for your suggestion, we have removed the redundant sentence here (Line 194).

25. L191 'radar signals.' citation needed.

**Reply:** We have added the missing citation (Hills et al., 2020, Line195). Thanks.

26. L209 'Considering....' Hard to parse this sentence. Please clarify.

**Reply:** Thanks for your feedback. We have updated this sentence with "*Considering the VAE-clustering has analyzed reflector features, and the simplified ice thickness is applied, a lower linear threshold in average echo power ...*"(Line 213).

27. L224 '...similar in the ...' > '...similar at the length-scales covered by the radar image.' Or similar.

**Reply:** Thanks for your great suggestion. We have modified this sentence following your suggestion.

28. L254 '...than the established thresholds.' > '...than the thresholds used here.'

**Reply:** We have modified this sentence according to your suggestion. Thanks!

29. L244 'long-distance' > 'spatially extensive'

**Reply:** Thanks. We have modified this phrases.

30. Figure 9 caption 'subglacial lakes or lakes' and 'regional subglacial lakes' meaning unclear, please clarify.

**Reply:** Thanks for your indication. We are sorry for the typo of 'subglacial lakes or lakes', which has been amended. 'regional subglacial lakes' means 'subglacial lakes with smaller sizes', which we noticed was a repeated explain here. We have removed the additional 'regional' in this sentence.

31. L291 'distinct phases' meaning unclear. Clarify.

**Reply:** Thanks for your indication, 'distinct phases' should be 'different phases' (Line 296). Thanks for you indication. We have amended this wrong description.

32. L297 'Besides, ' > 'Also, '

**Reply:** Thank you for you suggestion, we have modified this sentence following your suggestion.

33. L302 'submerged' > 'obscured'

**Reply:** Done. Thank you.

34. L303 'Besides, the' > 'The.....can also provide'

**Reply:** Done. Thanks.

35. L304 'are essential' > 'will help'

**Reply:** Thanks for you advice. We have replaced that.

36. L305 'miss-detection' > 'false positives'

**Reply:** Thank you for your suggestion. We agree that 'false positives' are better and have updated according to your suggestion.

37. Discussion. Much of this first paragraph belongs in Methods and Results.

**Reply:** We have moved the paragraphs to the Methods and Results sections. Thanks a lot for these helpful suggestions that improve our manuscript.

38. L353 'Compared with' > 'Compared to'

**Reply:** Done. Thanks for the suggestion that improves our manuscript.

39. L372 'automated' > 'mostly automated'

**Reply:** Done. Thanks.

40. L381-382 odd tangent to end on

**Reply:** Thanks for your feedback and indication. We agree that the tangent in the end indeed was quite odd here. Thus, we have removed the sentences about the potential Martian applications in this part.

41. Conclusions. These are very brief.

**Reply:** Thanks for your feedback. We have modified conclusion by adding an additional sentence for conclude the entire workflow (Line 385-386), and have appended more details for connecting reflection feature to latent space (Line 387-389).